# Selective catalytic oxidation of ammonia to nitric oxide via chemical looping

Chongyan Ruan[1,2,5], Xijun Wang [2,5], Chaojie Wang[1,3], Lirong Zheng[4], Lin Li[1], Jian Lin[1], Xiaoyan Liu [1], Fanxing Li [2✉] & Xiaodong Wang [1✉]

Selective oxidation of ammonia to nitric oxide over platinum-group metal alloy gauzes is the crucial step for nitric acid production, a century-old yet greenhouse gas and capital intensive process. Therefore, developing alternative ammonia oxidation technologies with low environmental impacts and reduced catalyst cost are of significant importance. Herein, we propose and demonstrate a chemical looping ammonia oxidation catalyst and process to replace the costly noble metal catalysts and to reduce greenhouse gas emission. The proposed process exhibit near complete $NH_3$ conversion and exceptional NO selectivity with negligible $N_2O$ production, using nonprecious $V_2O_5$ redox catalyst at 650 °C. Operando spectroscopy techniques and density functional theory calculations point towards a modified, temporally separated Mars-van Krevelen mechanism featuring a reversible $V^{5+}/V^{4+}$ redox cycle. The $V=O$ sites are suggested to be the catalytically active center leading to the formation of the oxidation products. Meanwhile, both $V=O$ and doubly coordinated oxygen participate in the hydrogen transfer process. The outstanding performance originates from the low activation energies for the successive hydrogen abstraction, facile NO formation as well as the easy regeneration of $V=O$ species. Our results highlight a transformational process in extending the chemical looping strategy to producing base chemicals in a sustainable and cost-effective manner.

[1] CAS Key Laboratory of Science and Technology on Applied Catalysis, Dalian Institute of Chemical Physics, Chinese Academy of Sciences, 116023 Dalian, China. [2] Department of Chemical and Biomolecular Engineering, North Carolina State University, Raleigh, NC, USA. [3] University of Chinese Academy of Sciences, 100049 Beijing, China. [4] Institute of the High Energy Physics, Chinese Academy of Sciences, 100049 Beijing, China. [5] These authors contributed equally: Chongyan Ruan, Xijun Wang. ✉email: fli5@ncsu.edu; xdwang@dicp.ac.cn

Catalytic oxidation of ammonia to NO is the core reaction for producing nitric acid, one of the top-ten bulk chemicals by annual consumption (over 100 million tons per year) and widely applied in the manufacturing of fertilizers, nylon, dyes, and explosives, among others[1–3]. Since the early 20th century, the Ostwald process ($4NH_3 + 5O_2 \rightarrow 4NO + 6H_2O$), which utilizes Pt-Rh alloy (5-10% Rh) gauzes as the catalysts for ammonia oxidation at high temperatures (800–950 °C) and pressures (up to 12 atm), has been the industrial standard[1,4–9]. The NO yield can reach 90−98% depending on the operation parameters, rendering it one of the most efficient catalytic reactions[1]. However, Pt loss through volatilization (0.05–0.3 g per ton of nitric acid produced) at high-temperature and strongly oxidizing atmosphere results in significant catalytic performance decays and high operating costs[1,6,7]. Moreover, nitric acid production is the largest source of the greenhouse gas $N_2O$ in the chemical industry (125 million tons $CO_2$-equiv per year)[10]. The above economic challenges and environmental concern stimulate research towards developing cost-efficient catalysts and new processes with favorable NO yield while reducing $N_2O$ emission.

Over the past decades, extensive efforts have been devoted to replacing noble metal catalysts with transition metal oxides[2,3,11–18]. A large number of metal oxide catalysts have claimed promising performance for ammonia oxidation, including spinels ($Co_3O_4$)[17,19], perovskites ($ABO_3$ where A = Ca, Sr, La, and B = Mn, Co, Fe)[3,12,13,15] and $K_2NiF_4$ structured oxides ($La_2BO_4$ where B = Co, Cu, Ni)[1]. The industrial implementation is impeded, however, by the insufficient NO selectivity and rapid catalyst deactivation. For instance, $Co_3O_4$ exhibited good activity towards $NH_3$ oxidation with a promising NO selectivity over 90%[1,17,19]. However, the rapid deactivation due to the reduction to less active CoO, despite in the presence of considerable excess $O_2$, has impaired its commercial deployment. Recently, an alternative approach based on oxygen-conducting membrane reactor was proposed, where the $O_2$ and $NH_3$ were supplied at feed and permeate side of the membrane reactor separately[3,15]. Pérez Ramírez et al. demonstrated that by exploiting Ca- and Sr-substituted lanthanum ferrite membranes for the high-temperature $NH_3$ oxidation, high NO selectivities (92–98%) can be attained with $NH_3$ conversion ranges between 80 and 95%[3]. However, the stability of the oxygen ionic transport membranes are often impaired by cation segregation and/or physical degradation. Therefore, it is imperative to research into new intensified processes for $NH_3$ oxidation.

One method towards process intensification is chemical looping, whereby oxidation reactions are decoupled into two-step, spatially separated cyclic redox processes[20–23]. Chemical looping is a new frontier for producing valuable chemicals in a clean and efficient manner[24–28], and has been demonstrated for processing methane[28–35], biofuels[36,37], syngas[38,39], coal and carbonaceous feedstocks[22,40,41] with low cost, reduced emissions and higher energy efficiency. Herein, we proposed and validated the chemical looping ammonia oxidation (CLAO) process. As depicted in Fig. 1, during the reduction step, $NH_3$ is oxidized by the active lattice oxygen derived from the metal oxide redox catalyst, producing concentrated NO. In the subsequent oxidation step, the reduced metal oxide redox catalyst is re-oxidized with air, completing the redox cycle. The CLAO process is potentially advantageous compared to the conventional co-feed approach where $NH_3$ and air are introduced simultaneously. First, since CLAO process decouples the reaction into two, the operation parameters can be optimized independently in order to achieve improved performances. Second, catalyst deactivation could be avoided as the reduced catalyst is regenerated in each oxidation half cycle. Third, the utilization of lattice oxygen instead of nonselective gaseous oxygen may effectively suppress the formation of undesirable $N_2O$, thus reducing the greenhouse gas footprint. That last point is that as large quantities of inert $N_2$ (about 3/4 of the total flow in a current plant)[3] are excluded from the product NO, allowing for a radically intensified process for nitric acid production.

The metal oxide redox catalyst functions as the reactive intermediate to transport oxygen ions between the cyclic steps. Hence, the techno-economic viability of the CLAO technology is critically dependent upon the redox properties of the metal oxide catalyst. By virtue of the excellent catalytic activity in various redox reactions, vanadium oxide catalysts have been intensively investigated in chemical and environmental industries, including oxidative dehydrogenation of alkanes, oxidation of methanol to formaldehyde, mass production of sulfuric acid, and selective catalytic reduction of $NO_x$[42–48]. In recent studies, $V_2O_5$-based redox catalysts have emerged as attractive candidates for oxidative dehydrogenation of ethane via chemical looping, due to the easy accessibility of lattice oxygen and appropriate bonding strength[49,50]. Generally, three types of lattice oxygen can be distinguished within the unit cell of $V_2O_5$ (Fig. S1): (i) singly coordinated terminal oxygen $O^I$, namely vanadyl (V=O); (ii) doubly coordinated oxygen $O^{II}$ bridging two vanadium atoms; and (iii) triply coordinated oxygen $O^{III}$ bridging three vanadium atoms[51,52]. The exact role of different lattice oxygen is crucial in determining the product distribution. However, controversy remains on what type of oxygen is the active site for selective oxidation reactions.

In this study, $V_2O_5$ was employed as a prototype catalyst to investigate the feasibility of the proposed CLAO process given the aforementioned distinct redox properties. The cyclic $NH_3$ conversion activity and $NO_x$ selectivity were evaluated in consecutive isothermal cycles under alternating $NH_3$ and $O_2$ atmospheres. Encouragingly, superior $NH_3$ conversion (97.0%) and NO selectivity (99.8%) were achieved with largely suppressed $N_2O$ emission (0.1%) at a low temperature of 650 °C, demonstrating exceptional efficacy of the CLAO technology. In comparison, the maximum $NH_3$ conversion and NO selectivity in the co-feed mode were only 87.1% and 32.4%, respectively, with a fairly high $N_2O$ selectivity of 16.8% at 600 °C. On the basis of detailed physicochemical characterization, we endeavored to gain a detailed understanding of the $NH_3$ oxidation mechanism complemented by density functional theory (DFT) calculations. In particular, the catalytic roles of different lattice oxygen of $V_2O_5$ were investigated in detail. In situ Raman spectroscopy revealed that terminal oxygen (V=O) being the active center for catalytic $NH_3$ oxidation. Furthermore, XPS and XANES indicated that the reaction involved the reversible reduction of $V^{5+}$ to $V^{4+}$, with V=O participated in the oxidation of $NH_3$ via a modified, temporally separated Mars-van Krevelen (MvK) mechanism. The subsequent re-oxidation of vanadium cation to $V^{5+}$ with molecular oxygen replenished V=O and completed the catalytic cycle. The facile reaction between V=O and $NH_3$ (activation energy: 16.0 kJ/mol) was responsible for the extraordinary $NH_3$ oxidation performance at a relatively low temperature, as substantiated by DFT calculations.

## Results

**Redox performance and stability**. In order to validate the feasibility of the proposed CLAO process, consecutive redox cycles were carried out to investigate the product distribution and the overall chemical looping process efficiency. Figure 2a illustrates the ammonia conversion and product distribution over $V_2O_5$ as a function of temperature in a typical chemical looping mode. The corresponding products evolution upon $NH_3$ injection are reproduced in Fig. S2a–h, which indicates that NO, $N_2O$, and $N_2$

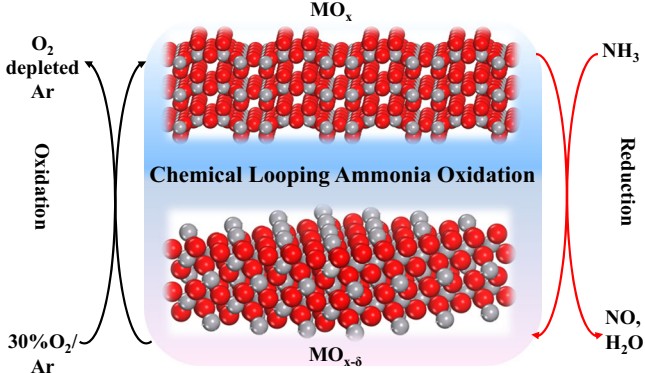

**Fig. 1 Schematic of the chemical looping ammonia oxidation (CLAO) process.** The red and gray spheres represent metal and oxygen atoms, respectively.

were the only N-containing products. While no $H_2$ was observed in the reaction temperature range, water being the only H-containing product that has been condensed out to avoid interference. As depicted in Fig. 2a, ammonia conversion increases from 78.5% to 97.0% at 300–650 °C and plateaus around 97.0% in the temperature range of 580 to 650 °C. Meanwhile, the selectivity to nitrogen-containing products depends critically upon temperature: $N_2$ is the dominant product at low temperature (<500 °C), whereas NO takes over at high temperature (580–650 °C). The selectivity to NO starts at 400 °C and increases continuously with increasing temperature, reaching a maximum of 99.8% at 650 °C. Accordingly, the increase of NO selectivity is coupled with the progressive decrease in $N_2$ and $N_2O$ selectivity. It is worth noting that $N_2O$ formation is greatly inhibited (0.1% selectivity at 650 °C). In comparison, $V_2O_5$ redox catalyst exhibits notably different results in a conventional co-feed mode. As shown in Fig. 2b, ammonia conversion increases from 26.1% to 87.8% at increasing temperature and reaches a plateau from 500 to 650 °C. The maximum ammonia conversion (87.8%) is considerably lower than that in the chemical looping mode (97.0%). In terms of product selectivity, NO peaks at 300 °C with an inferior selectivity of 34.2% and decreases significantly when rising the temperature to 400 °C. A further increase in temperature resulted in increased formation of NO. After passing through a maximum at 600 °C, NO formation slightly decreases with a further increase in temperature. The decreased NO formation with a concurrent increase in $N_2$ formation at higher temperatures (600–650 °C) can be related to the selective catalytic reduction (SCR) of NO with $NH_3$. A similar parallel SCR mechanism was also proposed in a recent study of Decarolis et al.[53], which indicates the oxidation of $NH_3$ to $NO_x$ by PdO and the subsequent catalytic reduction of $NO_x$ by $NH_3$ to produce $N_2$. In addition, a high $N_2O$ selectivity (13.6–26.9%) is observed over the reaction temperature range. Fig. S3a, b illustrates the $NH_3$ oxidation performance with different $O_2$ partial pressure. Furthermore, we also investigated the ammonia oxidation performance for typical single and binary redox metal oxide catalysts in chemical looping mode. The optimized $NH_3$ conversion and selectivity towards NO, $N_2O$, and $N_2$ are compared in Table S1. The $NH_3$ conversions are close to 100% irrespective of the catalyst used. Nonetheless, the NO selectivity is relatively lower than $V_2O_5$ in chemical looping mode and industrially relevant Pt-Rh in the conventional co-feed mode. Hence, superior $NH_3$ conversion can be achieved over $V_2O_5$ via chemical looping with close to complete NO selectivity at 650 °C, which is sufficiently lower than the industrial ammonia oxidation reaction temperature of 810–950 °C over Pt-Rh gauzes. Moreover, the $NH_3$ conversion and NO selectivity for $V_2O_5$ also compared very favorably with those reported in the membrane mode that carried out at

significantly higher temperatures (Table S1), evidencing the great efficiency of the proposed CLAO process.

Kinetically, chemical looping reactions are generally slower than conventional catalytic oxidation/reduction reactions, which are commonly encountered in chemical looping process. This is because the rates of redox reactions are limited by the removal and replenishment of lattice oxygen from/to the bulk of the redox catalyst. The CLAO process proposed here resulted in a normalized NO production rate of ~56.4 ml $g_{catalyst}^{-1}$ $h^{-1}$ (Table S1). The NO production rate of conventional co-feeding mode reported here ranged between 105.2-127.6 ml $g_{catalyst}^{-1}$ $h^{-1}$ (Table S1). While the NO production rates are on the same order of magnitude, the superior $NH_3$ conversion and NO selectivity of CLAO approach can potentially eliminate the costly separation units, ensure safer operation and inhibit nonselective oxidation reactions, which contribute to the economic attractiveness of the $NH_3$ oxidation process. Furthermore, to ensure a relatively faster surface oxygen removal rate, two potential strategies have been proposed for the chemical looping process[23]. One is through doping or lattice site substitution, in an attempt to introduce crystallographic defects that facilitate oxygen ion diffusion. The second strategy is to promote the redox metal oxide with catalytically active species on the surface to decrease the activation energy for reactant activation and to enhance the lattice oxygen consumption from the bulk. Both strategies may open up extensive possibilities for the CLAO process.

To elucidate the dynamics of surface lattice oxygen removal during the reduction step, the initial maximum lattice oxygen extraction rate for $NH_3$ oxidation during the first pulse was determined from Fig. S2a–h, considering the following stoichiometry reaction pathways as no $H_2$ is produced (all the H ends up into $H_2O$, which was condensed out before entering the quadrupole spectrometer):

$$NH_3 + 2.5O_{lattice} \rightarrow NO + 1.5H_2O \qquad (1)$$

$$2NH_3 + 3O_{lattice} \rightarrow N_2 + 3H_2O \qquad (2)$$

$$2NH_3 + 4O_{lattice} \rightarrow N_2O + 3H_2O \qquad (3)$$

And the results are summarized in Table S2. By fitting the maximum oxygen donation rate of $V_2O_5$ redox catalyst with the Arrhenius equation, the apparent activation energy for the surface lattice oxygen removal is estimated to be as low as 16.0 kJ $mol^{-1}$ (Fig. 2c), which is in line with its superior low-temperature $NH_3$ oxidation activity. To gain a quantitative picture of the effect of lattice oxygen extraction on the $NH_3$ conversion and product distribution over $V_2O_5$ in the CLAO process, the $NH_3$ conversion, NO, $N_2$, $N_2O$ selectivity as well as the accumulated lattice oxygen withdrawn from the original sample (δ) is plotted as a function of the $NH_3$ pulse numbers

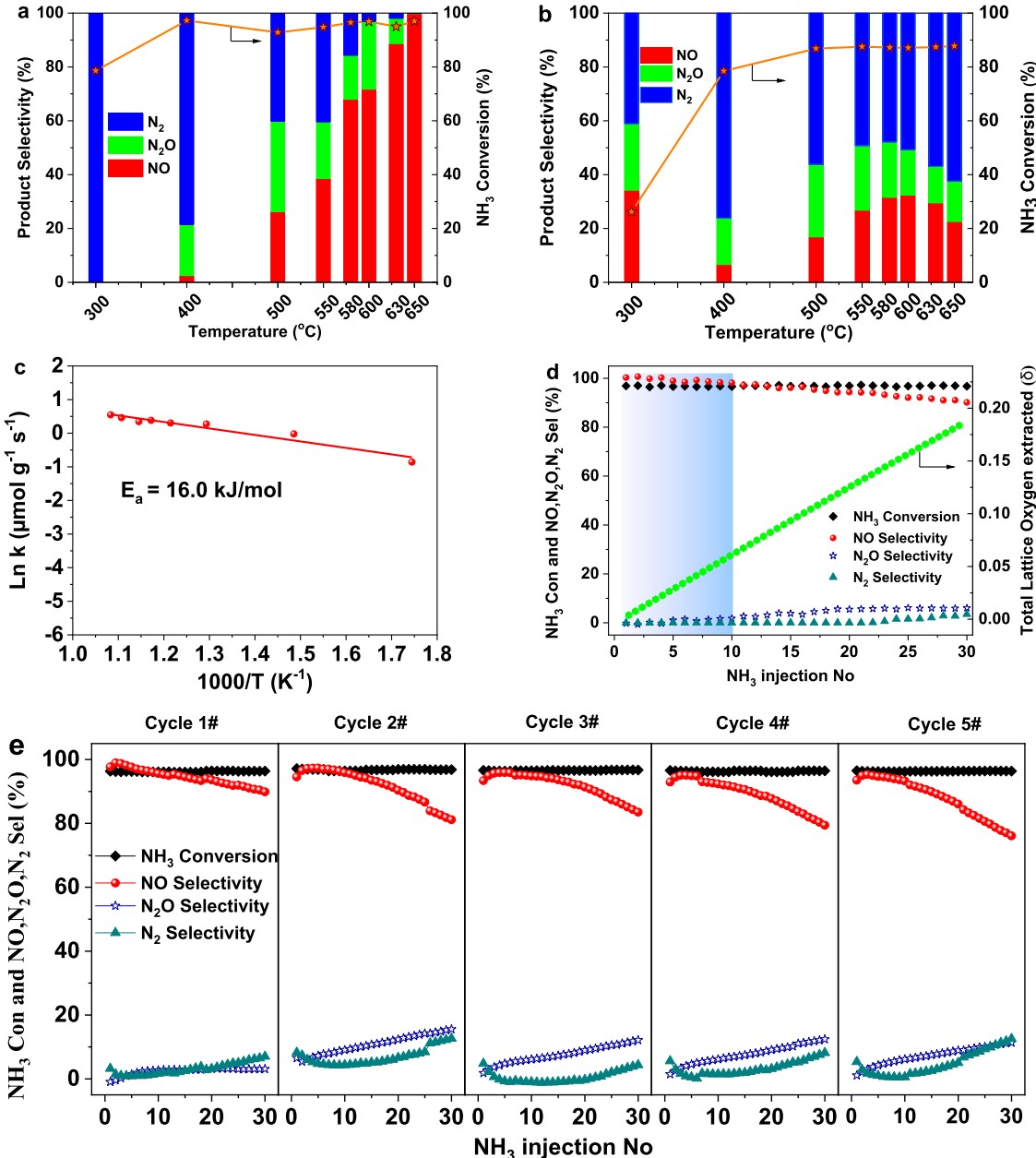

**Fig. 2 Reactivity and selectivity comparison over V₂O₅.** $NH_3$ conversion and NO, $N_2O$, $N_2$ selectivity versus temperature in $NH_3$ oxidation over $V_2O_5$ in **a** chemical looping $NH_3$ oxidation process and **b** steady-state $NH_3$ oxidation (co-feed mode). **c** Arrhenius plots of surface lattice oxygen removal of $V_2O_5$ in chemical looping $NH_3$ oxidation process at 300–650 °C. **d** $NH_3$ conversion, NO, $N_2O$, $N_2$ selectivity as well as total lattice oxygen consumption versus the number of pulses with $NH_3$ over $V_2O_5$ at 650 °C. **e** $NH_3$ conversion, NO, $N_2O$, $N_2$ selectivity versus the number of pulses over $V_2O_5$ at 650 °C during five consecutive chemical looping $NH_3$ oxidation redox cycles.

(Fig. 2d). Starting with a fully oxidized catalyst surface, $NH_3$ conversion and NO formation are initially very high, with $NH_3$ conversion amounts to 97% and NO selectivity approaching 100% in the first 10 pulses. The initial high NO selectivity can be associated with the presence of highly reactive and selective lattice oxygen species. Subsequently, the NO selectivity decreases with ongoing pulse number. The $NH_3$ conversion, in contrast, is practically constant with increasing pulse numbers. The total lattice oxygen consumption increases linearly with increasing number of pulses. It should be noted that $NH_3$ conversion above 97%, NO selectivity above 90% can be achieved, with insignificant $N_2O$ selectivity, even after 30 pulses of $NH_3$. And this corresponds to the catalyst where ~2.4% of the total oxygen is removed, equivalent to overall composition of $V_2O_{5-0.12}$. In

addition, we further explored the $NH_3$ conversion and product distribution over $V_2O_5$ with 180 pulses of $NH_3$. As shown in Fig. S4, the $NH_3$ conversion remains close to 100%. Meanwhile, the $N_2O$ formation is largely inhibited. On the other hand, the NO selectivity decreases accompanied by the increase of $N_2$ formation with increasing pulse number. The decreased NO selectivity can be tentatively attributed to the decreasing amount of active oxygen species available from the catalyst during the sequence. Figure 2e exemplifies $NH_3$ conversion and product distribution over 5 cycles for <1300 min of continuous operation. As can be seen, $NH_3$ conversion remains close to constant over the redox cycles. Despite the decrease in NO selectivity with increasing pulse number, a reoxidation of $V_2O_5$ catalyst fully restores the initial NO selectivity when pulsing $NH_3$, indicating

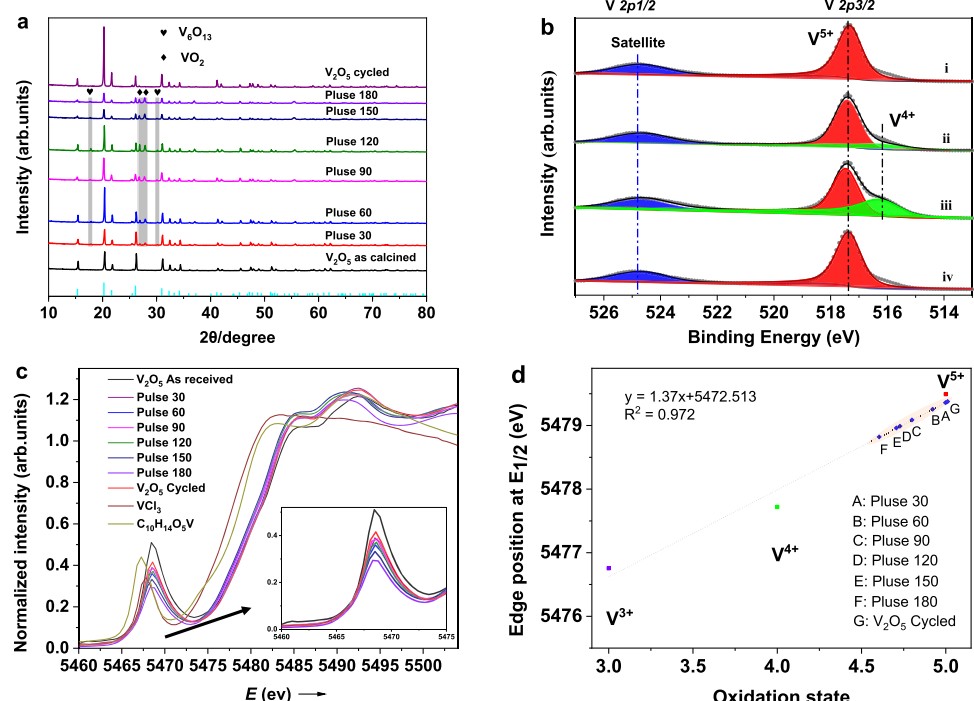

**Fig. 3 Structural analyses of $V_2O_5$ redox catalyst. a** X-ray powder diffraction patterns of $V_2O_5$ as a function of $NH_3$ pulses numbers. The following compounds were identified: $V_6O_{13}$ (black heart), $VO_2$ (black diamond), and $V_2O_5$ (vertical cyan line). **b** V 2p XPS spectra of $V_2O_5$ sample: (i) as calcined, (ii) after 30 pulses of $NH_3$ reduction, (iii) after 120 pulses of $NH_3$ reduction, and (iv) after CLAO redox cycling at 650 °C. **c** Normalized V K-edge XANES spectra of $V_2O_5$ catalysts with different $NH_3$ pulse numbers together with the standard compounds in different oxidation states. The insert shows the enlarged view of the pre-edge peak. **d** Plot of position of the absorption edge energies where the normalized absorption equals 0.5 (E1/2) vs the vanadium average oxidation state.

excellent recyclability through redox cycling. Based on the foregoing multipulse $NH_3$ experiments, it can be concluded that the ammonia oxidation to nitric oxide over $V_2O_5$ in the proposed CLAO process proceeds via a modified, temporally separated MvK scheme. $NH_3$ is first activated by the lattice oxygen of $V_2O_5$ redox catalyst in the absence of gas-phase oxygen, yielding NOx and a reduced VOx center. The reduction degree of $V_2O_5$ dictates the product distribution but does not alter the catalytic $NH_3$ oxidation activity. The reduced vanadium oxide surface sites are subsequently replenished by the spatially separated reoxidation step, closing the redox cycle.

**Redox catalyst characterizations.** To rationally develop a process for ammonia oxidation to nitric oxide catalyzed by redox metal oxide, in-depth understanding of the redox chemistry and reaction pathways is of prime importance. In this section, thorough physicochemical characterization combined with density functional theory calculations were employed to reveal the catalyst structure–activity relationship. Powder X-ray diffraction (XRD) was conducted to investigate the possible crystalline phase transition and structural stability of as-prepared, reduced, and reoxidized catalyst. As illustrated in Fig. 3a, the XRD pattern of the as calcined sample corresponds well to single-phase orthorhombic $V_2O_5$ structure (JCPDS 01-089-0612). Upon $NH_3$ pulse reduction, new reflections associated with $V_6O_{13}$ (17.76° and 30.15°, JCPDS 00-019-1399) and $VO_2$ (26.92° and 27.88°, JCPDS 03- 065-2358), respectively, are identified apart from $V_2O_5$. This indicates that $V_2O_5$ was partially reduced due to the stoichiometric reaction of the active lattice oxygen species with $NH_3$. In addition, peaks attributed to $V_6O_{13}$ and $VO_2$ progressively increased in intensity with increasing $NH_3$ pulse number as more lattice oxygen was extracted. After CLAO cycling, $V_6O_{13}$ and $VO_2$ peaks vanished, implying that the reduced phase was

completely reoxidized. Therefore, it was confirmed that $V_2O_5$ redox catalyst possesses excellent cycling stability over the redox cycles. For comparison, the XRD pattern of $V_2O_5$ in conventional co-feed mode is also shown, which indicates negligible structural change after the activity measurement (see Fig. S5 for more details).

In order to track the surface chemical state evolution of vanadium during the CLAO process, X-ray photoelectron spectroscopy (XPS) measurements were employed. The high-resolution XPS spectra in the V 2p region of $V_2O_5$ redox catalyst after different treatments are presented in Figs. 3b, S6a–d, and S7. For the initial catalyst (i), a strong peak centered at 517.4 eV (red) along with a broad satellite peak at a higher binding energy (blue) was observed, which agrees well with the signature of $V^{5+}$ in $V_2O_5$[54,55]. After 30 pulses of $NH_3$ reduction (ii), the $V^{5+}$ peak decreased and an additional peak emerged at a lower binding energy of 516.3 eV. Deconvolution of the V $2p_{3/2}$ region suggests that V was predominantly present in two different oxidation states, namely, $V^{5+}$ (79.7%) and $V^{4+}$ (20.3%) (see Table S3 for further details). For $V_2O_5$ sample subjected to 180 pulses of $NH_3$ reduction (iii), the $V^{4+}$ component further increased ($V^{4+}$, 38.4%) due to the enhanced reduction extent (Table S3). Note that no $V^{3+}$ or $V^{2+}$ was identified, evidencing that the vanadium existed only in the form of $V^{5+}$ and $V^{4+}$ after reduction with $NH_3$. For $V_2O_5$ sample after the CLAO redox cycling (iv), the intensity of $V^{5+}$ peak at 517.4 eV was recovered. Meanwhile, the contribution of $V^{4+}$ completely disappeared, pointing toward a dynamic recovery of vanadium following reoxidation. For comparison, no apparent reduction was observed for $V_2O_5$ after $NH_3$ oxidation reaction in the co-feed mode (Fig. S7). The above results highlight that vanadium could undergo reversible reduction and oxidation featuring a $V^{5+}/V^{4+}$ redox cycle in the CLAO process.

To obtain a deeper insight into the mechanistic details, XANES was used to determine the evolution of the vanadium oxidation state and coordination environment during the reaction. The position of the primary absorption and the intensity of the pre-edge feature serve as effective probes of the local electronic and geometric structure of vanadium[56–58]. Figure 3c depicts the normalized K-edge XANES spectra of the $V_2O_5$ samples as a function of $NH_3$ pulses number alongside the standard references with nominal valences of $+3$ ($VCl_3$), $+4$($C_{10}H_{14}O_5V$), and $+5$ ($V_2O_5$), respectively. With increasing $NH_3$ pulses number, a monotonic shift of the absorption edge to successively lower energies and an associated decrease in pre-edge peak intensity were observed (see Fig. S8 for further details). The observed shift towards lower energies is consistent with increased reduction of vanadium sites. Notably, the absorption edges for all $V_2O_5$ samples after $NH_3$ pulse reduction fell between $V_2O_5$ and $C_{10}H_{14}O_5V$, revealing an oxidation state intermediate between $+4$ and $+5$, which is well in accordance with the XPS results mentioned above. In addition, linear correlation between the oxidation state of reference compounds and the edge position of E1/2 (where the normalized absorbance equals 0.5) was employed to evaluate the oxidation degree of vanadium after $NH_3$ pulses reduction. The average vanadium oxidation state decreased to $+4.6$ after 180 pulses of $NH_3$ reduction (Fig. 3d). The reduction degree of $V_2O_5$ determined by XRD, XPS, and XANES was compared in Table S3. The mean oxidation state of vanadium and the percentage of $V^{4+}$ and $V^{5+}$ determined by XRD and XANES (obtained by comparison of the main edge energies and XANES linear combination fitting in Fig. S9) are in qualitative agreement with each other. The mean vanadium oxidation state is slightly higher than the average value of $+4.4$ of a global composition of $V_2O_{5-0.6}$ presented in Fig. S4. The small difference is most likely induced by post-reaction air exposure due to the ex-situ XRD and XNAES measurement. Meanwhile, XPS results give a higher $V^{4+}$ portion (a higher reduction extent) at different $NH_3$ pulse numbers, which is due to the surface sensitivity of the XPS measurement. Overall, the XRD, XPS, and XANES techniques indicate a monotonic increase in vanadium reduction extent with increasing $NH_3$ pulses number. EXAFS analysis of the data collected on the reduced samples are in line with the increased reduction of $V_2O_5$ (see Figs. S10 and 11 and Table S4 for more details). On the other hand, the decrease of the pre-edge peak intensity (inset in Fig. 3c) reflects an enhanced local symmetry around vanadium atoms[59,60]. Combined with the XRD results, the increase in local symmetry originates from the progressive evolution from strongly distorted square-pyramidal $VO_5$ groups in $V_2O_5$ to less distorted octahedral $VO_6$ groups in $V_6O_{13}$ and $VO_2$ (Fig. S12). Moreover, it has been well documented that the pronounced pre-edge peak is assignable to the hybridization of V 3d with O 2p of the nearest oxygen, which is roughly proportional to the number of vanadyl (V=O) group present[56,57,59]. The characteristic pre-peak of the V K-edge XANES spectrum of $V_2O_5$ is concurrently related to the breaking of the central symmetry and the presence of the short V=O bond (1.585 Å). Density functional theory calculations also assign the pre-edge feature to the V 3d−O 2p hybridization of the vanadyl bond (V=O) and the local geometry of the distorted $[VO_5]$ square pyramid[61]. Thus, the decreased pre-edge intensity evidences that with increasing $NH_3$ pulses number vanadyl bond diminished after reacting with $NH_3$. Upon CLAO cycling, the absorption edge shifts back to higher energies while the pre-edge intensity increases, showing that the reduction and re-oxidation processes are entirely reversible.

In situ Raman spectroscopy was further employed to determine the nature of the active centers on $V_2O_5$ redox catalyst, as it constitutes a highly sensitive tool for monitoring the rearrangements of V–O at an atomic level[62–64]. Fig. S13a presents the Raman spectrum of the fresh $V_2O_5$ sample. A set of intense Raman bands at 100, 144, 198, 284, 303, 406, 481, 525, 701, and 994 cm$^{-1}$ were identified, which correspond well to crystalline $V_2O_5$ nanoparticles[62,63]. The highest frequency mode at 994 cm$^{-1}$ has been attributed to the shortest vanadyl (V=O) stretch within $VO_5$ square pyramids. Whereas, the low-frequency modes at 100, 144 and 198 cm$^{-1}$ have been assigned to external modes corresponding to the displacements of $[VO_5–VO_5]$ units, which serve as a good baseline for reflecting the long-range order within $V_2O_5$. The next highest modes at 701 and 525 cm$^{-1}$ are ascribed to the stretching vibration of V–O$^{II}$–V bands, while the relatively weak mode at 481 cm$^{-1}$ is associated with the bending of V–O$^{III}$–V bridging units. Three additional modes at 284, 303, and 406 cm$^{-1}$ are assigned to bond rocking oscillations involving the terminal oxygen atoms. The in situ Raman spectra for $V_2O_5$ during $NH_3$ reduction and the following $O_2$ oxidation step are depicted Figs. 4 and S13b, c. As shown in Fig. 4a, both V=O and $[VO_5–VO_5]$ bands lost intensity with increasing reduction time, implying the removal of terminal oxygen and the continual loss of long-range order within $V_2O_5$. After re-oxidation, the V=O and $[VO_5–VO_5]$ bands were reinstated (Fig. 4b), indicating the reversible structural changes upon re-oxidation. Contrarily, no notable change in intensity was observed for V–O$^{II}$–V and V–O$^{III}$–V peaks during the reduction and oxidation steps (Fig. 4a, b). By tracking the evolution of V=O vibration (∼994 cm$^{-1}$) as a function of time, it was noticed that the V=O Raman band gradually diminished in intensity during the reduction step (Figs. 4c and S13b), which strongly indicate the destroying of V=O sites after reacting with $NH_3$. Interestingly, during the following re-oxidation step (Figs. 4b and S13c), V=O Raman band increased slowly within the first 10 min. Subsequently, the V=O signal rapidly approached to a maximum within the next 5 min, indicating that the V=O sites were rapidly regenerated after re-oxidizing with $O_2$. Hence, the above XANES and Raman spectroscopy results indicate that the reaction proceeds via a unique, temporally separated MvK mechanism associated with the breaking and regeneration of V=O bonds. While a classical MvK mechanism would involve concurrent removal and restoration of surface lattice oxygen species in separate steps, and often at spatially separated active centers[65,66]. Under the CLAO reaction scheme, the active V=O sites selectively activates and reacts with ammonia, while being removed from the surface. In a subsequent, temporally separated oxidation step, the active V=O sites were restored as confirmed by in-situ Raman spectra.

The crystallinity and surface morphology of $V_2O_5$ redox catalyst were investigated by HRTEM and SEM techniques. The high-resolution TEM image and the corresponding Fast Fourier transform (FFT) pattern are demonstrated in Fig. 5a. The as-calcined sample exhibited well-resolved lattice fringes of 3.4 and 5.7 Å (Fig. 5a), which coincide with the (110) and (200) facets of rhombohedral $V_2O_5$, respectively. Figure 5b–d shows representative SEM images of the as-calcined $V_2O_5$ sample, $V_2O_5$ after 30 $NH_3$ pluses reduction as well as $V_2O_5$ after the CLAO redox cycling. The as-calcined sample are platelet-like in shape with thicknesses ranging from 0.8–1.5 μm, lateral and longitudinal dimensions of 2–6 μm (Fig. 5b). In the case of $V_2O_5$ after 30 $NH_3$ pluses reduction (Fig. 5c) and $V_2O_5$ after the CLAO redox cycling (Fig. 5d), severe sintering and substantial morphological changes were not observed, confirming the robustness and stability of $V_2O_5$ redox catalyst under the demanding reducing and oxidizing cycling.

**DFT studies**. To gain a more fundamental understanding of the CLAO reaction mechanism, DFT calculations were carried out to elucidate the reaction landscape associated with the dissociation

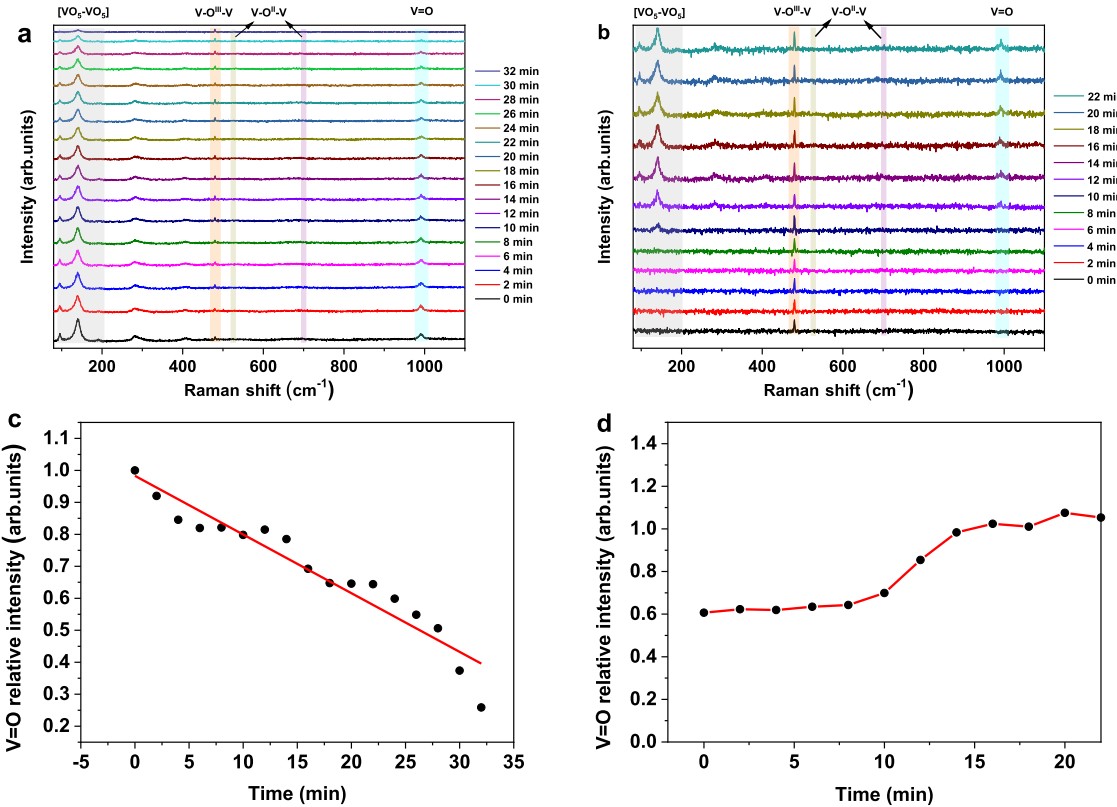

**Fig. 4 In-situ Raman spectroscopy.** In situ Raman spectra of $V_2O_5$ redox catalyst during the **a** reduction and **b** subsequent re-oxidation as a function of time. Reaction conditions: reduction, 5% $NH_3$/Ar (2.2 ml/min); oxidation, 30% $O_2$/Ar (2.2 ml/min); temperature, 650 °C. The corresponding V=O relatively intensity evolution during the **c** reduction and **d** re-oxidation step. V=O relatively intensity equals V=O reduced (oxidized)/V=O at full oxidation state.

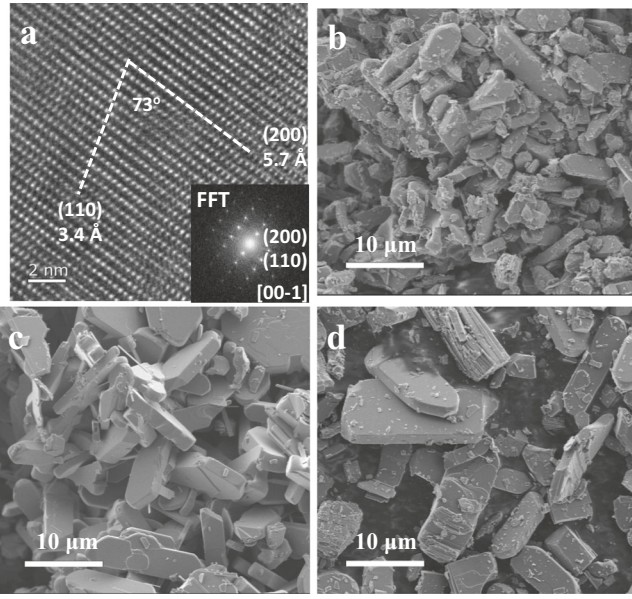

**Fig. 5 Morphological characteristics of $V_2O_5$.** **a** HRTEM image of the as-calcined $V_2O_5$ sample and the corresponding Fast Fourier transforms (FFTs) image (inset). Representative SEM images of $V_2O_5$ sample: **b** as-calcined, **c** after 30 pulses of $NH_3$ reduction at 650 °C and **d** after CLAO redox cycling at 650 °C.

of $NH_3$ and the formation of NO, $N_2$, $N_2O$, and $H_2O$. In particular, the catalytic roles of different oxygen sites are investigated in detail. The reaction is initiated by the adsorption of $NH_3$ on $V_2O_5$ (001) surface (Fig. S14), where the single $NH_3$ adsorption and two $NH_3$ co-adsorption were both studied. As shown in Fig. S15, the reaction pathway for single $NH_3$ adsorption is less favorable due to the very high reaction barrier (1.42 eV) of the rate-determining step (refer to Supplementary information for more details). Meanwhile, NO formation starting from the two $NH_3$ co-adsorbed configurations ($E_{ads} = 3.95$ eV) is also investigated, in which the surface of $V_2O_5$ (001) retains the original structure without serious reconstructions. As illustrated in Fig. 6a, the first hydrogen migrates to adjacent $O^{II}$, forming an $O^{II}H$ and $NH_2^*$, which is the rate-determining step endothermic by 0.21 eV with a medium barrier of 1.06 eV. The first hydrogen tends to continue migrating to the neighboring $O^I$ bonding with neighboring V, which is more energetically favorable with a much lower barrier of 0.47 eV in comparison with the direct dehydrogenation of the second hydrogen (1.80 eV, Fig. S16). This step liberates the $O^{II}$ site in the vicinity of $NH_2^*$, promoting the second hydrogen transfer with a relatively small barrier of 0.79 eV. Subsequently, the $NHO^*$ intermediate is formed via the combination of $NH^*$ and $O^I$ (activated by 0.15 eV), followed by $H_2O^I$ formation via the second hydrogen transfer from $O^{II}$ site to the $O^IH$ site with an extreme small barrier of 0.09 eV. This step again liberates the $O^{II}$ site to accommodate the last hydrogen, resulting in $NO^*$ formation with a negligible barrier of 0.03 eV. The resulting $NO^*$ can be released from the surface with 0.16 eV energy increase. The above DFT calculations clearly indicate that the oxidation of $NH_3$ to NO and $H_2O$ is favorably occurring on vanadyl oxygen $O^I$ site (V = O), while the successive stripping of

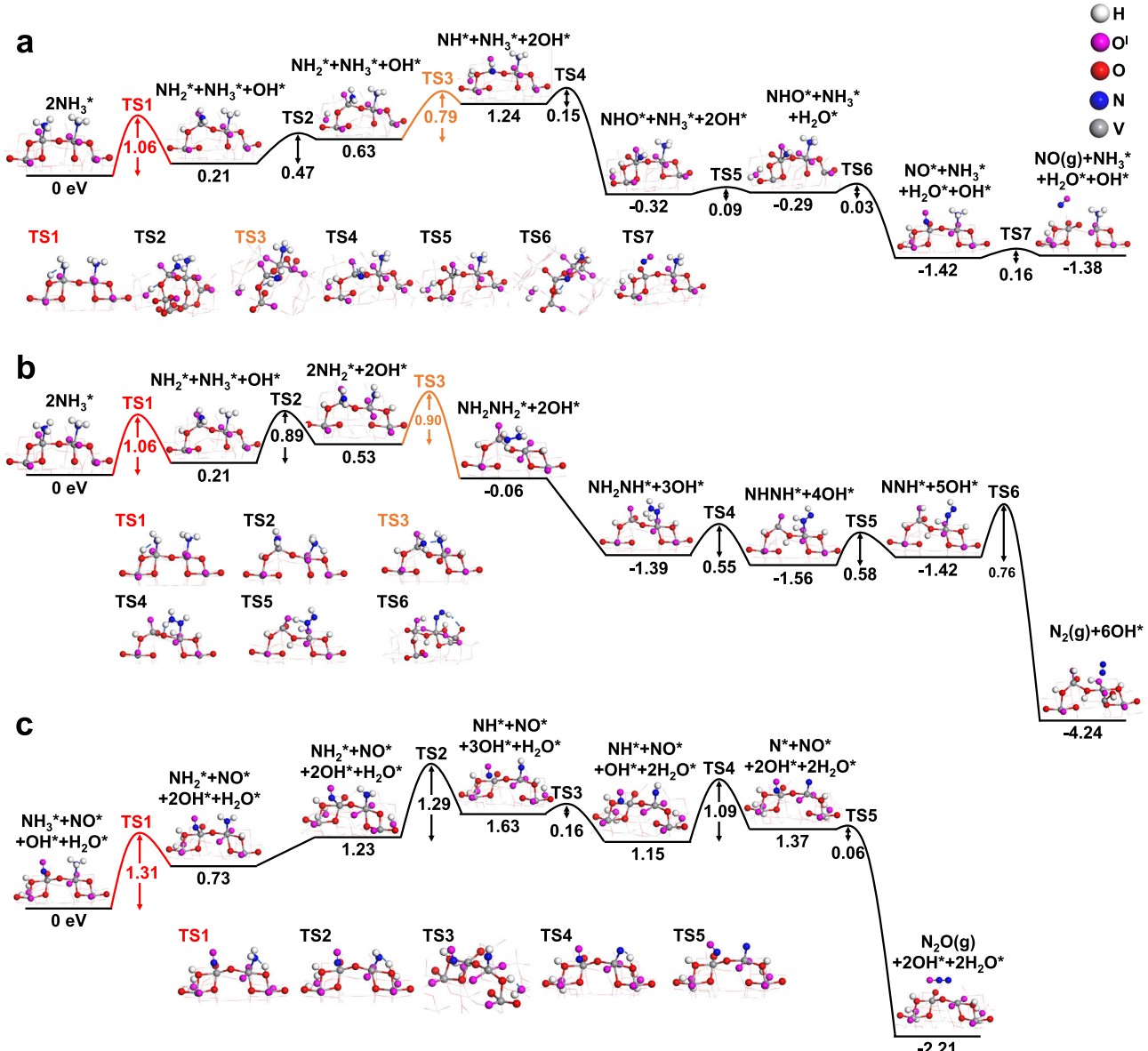

**Fig. 6 DFT calculations of CLAO paths on V₂O₅.** Computed energy potential profiles of each elementary step for the formation of **a** NO, **b** N₂ and **c** N₂O on V₂O₅ (001) surface. The kinetic barrier of each elementary step is given in eV.

hydrogen is catalyzed by both $O^I$ and $O^{II}$ sites. This result is in line with the in situ Raman spectroscopy, in which only V = O consumption was observed when reacting with $NH_3$.

For $N_2$ formation, two possible reaction pathways were considered according to whether atomic $N^*$ intermediates are produced. The energetically more favorable pathway is depicted in Fig. 6b. Initially, two $NH_3$ dehydrogenate into two $NH_2^*$ species with the activation barriers of 1.06 (rate-determining step) and 0.89 eV, respectively. Instead of continuing dehydrogenation, the two $NH_2^*$ tend to combine into the $NH_2NH_2^*$ intermediate due to the relatively low reaction barrier (0.90 eV). Upon $NH_2NH_2^*$ formation, the final 4 steps of hydrogen abstraction proceed, each with a relatively low activation barrier, resulting in $N_2$ formation. For each dehydrogenation step, all possible hydrogen accepter sites were examined. Our calculations reveal that both $O^I$ and $O^{II}$ could be hydrogen acceptors during the dehydrogenation steps due to the close energy barriers. For comparison, the atomic $N^*$ was also considered as the intermediates. As shown in Fig. S17, the entire reaction process

is exothermic (−5.10 eV), however, all dehydrogenation and N–N combination steps are highly endothermic with the highest activation barrier being 1.49 eV, making this pathway less feasible. It should be noted that since both NO and $N_2$ formation share the same rate-determining step (TS1 in Fig. 6a, b), their relative rates are largely determined by the second highest barriers as highlighted in orange, i.e. the second hydrogen transfer (0.79 eV) ($NH_2^* \rightarrow NH^*$) for NO formation and the two $NH_2^*$ combination ($2NH_2^* \rightarrow NH_2NH_2^*$) (0.90 eV) for $N_2$ formation. To take the temperature effects into account, the Gibbs free energy corrections over temperature for these two key steps were calculated as shown in Fig. S18. Results show that the free energy barriers of $NH_2^* \rightarrow NH^*$ are always much lower than those of $2NH_2^* \rightarrow NH_2NH_2^*$, which is coherent with experimental observations whereby NO prevails at high temperature. It should be noted that at this stage, the existing DFT calculations are not sufficient to justify the product selectivity considering the complexity of the reaction networks. This problem may be addressed in future studies by using more advanced kinetic

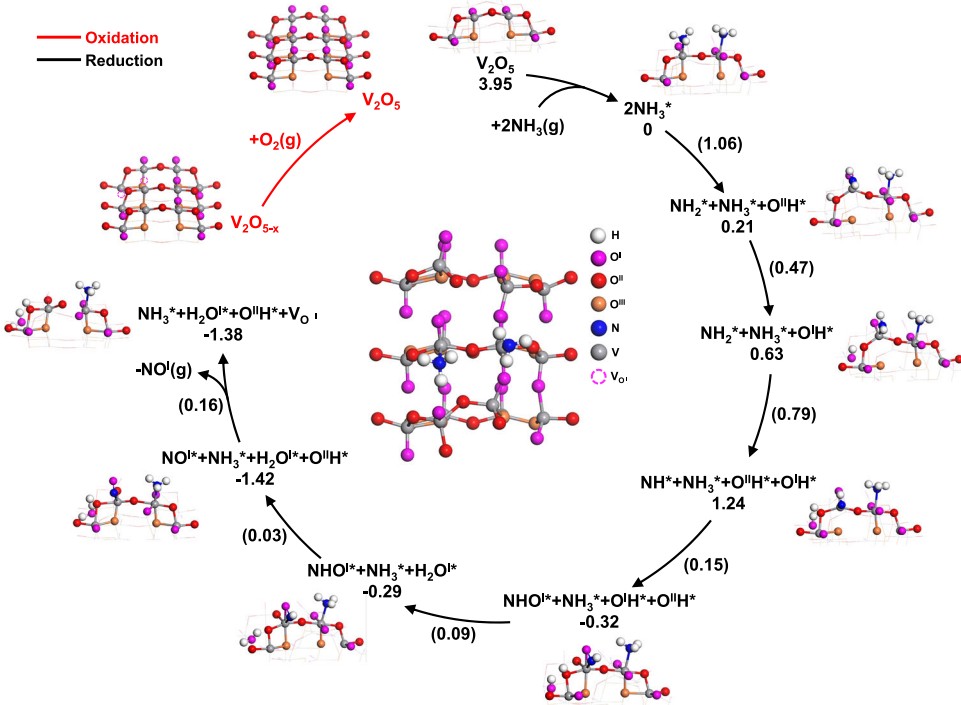

**Fig. 7 Proposed reaction mechanism.** Schematic of the proposed reaction mechanism for CLAO process over $V_2O_5$ redox catalyst. The elementary reaction barriers are given in eV.

analysis methods like microkinetic modeling. Additionally, the energy potential profiles as well as the possible intermediates for $N_2O$ formation is illustrated in Fig. 6c. Starting from the $NO^* + NH_3^*$ configuration, the first N–H bond cleavage is also the rate-determining step with a relatively large activation barrier of 1.31 eV higher than that of NO and $N_2$ formation (1.06 eV). Thus, $N_2O$ is largely suppressed during the reaction, which is in well accordance with the experiments. Notably, the oxygen atoms in the resulting oxidation products also arise from $O^I$. Based on combined experimental and theoretical studies, it is therefore concluded that $O^I$ sites are the major oxidants leading to the formation of oxidation products, meanwhile both $O^I$ and $O^{II}$ sites participate in the hydrogen transfer processes. The inhibition of $N_2O$ formation makes $V_2O_5$ a very selective catalyst for CLAO processes.

**Reaction mechanism**. Combining catalytic measurements, spectroscopic studies as well as theoretical calculations presented above allows for a molecular-level understanding of the potential reaction pathways for the proposed CLAO process. As illustrated in Fig. 7, this process follows a unique, temporally separated MvK redox mechanism whereby lattice oxygen species participated in the reaction. In the first reduction step, $NH_3$ was activated by the lattice oxygen originated from $V_2O_5$ catalyst, leading to NO formation and a reduced $VO_x$ center ($VO_2$ and $V_6O_{13}$). In the subsequent re-oxidation step, the reduced vanadium oxide site was replenished by gas-phase $O_2$, closing the catalytic cycle. XPS and XANES studies demonstrated that the CLAO process involved the reversible reduction of vanadium cations with a $V^{5+}/V^{4+}$ redox couple easily accessible. Combined operando Raman and XNAES spectroscopy provided concrete evidence that among the three lattice oxygen sites on $V_2O_5$: the vanadyl (V = O), the bridge oxygen $O^{II}$, and the triply coordinated oxygen $O^{III}$, the V = O site was the catalytic active site for the selective oxidation of $NH_3$ to NO. The initial high NO formation can be attributed to the presence of highly reactive and selective

V = O species, which decreased with increasing $NH_3$ pulse numbers due to their removal for $NH_3$ oxidation to NO. The V = O species were regenerated following oxidation, restoring the high NO selectivity. This temporally separated MvK scheme avoids direct competition of reduction and reoxidation on the same V = O active sites that would otherwise be inevitable in a co-feed scheme, leading to remarkable enhancement in catalyst performance. Furthermore, in the proposed CLAO process, the activation of oxygen is spatially and temporally separated from the catalytic $NH_3$ oxidation to effectively suppress the formation of nonselective surface molecular oxygen species and thus the undesired nitrogen byproduct[15,16]. Additionally, the chemical looping strategy allows for the precise control of the residence time of reactants, the reduction extent of the redox metal oxide and ratios of the reaction intermediates ($O/NH_x$) on the catalyst surface. Therefore, a high $NH_3$ conversion can be achieved, with can potentially suppress the $N_2$ formation via the catalytic reduction of NO with $NH_3$. Consequently, a high selectivity for NO can be anticipated. Kinetic studies indicated that the apparent activation energy for the lattice oxygen removal is as low as 16.0 kJ mol$^{-1}$, which is consistent with the superior low-temperature selective $NH_3$ oxidation performance. In agreement with the experimental observations, DFT calculations indicate that only $O^I$ and $O^{II}$ are terminated and can participate in the reaction. The CLAO process is initiated by the co-adsorption of $2NH_3$ with $E_{ads} = 3.95$ eV, followed by the first dehydrogenation step, activated by 1.06 eV and endothermic by 0.21 eV, which is catalyzed by adjacent two-coordinate oxygen site ($O^{II}$) and represents the rate-determining step. The hydrogen continues to transfer from the $O^{II}$ to the neighboring $O^I$ to form $O^IH^*$ (activated by 0.47 eV). The second dehydrogenation step occurs afterwards with a relatively small barrier of 0.79 eV, which is also catalyzed by the nearby $O^{II}$ site, forming $NH^*$ and $O^{II}H$. The resulting $NH^*$ can easily bonds with adjacent $O^I$ site to form the $NHO^{I*}$ intermediate due to the very small barrier of 0.15 eV. The two hydroxyl groups ($O^{II}H$ and $O^IH$) can also easily interact to

form $H_2O^I$ molecule due to the extremely low barrier of 0.09 eV. The liberated $O^{II}$ site in the vicinity of $NHO^I*$ induces the final dehydrogenation step, leading to the facile $NO*$ formation with a negligible barrier of 0.03 eV. Subsequently, $NO*$ can be easily released from the catalyst surface with a very low activation barrier (0.16 eV). The remaining $NH_3*$ on the surface can be co-adsorbed with another $NH_3$ molecule to generate NO mediated by $O^I$ and $O^{II}$ species. Upon reoxidation, the oxygen-deprived $V_2O_5$ is regenerated, completing the CLAO redox cycle.

## Discussion

In this contribution, catalytic ammonia oxidation to nitric oxide via a chemical looping approach was proposed and demonstrated in absence of noble metal catalyst. Among many metal oxide redox catalysts investigated, $V_2O_5$ exhibited superior $NH_3$ conversion at a substantially lower temperature with an outstanding NO selectivity (99.8%), outperforming the expensive, commercially applied Pt−Rh gauzes at a temperature up to 300 °C lower. In addition, the selectivity to undesirable $N_2O$ was largely suppressed, which was distinctly different from $NH_3$ oxidation in the conventional co-feed mode. Repeatable $NH_3$ conversion and product selectivity were achieved during consecutive redox cycles over 1300 min. The superior performance was rationalized down to the atomic level by both operando and ex-situ characterization techniques complemented by DFT calculations. Results show that the proposed CLAO process operates via a modified, temporally separated MvK redox mechanism featuring a reversible $V^{5+}/V^{4+}$ redox cycle. The vanadyl (V=O) centers act as active sites for $NH_3$ oxidation, leading to the formation of the oxidation products. Meanwhile both V=O and doubly coordinated oxygen ($O^{II}$) participate in the hydrogen transfer process. The superior activity and selectivity can be attributed to the low activation energies for the successive hydrogen abstraction and facile NO formation facilitated by V=O. The fundamental understanding and the proof-of-concept for CLAO can open exciting perspectives for nitric acid manufacturing in a cleaner, more sustainable and cost-efficient manner.

## Methods

**Catalyst synthesis**. $V_2O_5$ was purchased from Energy chemical (+99.5%). The as-received powder was calcined under stagnant air at 650 °C for 4 h with a ramp rate of 2 °C min$^{-1}$. Prior to use, the calcined sample was ground into a fine powder using a mortar and pestle. Additionally, commercially available $CeO_2$ (99.5%, Aladdin Industrial Corp.), $SnO_2$ (99.5%, Aladdin Industrial Corp.), $Fe_3O_4$ (99%, Aladdin Industrial Corp.), $NiFe_2O_4$ (99.5%, Aladdin Industrial Corp.), $CoFe_2O_4$ (99.5%, Aladdin Industrial Corp.), and $ZnFe_2O_4$ (99.5%, Aladdin Industrial Corp.) were also investigated for comparison.

**Catalytic activity measurements**. The ammonia oxidation process was conducted at atmospheric pressure in a fixed-bed quartz U-tube reactor (i.d. = 9.75 mm), which was enclosed by an electric furnace (AutoChem II 2920 Micromeritics). Both chemical looping and co-feed modes were carried out for reactivity test in the temperature range of 300–650 °C. In both modes, 200 mg of the catalyst was packed between two layers of quartz wool within the isothermal zone of the reactor. The temperature was controlled using a K-type thermocouple in direct contact with the catalyst bed. The inlet gas flow rates were regulated by calibrated mass flow controllers.

(i) In typical chemical looping mode, pulses of 5% $NH_3$ (765 μL, balance Ar) were injected into the reactor using Ar (30 mL min$^{-1}$) as the carrier gas. Subsequently, a regeneration step was conducted using 30% $O_2$ (30 mL min$^{-1}$, balance Ar). To study the evolution of catalytic performances, 30 consecutive ammonia pulses were injected to reduce the redox catalysts, followed by an oxidation step with simulated air (30% $O_2$/Ar) at a total flow rate of 30 mL min$^{-1}$ for 30 min. In between the reduction and oxidation step, a purge step with pure Ar (30 mL min$^{-1}$) was introduced for 15 min to assure well-defined conditions.

(ii) In typical co-feed mode, steady-state ammonia oxidation was carried out using feeds containing 5% $NH_3$/Ar (30 mL min$^{-1}$) and $O_2$ (12.86 mL min$^{-1}$) with the effective $O_2$ percentage of 30%.

Outlet gas concentrations were monitored online after condensation of moisture by a quadrupole spectrometer (Pfeiffer Omnistar QMS 200), using Ar as internal standard.

And the corresponding molar flow rates ($n_i$) were calculated. The spectra signals were calibrated before each experiment with calibration gas of known compositions (5.0% $NH_3$/Ar, 6006 ppm NO/Ar, 3002 ppm $N_2O$/Ar, 5007 ppm $N_2$/Ar, 4995 ppm $H_2$/Ar). The atomic mass units (amu) related to $NH_3$ (17), NO (30), $N_2O$ (44, 30, 28), $N_2$ (28), $H_2$ (2), and Ar (40) were analyzed. The fraction originated from $N_2O$ was deducted from the signal at 30 and 28 amu before NO and $N_2$ quantification. The accuracy of the measurement was assured by performing nitrogen balances and the deviations were closed within 10% for all reported experiments.

$$\text{Nitrogen balance}: \int_0^t n_{NH_{3,in}} dt = \int_0^t n_{NH_{3,out}} dt + \int_0^t n_{NO,out} dt \\ + 2\int_0^t n_{N_2O,out} dt + 2\int_0^t n_{N_2,out} dt \quad (4)$$

The $NH_3$ conversion and NOx selectivity were calculated using the following formulas:

$$X_{NH_3} = \frac{\int_0^t n_{NH_{3,in}} dt - \int_0^t n_{NH_{3,out}} dt}{\int_0^t n_{NH_{3,in}} dt} \quad (5)$$

$$S_{NO} = \frac{\int_0^t n_{NO,out} dt}{\int_0^t n_{NO,out} dt + 2\int_0^t n_{N_2O,out} dt + 2\int_0^t n_{N_2,out} dt} \quad (6)$$

$$S_{N_2O} = \frac{2\int_0^t n_{N_2O,out} dt}{\int_0^t n_{NO,out} dt + 2\int_0^t n_{N_2O,out} dt + 2\int_0^t n_{N_2,out} dt} \quad (7)$$

$$S_{N_2} = \frac{2\int_0^t n_{N_2,out} dt}{\int_0^t n_{NO,out} dt + 2\int_0^t n_{N_2O,out} dt + 2\int_0^t n_{N_2,out} dt} \quad (8)$$

where $n_{NH_{3,in}}$ is the molar flow rate of $NH_3$ introduced into the reactor, $n_{NO,out}$, $n_{N_2O,out}$, and $n_{N_2,out}$ is the molar flow rate of $NH_3$, NO, $N_2O$, and $N_2$ from the gas outlet.

**Catalyst characterization**. Powder X-ray diffraction (XRD) measurements were carried out to analyze the crystalline phases of the catalysts during the redox reaction. XRD patterns were recorded on a PANalytical diffractometer with Cu Kα radiation ($\lambda = 1.5418$ Å) at 40 kV and 40 mA. Recording was performed in the 2θ range of 10−80° with a step size of 0.02°. HighScore plus software was employed for phase identification with reference to the International Center for Diffraction Data (ICDD) database. The peak position gave information about the lattice spacing based on the Bragg law of diffraction.

X-ray photoelectron spectroscopy measurements were adopted for near-surface chemical states analysis using Thermo Fisher Scientific ESCALAB 250 equipped with a monochromatic Al Kα (hυ = 1486.6 eV) in ultrahigh vacuum conditions (base pressure $P \sim 3 \times 10^{-8}$ Pa). Pressed samples were attached to conductive carbon adhesive tapes and the XPS spectra were recorded at a constant pass energy 20 eV and a resolution energy of 0.1 eV. Peak fitting was performed with XPSPEAK program Shirley background subtraction together with a combination of Gaussian–Lorentzian functions. The C 1 s peak at 284.6 eV was used for calibration of all binding energies.

X-ray absorption near-edge spectroscopy (XANES) measurements around the V K-edge (~5465 eV) were performed at beam line 4W1B of Beijing Synchrotron Radiation Facility (BSRF). The storage ring was operated at 2.2 GeV with a typical electron current of 50 mA. The redox catalysts were homogeneously smeared on Scotch adhesive tape and folded to reach the optimum absorption thickness. Each sample was packed in a polyethylene bag under an argon atmosphere. Spectra were collected at ambient temperature in transmission mode with a double crystal Si(111) monochromator and vanadium foil as a reference material for edge-shift calibration. Reagent grade $C_{10}H_{14}O_5V$ and $VCl_3$ were used as $V^{4+}$ and $V^{3+}$ standards, respectively. Data were normalized in a standard curve-fitting procedure using the Athena software program.

In situ Raman spectroscopy studies were performed using a high-resolution LabRam HR800 confocal Raman microprobe (Horiba-Jobin Yvon, France) equipped with three laser excitations (532, 442, and 325 nm). The experiments were performed in an in situ high-temperature Linkam CCR1000 cell attached to a gas manifold. The spectra were collected using the 532-nm laser with five accumulations at 10 s per scan and a 200-μm laser hole size. Prior to the measurements, the sample was heated to 650 °C (10 °C min$^{-1}$ ramp) under 2.2 ml/min Ar. The redox behavior and molecular structures of the catalyst was fingerprinted via the following two-step procedure. (i) Reduction under 5% $NH_3$/Ar (2.2 ml/min). Raman spectra were collected during the reduction process until a constant intensity (or complete disappearance of the peak) was achieved. (ii) Oxidation under 30% $O_2$/Ar (2.2 ml/min). Raman spectra were taken during the oxidation process until a constant intensity was achieved.

The surface morphology of the fresh and cycled catalyst was observed by scanning electron microscopy (SEM) on a JSM-7800F instrument using an accelerating voltage of 1 kV. Samples were dusted on an adhesive conductive carbon belt attached to a copper disk. High-resolution transmission electron microscopy (HRTEM) characterization was

implemented on a JEOL JEM-2200F microscope operated at 200 kV with a linear resolution of 0.10 nm. Prior to measurement, catalyst sample was ultrasonicated in ethanol and deposited on copper grids coated with a holey carbon film, followed by drying at room temperature.

**Computational details**. DFT calculations were performed using the Vienna ab initio Simulations Package (VASP)[67] with the frozen-core all-electron projector augmented wave (PAW) model[68] and Perdew-Burke-Ernzerhof (PBE) functional[69]. A kinetic energy cutoff of 400 eV was used for the plane-wave expansion of the electronic wave functions, and the convergence criteria of force and energy were set to 0.01 eV/Å and $10^{-5}$ eV respectively. A Gaussian smearing of 0.1 eV was applied for optimizations and a k-point grid with a $9 \times 7 \times 3$ Gamma centered mesh was chosen for single $V_2O_5$ bulk structure to sample the first Brillouin zone, and the same k-mesh spacing was kept for supercells. The GGA + U method was chosen to describe partially filled 3d orbitals in V with $U = 4$ eV and $J = 0.68$ eV for Hund's exchange interaction[70]. The SSW HOWTOs program was applied to search for reasonable testing structures which were used for later transition state (TS) optimization with the climbing image nudged elastic band (CI-NEB) method[71]. For a reaction that has multiple final states, the most stable one was selected. All the computed energies are the total electronic energies unless otherwise stated. The Gibbs free energy corrections were calculated using VASP-KIT, which includes the zero-point energy (ZPE), enthalpy, and entropy contributions. For surface adsorbates, free energies were approximated using the harmonic approximation that treats all degrees of freedom as vibrational modes[72]. The $V_2O_5$ (001) surface (Fig. S14a) was chosen for all the studied reactions as it is the most thermodynamically stable plane exposed in crystalline $V_2O_5$[73].

## Data availability

The data that support the plots within this paper and other findings of this study are available in the Supplementary Information and are also available from the corresponding author upon reasonable request. Source data are also provided with this paper.

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

## Acknowledgements
This research was supported by National Natural Science Foundation of China (22022814, 21878283, 22178337, 21776271), the Youth Innovation Promotion Association CAS (2017223), the Strategic Priority Research Program of the Chinese Academy of Sciences (XDB17020100), the National Key projects for Fundamental Research and Development of China (2016YFA0202801), and the US National Science Foundation (CBET-1923468).

## Author contributions
C.R. conducted the experimental studies and wrote the paper. Xij.W. finished the DFT calculations and wrote the paper. L.L. contributed to the In situ Raman experimental design. C.W performed some XPS measurements and reactivity tests. X.L and L.Z contributed to the discussion of the XANES results. J.L participate in the discussion. Xia.W. and F.L conceived this project revised the manuscript, triggering helpful discussions. All authors discussed the results and commented on the manuscript.

## Competing interests
The authors declare no competing interests.
