## [Peer Review File · Nature Communications]

REVIEWER COMMENTS

Reviewer #1 (Remarks to the Author):

This paper reports selective NH₃ oxidation to NO via chemical looping using a commercial V₂O₅. While the work herein has evidently been carried out with care on the reaction, catalyst characterization, and DFT calculation, the novelty and performance of the presented material system seem to be not enough. The high NH₃ conversion and/or NO selectivity at the very early stage of NH₃ injections would be understandable but the effect does not last so long. Overall, the reviewer cannot find any significant advances that are suitable for the publication in the NatComm. Detailed comments are as follows.

Page 1

The authors point that Pt loss through volatilization at high-temperature in the conventional Ostwald process. The reviewer feels that the same stability problem will occur when V₂O₅ is regenerated at 650 degree C under air, because it is exothermic and just below the melting point of V₂O₅, where the vapor pressure is high.

Pages 2-3, Fig. 2, and Table S1

The pulse mode reaction is not appropriate for the evaluation of NH₃ oxidation. The rate is too slow to compare with the very fast NH₃+O₂ reaction in the conventional co-feeding mode. Switching two flows (NH₃ and O₂) should be a better approach for the evaluation of the performance. Not only NH₃ conversion and NO selectivity but the reaction rate, W/F, or GHSV should be compared in Table S1. The practical value of this chemical looping system is dependent on the NO production rate.

Page 4, Fig.S3

As shown in Fig. S3, the selective NH₃ conversion to NO occurs only at the beginning of NH₃ injections. However, the NO selectivity decreases with increasing pulse number. This is what happens in a flow mode. Based on the results, the amount of V₂O₅ required for selective NH₃ oxidation to NO will be very huge, e.g., more than 20 mol-V₂O₅ for 1 mol-NO, and unrealistic.

Reviewer #2 (Remarks to the Author):

The paper by Ruan et al. describes selective production of NO via a chemical looping experiment (NH₃/Ar and O₂/Ar) on V₂O₅. Overall, the most impressive part of the manuscript are the results from the chemical looping experiment. The side-by-side comparison of the continual flow experiment and CL is really nice and shows the value of this approach. It is difficult to assess the viability of this as a replacement technology, however, it is still extremely valuable that these results have been established. They will be of value of the field and are able to shape future thinking on how catalytic processes can be performed and optimised. Elsewhere, the manuscript contains physical characterisation (with in situ Raman) and DFT calculations of the reaction pathways. These seem a useful addition, however, it was unclear if they really enhanced the paper significantly beyond the core result in the catalytic testing. I was hoping to find more on the origin of the improved selectivity through CL - perhaps I am too greedy here - but I think it would help to expand on this point to be accepted in Nature Communications. To help improve the manuscript further, a point-by-point list of comments are detailed below:

- 1) In Fig. 1 the schematic says O₂ depleted air. However, the O₂ source is 30% O₂ in Ar.
- 2) Can the authors comment on the productivity per unit time as well as selectivity - this is helpful to understand the comparison between flow and CL.
- 3) Figure 2a and 2b nicely show the catalytic testing data. One of the noteworthy observations is that for the flow experiments, the N₂ selectivity decreases, before rising for the final temperature. In the discussion on the reaction pathways the only reaction route considered is NH₃ oxidation. However, there is a recent study that demonstrates that the NO_x produced through NH₃ oxidation can also create the conditions for a parallel SCR mechanism (<https://doi.org/10.1021/acscatal.0c05356>). Considering the increase of N₂ at high temperature is this something the authors have considered?
- 4) Following on from point 3, clearly there are structural changes to the catalyst and compositional changes of the reactant environment. Are the authors able to comment on the impact of these with respect to the catalytic properties?
- 5) At 300C there is 100% selectivity to N₂ for the CL experiment. What is the fate of the H₂. It is not seen in the MS. If it generates surface hydroxyls was this identified through XPS/Raman. Clearly, the focus of this paper is NH₃ oxidation, however, perhaps this methodology has translational impact to NH₃ decomposition and this could be an important point.
- 6) In figure 2e, why are only 9 injections shown for each repeat and not the full 30?
- 7) The physical characterisation is adequate and appear self-consistent, however in this work the XRD seems the most useful as it is able to identify the phases present. With the characterisation that is present can the authors present the degree of reduction through each technique (XRD, XPS, XANES) and comment on this comparison. For the XANES a LCF of reference standard would help.
- 8) If XAS was measured can the EXAFS also be presented and discussed.

9) For the XANES discussion the authors comment that the pre-edge feature can change in intensity based on the symmetry of the V centres and the number of V=O units. It seems the group use the same change in the data to potentially support different ideas. There authors need to make clear what they think the major influence is here.

10) For oxidation state changes in XANES analysis the edge position is not always reliable. Considering the size of the particles I think a XANES LCF with reference standards found in XRD would be more useful.

11) For figure 4 it would be better not to include the lines of fits (especially where the points are joined). Also the data quality is very noisy and should be commented on.

Reviewer #3 (Remarks to the Author):

The manuscript titled "Selective catalytic oxidation of ammonia to nitric oxide via chemical looping" by Wang and coworkers is focused on the use of earth abundant V₂O₅ catalyst for the selective conversion of NH₃ to NO using a chemical looping approach, as an alternative to noble metal-based catalysts. Overall, the work has been well performed. The results are interesting and broadly applicable. I think this will be of interest to Nature Communications' readers. This manuscript may be published at the discretion of the editor after the following technical and general comments have been addressed.

1. For comparison of continuous and CLAO operations have been made at different chemical potential of O₂. The reviewer suggests that in continuous operation, the effective O₂ percentage of 30% in the combined feed is perhaps a better comparison of selectivity. In this study, the O₂ is further diluted by 5 times giving an effective percentage of 6% I believe. It may not change the overall conclusion, but this will be a good experiment to show the readers. A more systematic investigation of O₂ partial pressure and its influence on selectivity for continuous operation may be helpful.

2. The calculation of rates of oxygen removal are valid at low rates of oxygen removal. The reviewer is curious at what level of conversion of perhaps V₂O₅ were these oxygen removal rates estimated from. In other words, were these initial rates? It appears that these levels of V₂O₅ conversions are low as seen in Figure 2d. If it is the case, I would recommend the authors mention it in the context that these rates reflect the initial oxygen removal rates from V₂O₅.

3. The XPS of the catalyst post reaction in continuous mode would be helpful to include. From the data presented, it appears that the authors find that over reduction is the cause of the change in product distribution. Can this be arrested with using a higher concentration of O₂ in continuous mode (Please see comment 1)?

4. The reviewer suggests that Figure 4 a and b be shown in alternative 2 d format. In the current 3 d format, it is difficult to read and understand the figure.

5. The authors are recommended to clearly mention whether they are using electronic energies and not enthalpies or free energies for the DFT calculations.

6. The interpretation of the DFT data on the temperature effects on product selectivity is weak in the opinion of the reviewer. Overall, this section needs some realignment. The authors mention, "At low temperature, both reactions can occur, however, the forward reaction of NO formation is slower than its reverse reaction due to its endothermic nature, making the overall NO formation rate lower than that of N₂ formation. Whereas, at high temperature, NO formation is more favourable due to its overall lower reaction barriers from the thermodynamics standpoint. The calculation results are coherent with experimental observations."

In the opinion of the reviewer, this argument is not correct. The reaction barriers do not determine thermodynamics. The only consistent feature here is the lower barrier towards NO formation than N₂ that explains the high temperature observation. If the authors want to use the calculations to justify the selectivity, I recommend using a microkinetic model that will help account for surface coverages and the relative product formation.

7. If the redox cycles are indeed useful for selective conversion of NH₃ to NO on V₂O₅, can the authors comment or have they tried to use other single metal oxides such as WO₃ or MoO₃ for the same reaction. If yes, can they include the data to generalize their conclusions. If not, can the authors perform these studies for comparison?

GENERAL COMMENTS TO THE REFEREES AND EDITOR:

We want to extend our appreciation for all the reviewers for their insightful guidance.

We carefully considered the constructive comments offered by the three reviewers and tried our best to revise the paper based on these comments and recommendations. We hope that these revisions improve the paper such that you and the reviewers now deem it worthy of publication. A point-by-point response to each reviews' concerns and comments are provided below.

RESPONSES TO REFEREE #1's COMMENTS:

Remarks to the Author

This paper reports selective NH_3 oxidation to NO via chemical looping using a commercial V_2O_5 . While the work herein has evidently been carried out with care on the reaction, catalyst characterization, and DFT calculation, the novelty and performance of the presented material system seem to be not enough. The high NH_3 conversion and/or NO selectivity at the very early stage of NH_3 injections would be understandable but the effect does not last so long. Overall, the reviewer cannot find any significant advances that are suitable for the publication in the NatComm. Detailed comments are as follows.

Response: We want to extend our appreciation for taking the time and effort

necessary to provide your helpful suggestions. In the following sections, you will find our responses to each of your point and suggestion.

The development of cost-effective and environmentally sound ammonia oxidation technologies is a grand challenge for modern chemical engineers. This work highlights the unique capability of a novel chemical looping ammonia oxidation (CLAO) system to replace the costly noble metal catalysts and to reduce greenhouse gas emission, which no current technology is able to achieve in a commercially relevant way. The novel chemical looping system we proposed and demonstrated for the first time, utilizes the nonprecious V_2O_5 redox catalyst (0.02%- 0.04% of price of Pt per lb; 0.001%- 0.002% of price of Rh per lb) for high purity NO generation at a record low temperature of 650 °C with negligible N_2O production. To comprehend the novelty of this chemical looping based ammonia oxidation technology, it is first paramount to identify fundamental limitations in existing commercial scale technologies.

Nitric acid production involving the catalytic oxidation of ammonia by air (oxygen) is one of the large-scale processes in chemical industry. Today the worldwide capacity of the manufacture of nitric acid, an important inorganic intermediate for the production of fertilizers and other chemical products such as explosives, is over 100 million tons per year.¹⁻³ Platinum and platinum-rhodium (with 5-10% Rh content) alloy gauzes are usually employed as the catalysts for this process. However, the platinum loss (0.05-0.3 g per ton of produced nitric acid) caused by volatile PtO_2 generated under high-temperature conditions in an oxygen atmosphere

results in a significant catalytic performance decay and cost increase.^{1, 4-9} The cost of lost Pt is the second largest expense of the operation,¹ which worth approximately 1.4 billion dollars annually and exceeded only by the cost of ammonia feedstock. Moreover, nitric acid manufacture is the largest single source of N₂O (125 × 10⁶ t CO₂-equiv per annum), a greenhouse gas and stratospheric ozone depletor with a very long life time of approximately 150 years.¹⁰ The significant catalyst consumption and N₂O emissions associated with this important industrial sector offer excellent opportunities for reducing catalyst cost, process intensification, emission reduction, as well as profitability enhancement.

Compared to conventional catalytic ammonia oxidation processes, the CLAO strategy is capable of:

- (i) Potentially replacing the scarcely limited platinum group metal (PGM) alloy gauzes with earth abundant metal oxide redox catalysts as a solid oxidant;
- (ii) Providing inherently separated product streams to eliminate or alleviate the needs for separation, thus intensifying the production of high-purity NO from NH₃ as large quantities of inert N₂ (about 3/4 of the total flow in a current plant) are excluded;
- (iii) Enhancing process safety by avoiding the direct mixing between a gaseous oxidant (Air) and reductant (NH₃);
- (iv) Redistributing the overall heat of reaction into two or more steps to simplify heat integration;
- (v) Potentially achieving higher product selectivity by using lattice oxygen as opposed to gaseous oxidants, reducing environmental pollution.

We believe the fundamentally new and commercially promising CLAO scheme holds great promise to address the aforementioned challenges and opportunities.

Starting with a fully oxidized catalyst surface, NH_3 conversion and NO formation are initially very high, with NH_3 conversion amounts to 97% and NO selectivity approaching 100% in the first 10 pulses. The high NH_3 conversion and NO selectivity at the early stage of NH_3 injection can be associated to the presence of highly reactive and selective lattice oxygen species ($\text{V}=\text{O}$). Due to the dynamic nature of redox catalysts undergoing chemical looping reactions, the extent of redox catalyst conversion (the stoichiometric ratios for reducible oxygen in the active compound to NH_3) in the reduction step dictates the equilibrium NH_3 conversion and the molar distribution of nitrogen species at the equilibrium state. This sensitivity necessitates precise control of the relative ratios of reducible oxygen to NH_3 ($\text{NO} : \text{N}_2\text{O} : \text{H}_2\text{O} : \text{NH}_3$). The unique periodic feature of the chemical looping strategy allows for the effectively controlling the residence time of the gas-solid reaction, and hence the conversion of the redox catalysts and NH_3 , and the composition of the nitrogen oxides product. In our study, by the precise control of the NH_3 pulse number, the molar ratios for reducible oxygen in the V_2O_5 to NH_3 can be finely tuned. NH_3 conversion above 97%, NO selectivity above 90% can be achieved with insignificant N_2O selectivity even after 30 pulses of NH_3 with a considerable amount of lattice oxygen utilized (2.4% of the total available oxygen). The novel CLAO strategy coupled with V_2O_5 redox catalyst can selectively oxidize NH_3 to NO, minimizing N_2O formation and thus eliminating the need for the additional N_2O removal unit. As a result, the CLAO

process facilitated by metal oxide redox catalyst has significant merits in efficiency improvement, emission reduction and potential cost saving.

In comparison, for the conventional ammonia oxidation via the Ostwald process, the selective oxidation of NH_3 to NO is considerably favored at a high oxygen coverage, which can be achieved by using an excess of oxygen over ammonia. In practice, such a situation can be accomplished with a larger excess of oxygen in the feed. However, this operation would dramatically reduce the lifetime of a gauze pack, since an increase in partial O_2 pressure accelerates platinum loss in the form of volatile PtO_2 .⁸

In the current study, typical single (V_2O_5 , CeO_2 , Fe_3O_4 , SnO_2) and binary redox metal oxide catalysts (ZnFe_2O_4 , CoFe_2O_4 , NiFe_2O_4) were investigated for CLAO. The desirable thermodynamic property of V_2O_5 with proper surface oxygen bonding strength is a key factor for the generation of high quality NO. The nature of the oxygen species is also suggested to influence the product distribution of NO, N_2O , and N_2 . Meanwhile, from both scientific and engineering standpoints, it would be highly desirable to determine the active sites and elementary reaction pathways of the chemical looping reactions. However, such tasks can be daunting due to the highly dynamic nature of chemical looping reactions. Here, we endeavored to provide an in-depth understanding of the active sites and reaction pathways by combining operando characterization with density functional theory (DFT) calculations, which could be crucial for rational optimization of the redox catalysts and chemical looping reaction system. The V=O sites are suggested to be the catalytically active center

leading to the formation of the oxidation products. While both V=O and doubly coordinated oxygen (V-O^{II}-V) participate in the hydrogen transfer process.

The ammonia oxidation is one of the most investigated reactions in heterogeneous catalysis from mechanistic and kinetic viewpoints. Selectivity in ammonia oxidation via the conventional co-feed process is suggested to be determined by competition between NH₃ and O₂ molecules for active sites, and the intermediates' surface concentration.^{8, 11} From a reaction chemistry standpoint, the chemical looping approach we proposed here proceeds via a unique, temporally separated MvK mechanism associated with the breaking and regeneration of V=O bonds. Under the CLAO reaction scheme, the active V=O sites selectively activate and react with ammonia, while being removed from the surface. In a subsequent, temporally separated oxidation step, the active V=O sites were restored when reacting with O₂. The CLAO reaction scheme eliminates the competition between NH₃ and O₂ molecules for same active sites, in addition, the unique periodic feature of the chemical looping strategy allows for precise control of intermediates' surface concentration, both of which can contribute to the exceptional NH₃ oxidation performance.

Our results lead to an improved understanding dynamic nature of the catalytic active sites and reaction pathways of the mechanism of product formation, which remains largely unexplored and may provide hints for chemical looping community. We envision that the fundamental understanding on the crucial roles of the catalytic sites for reactants activation and the reaction pathways of the mechanism for product

formation can open new and exciting perspectives for nitric acid manufacturing in a cleaner, more sustainable and cost-efficient manner. We hope that you agree.

1. COMMENT: Page 1, the authors point that Pt loss through volatilization at high-temperature in the conventional Ostwald process. The reviewer feels that the same stability problem will occur when V_2O_5 is regenerated at 650 degree C under air, because it is exothermic and just below the melting point of V_2O_5 , where the vapor pressure is high.

RESPONSE: Thank you for this point. Ammonia oxidation is the core reaction for converting ammonia to NO_x for nitric acid. However, its strong exothermicity produces high reaction temperatures and, consequently, a serious loss of platinum, which serves as the main catalyst for ammonia oxidation and can generate volatile PtO_2 at higher temperatures (800-950 °C). Reducing the reaction temperature of ammonia oxidation with acceptable production yield and consequently reducing the catalyst consumption is attractive for nitric acid development. In the CLAO process, while the highest reaction temperature (650 °C) is below the melting point of bulk V_2O_5 (690 °C), the temperature is above the Tammann temperature so that surface spreading of V_2O_5 might occur. The stability of V_2O_5 redox catalyst was investigated for more than 1300 min of continuous operation. As shown in Fig. 2e, NH_3 conversion remains close to constant over the redox cycles. The NO selectivity keeps close to 100% with largely suppressed N_2O formation, demonstrating the excellent recyclability through redox cycling. For V_2O_5 after the CLAO redox cycling, based on

XRD and SEM characterizations, severe sintering and substantial morphological and phase changes were not observed, confirming the robustness and stability of V_2O_5 redox catalyst under the demanding reducing and oxidizing cycling. Although for a thorough cyclability investigation for practical applications at least hundreds or even thousands of hours are required to be performed. These data clearly indicate that the V_2O_5 redox catalyst is highly effective for CLAO without obvious deactivation, which is encouraging with respect to long-term use.

Furthermore, the expense of the V_2O_5 metal oxide redox catalyst (V_2O_5 : 5-7 USD/lb) is significantly lower than the scarcely limited Pt-Rh (Pt: 12,800-20,800 USD/lb; Rh: 272,400-435,800 USD/lb) catalyst, which can greatly reduce the nitric acid production cost induced by catalyst loss. Additionally, it has been proposed recently to use molten/liquid oxygen carriers or reaction system for chemical looping based methane partial oxidation, methane combustion, and Air separation to circumvent the challenges associated with carbon deposition, thermal sintering and low reaction rates.¹²⁻¹⁸ The molten reaction system enables intensified heat and mass transfer during the cyclic redox cycle, increasing the energy utilization efficiency.

2. COMMENT: Pages 2-3, Fig. 2, and Table S1, the pulse mode reaction is not appropriate for the evaluation of NH_3 oxidation. The rate is too slow to compare with the very fast NH_3+O_2 reaction in the conventional co-feeding mode. Switching two flows (NH_3 and O_2) should be a better approach for the evaluation of the performance. Not only NH_3 conversion and NO selectivity but the reaction rate, W/F, or GHSV

should be compared in Table S1. The practical value of this chemical looping system is dependent on the NO production rate.

RESPONSE: Thank you for this point. Kinetically, chemical looping reactions are generally slower than conventional catalytic oxidation/reduction reactions, which are commonly encountered in chemical looping process. This is due to the fact that the rates of redox reactions are limited by the removal and replenishment of lattice oxygen from/to the bulk of the redox catalyst. The CLAO process (pulse mode) proposed here resulted in a normalized NO production rate of $\sim 0.94 \text{ ml g}_{\text{catalyst}}^{-1} \text{ min}^{-1}$ with superior NH_3 (97.0%) conversion and NO selectivity (99.8%) at 650 °C. The NO production rate of conventional co-feeding mode reported here is $2.13 \text{ ml g}_{\text{catalyst}}^{-1} \text{ min}^{-1}$ with inferior NH_3 (87.8%) conversion and NO selectivity (22.6%) at 650 °C. While the NO production rates are on the same order of magnitude; however, the CLAO approach eliminates the costly separation units, ensures safer operation and inhibits non-selective oxidation reactions, which can contribute to the economic attractiveness of the NH_3 oxidation process.

In addition, we had also evaluated the NH_3 oxidation performances switching between two flows with different NH_3 reduction times. As shown in Fig. R1, a significantly reduced NH_3 peak with comparably increased NO peak was observed when decreasing the NH_3 reduction time from 5 min to 3 min. The NH_3 conversion and NO selectivity are calculated to be 88% and 11% at a reduction time of 5 min, and 96% and 26% at a reduction time of 3 min. These results indicate that an enhanced NH_3 conversion and NO selectivity can be realized with decreased NH_3

reduction time and optimized reducible oxygen to NH_3 ($\text{NO} : \text{N}_2\text{O} : \text{H}_2\text{O} : \text{NH}_3$) ratio as discussed above. In light of this evidence, the NH_3 oxidation performance for V_2O_5 during the NH_3 reduction half cycle of the CLAO process at $650\text{ }^\circ\text{C}$ using the pulse mode switching between NH_3 and O_2 (every single NH_3 pulse reduction is followed by the O_2 regeneration) was further investigated. Result in Fig. R1b suggests that promising NH_3 conversion ($\sim 98\%$) and NO selectivity ($\sim 99\%$) can be achieved with stable redox cycling performance. In our study, NH_3 conversion above 97% , NO selectivity above 90% can be achieved with insignificant N_2O selectivity even after 30 pulses of NH_3 followed by the O_2 regeneration (Fig. 2d in the main manuscript), with a considerable amount of lattice oxygen utilized (2.4% of the total available oxygen) during one redox cycle. Per your suggestion, the GHSV and NO production rates are compared in Table S1.

It is further noted that to ensure a relatively faster surface oxygen removal rate, two potential strategies have been proposed for the chemical looping process. The first one is through doping or lattice site substitution, in an attempt to introduce crystallographic defects that facilitate oxygen ion diffusion. The second strategy is to promote the redox metal oxide with catalytically active species on the surface to decrease the activation energy for reactant activation and to enhance the lattice oxygen consumption from the bulk. They are the focus of our upcoming work and may open up extensive possibilities for CLAO process.

From a practical standpoint, the redox catalyst's activity, available oxygen storage capacity, mechanical strength and stability, heat management, cost and environmental

risk and benefit should all be taken into consideration to optimize the redox catalyst and the chemical looping process. The great margin in redox catalyst cost (V_2O_5 : 5-7 USD/lb) over the scarcely limited Pt-Rh (Pt: 12,800-20,800 USD/lb; Rh: 272,400-435,800 USD/lb) can afford a considerable slower NO production rate in CLAO. Meanwhile, CLAO has the potential to greatly reduce N_2O emissions considering the low N_2O selectivity (0.1%). The aforementioned factors should also be considered and can contribute to the techno-economic feasibility of CLAO process development.

Fig. R1 (a) Temporal gaseous production for V_2O_5 during the NH_3 reduction half cycle of the CLAO process at 650 °C with NH_3 reduction times of 5 min and 3 min switching between NH_3 and O_2 (the gas effluent during the oxidation cycle is not shown). (b) Temporal gaseous production for V_2O_5 during the NH_3 reduction half cycle of the CLAO process at 650 °C using the pulse mode switching between NH_3 and O_2 (the gas effluent during the oxidation cycle is not shown).

3. COMMENT: Page 4, Fig.S3, as shown in Fig. S3, the selective NH_3 conversion to NO occurs only at the beginning of NH_3 injections. However, the NO selectivity

decreases with increasing pulse number. This is what happens in a flow mode. Based on the results, the amount of V_2O_5 required for selective NH_3 oxidation to NO will be very huge, e.g., more than 20 mol- V_2O_5 for 1 mol-NO, and unrealistic.

RESPONSE: Thank you for this point. Fig. S3 (now Fig. S4) illustrates the NH_3 conversion and product distribution over V_2O_5 with 180 pulses of NH_3 . The NH_3 conversion remains close to 100% with the N_2O formation largely inhibited. On the other hand, the NO selectivity decreases accompanied by the increased N_2 formation with increasing NH_3 pulse number. The NO selectivity is above 54% with 180 pulses of NH_3 despite a significant amount of lattice oxygen was extracted. The decreased NO selectivity can be attributed to the decreasing amount of active oxygen species available from the V_2O_5 redox catalyst. Due to the dynamic nature of redox catalysts undergoing chemical looping reactions, the NH_3 conversion and the molar distribution of nitrogen species largely depend upon the concentration of surface oxygen species ($V=O$), the extent of redox catalyst reduction, the stoichiometric ratios for reducible oxygen in V_2O_5 to NH_3 . This sensitivity necessitates precise control of the residence time of reactants and the conversion of the redox metal oxide. In our study, by the rational control of the NH_3 pulse number, the molar ratios for reducible oxygen in the V_2O_5 to NH_3 can be finely tuned. NH_3 conversion above 97%, NO selectivity above 90% can be achieved with insignificant N_2O selectivity even after 30 pulses of NH_3 with a considerable amount of lattice oxygen utilized (2.4% of the total available oxygen). Afterward a re-oxidation of the catalyst restored the initial NO selectivity when pulsing NH_3 . Hence, a rationally designed chemical looping ammonia oxidation

process can be envisaged by carefully controlling the residence time of reactants and reduction extend of the redox catalyst.

Compared to conventional NH_3 oxidation processes that utilize gaseous oxidants, the two-step CLAO approach relies on the lattice oxygen of a solid oxygen carrier to oxidize NH_3 . A subsequent re-oxidation step then replenishes the lattice oxygen. As a result, during the whole NH_3 oxidation process, the net oxidant consumption should be oxygen instead of V_2O_5 . Meanwhile, for a reasonable evaluation of the catalytic activity, it should be better to focus on the total amount of NO produced over the lifetime of the catalyst. The long-term stability indicates the excellent recyclability of the V_2O_5 catalyst through redox cycling. Although the direct comparison of total amount of NO produced over the lifetime of V_2O_5 and platinum gauzes is difficult due the limited data and greatly varied operation parameters. The great margin in redox catalyst cost (V_2O_5 : 5-7 USD/lb) over the scarcely limited Pt-Rh (Pt: 12,800-20,800 USD/lb; Rh: 272,400-435,800 USD/lb) can potentially lead to a higher NO production with the same catalyst cost. Additionally, it is not outrageous for a practical process with a 20:1 solid: fuel ratio. For instance, for a well-established fluidized catalytic cracking (FCC) process, the reactor operates with a 15:1 to 30:1 cat:oil ratio at the temperature of 1000 to 1100 F.¹⁹ We agree with the reviewer that the current results may not translate to practical applications in the near term. Rather, it provides some fundamental basis to design a completely different approach for NO synthesis in a cleaner, more sustainable and cost-efficient manner.

Thank you once again for your valuable comments and suggestions.

RESPONSES TO REFEREE #2's COMMENTS:

Remarks to the Author

The paper by Ruan et al. describes selective production of NO via a chemical looping experiment (NH₃/Ar and O₂/Ar) on V₂O₅. Overall, the most impressive part of the manuscript are the results from the chemical looping experiment. The side-by-side comparison of the continual flow experiment and CL is really nice and shows the value of this approach. It is difficult to assess the viability of this as a replacement technology, however, it is still extremely valuable that these results have been established. They will be of value of the field and are able to shape future thinking on how catalytic processes can be performed and optimised. Elsewhere, the manuscript contains physical characterisation (with in situ Raman) and DFT calculations of the reaction pathways. These seem a useful addition, however, it was unclear if they really enhanced the paper significantly beyond the core result in the catalytic testing. I was hoping to find more on the origin of the improved selectivity through CL - perhaps I am too greedy here - but I think it would help to expand on this point to be accepted in Nature Communications. To help improve the manuscript below, a point-by-point list comments are detailed below:

Response: Thank you very much for your kind words about our paper. We appreciate you taking the time to offer us your insightful comments and constructive suggestions. In the following sections, you will find our responses to each of your point and

suggestion.

Despite the core role and large utilization of metal oxide catalysts in chemical looping, the lack of a better understanding of the chemical nature of metal oxide catalysts and reaction pathway has limited the establishment of extensive quantitative and universal structure–reactivity–selectivity–stability relationships with the redox metal oxides. Meanwhile, the highly dynamic nature of redox catalysts undergoing chemical looping reactions also makes mechanistic investigation even more challenging than in heterogeneous catalysis. This, in turn, has delayed the discovery of new catalysts and the rational design of the chemical looping process. In order to gain insights into the dynamic nature of the active sites in metal oxide catalysts, we are coupling operando Raman spectroscopy, XRD, XPS, XANES as well as DFT calculations with catalytic reactivity studies to contribute to the endeavor for a more rational design of metal oxide catalysts. These are also fundamental to extending our understanding of redox metal oxide catalysts for the chemical looping reactions. The above techniques show that the proposed CLAO process operates via a modified, temporally separated MvK redox mechanism featuring a reversible V^{5+}/V^{4+} redox cycle. The vanadyl ($V=O$) centers act as active sites for NH_3 oxidation, leading to the formation of the oxidation products. Meanwhile both $V=O$ and doubly coordinated oxygen (O^{II}) participate in the hydrogen transfer process. The superior activity and selectivity can be attributed to the low activation energies for the successive hydrogen abstraction and facile NO formation facilitated by $V=O$.

Due to the dynamic nature of redox catalysts undergoing chemical looping

reactions, the NH_3 conversion and the molar distribution of nitrogen species depend strongly upon the concentration of surface oxygen species ($\text{V}=\text{O}$), the extent of redox catalyst reduction, the stoichiometric ratios for reducible oxygen in V_2O_5 to NH_3 . The spatially and temporally separated feature of the chemical looping strategy allows for the precise control of the residence time of reactants, the reduction extent of the redox metal oxide and ratios of the reaction intermediates (O/NH_x) on the catalyst surface. In contrast, the conventional co-feed mode faces the difficulty of maintaining these precise ratios of the reaction intermediates at every point of the catalyst. Additionally, the ammonia oxidation is one of the most investigated reactions in heterogeneous catalysis from mechanistic and kinetic viewpoints. Selectivity in ammonia oxidation via the conventional co-feed process is suggested to be determined by competition between NH_3 and O_2 molecules for the same active sites, and the intermediates' surface concentration. From a reaction chemistry standpoint, the chemical looping approach we proposed here proceeds via a unique, temporally separated MvK mechanism associated with the breaking and regeneration of $\text{V}=\text{O}$ bonds. Under the CLAO reaction scheme, the active $\text{V}=\text{O}$ sites selectively activate and react with ammonia, while being removed from the surface. In a subsequent, temporally separated oxidation step, the active $\text{V}=\text{O}$ sites were restored when reacting with O_2 . The CLAO reaction scheme eliminates the competition between NH_3 and O_2 molecules for same active sites, which can potentially contribute to the superior NH_3 oxidation performance. Therefore, we tentatively attribute high NO selectivity of CLAO to a desirable thermodynamic property of the redox metal oxide, an optimal

ratio of reducible oxygen in V_2O_5 to NH_3 (O/NH_x) on the catalyst surface, as well as the elimination of the competition between NH_3 and O_2 molecules for same active sites enabled by the unique periodic feature of the chemical looping strategy. We hope that you agree.

1. COMMENT: In Fig. 1, the schematic says O_2 depleted air. However, the O_2 source is 30% O_2 in Ar.

RESPONSE: Thank you for this point. Figure 1 is a representation of the chemical looping ammonia oxidation (CLAO) concept. During the reduction step, NH_3 is oxidized by the active lattice oxygen derived from the metal oxide redox catalyst, producing concentrated NO. In the subsequent oxidation step, the reduced metal oxide redox catalyst is re-oxidized with air, completing the redox cycle. From the potential application standpoint, Air instead of O_2 is used, which can make the system more cost-effective. To prove our proposed concept, 30% O_2 /Ar rather than Air is applied for the reoxidation step in our experiment, which can eliminate the interference of N_2 and CO_2 (amu 44, the mass overlaps with N_2O), allowing for an accurate determination of the NH_3 oxidation products. Meanwhile, for chemical looping research community, air and O_2 depleted air are commonly used as regeneration agent and depleted regeneration outlet²⁰⁻²⁴. For the sake of consistency, Air instead of 30% O_2 /Ar is employed for schematic description of the CLAO process. We hope that you agree.

2. COMMENT: Can the authors comment on the productivity per unit time as well as selectivity - this is helpful to understand the comparison between flow and CL.

RESPONSE: Thank you for this point. Kinetically, chemical looping reactions are generally slower than conventional catalytic oxidation/reduction reactions, which are commonly encountered in chemical looping process. This is due to the fact that the rates of redox reactions are limited by the removal and replenishment of lattice oxygen from/to the bulk of the redox catalyst. The CLAO process proposed here resulted in a normalized NO production rate of $\sim 0.94 \text{ ml g}_{\text{catalyst}}^{-1} \text{ min}^{-1}$ with superior NH_3 (97.0%) conversion and NO selectivity (99.8%) at 650 °C. The NO production rate of conventional co-feeding mode reported here is $2.13 \text{ ml g}_{\text{catalyst}}^{-1} \text{ min}^{-1}$ with inferior NH_3 (87.8%) conversion and NO selectivity (22.6%) at 650 °C. While the NO production rates are on the same order magnitude; however, the CLAO approach eliminates the costly separation units, ensures safer operation and inhibits non-selective oxidation reactions, which can contribute to the economic attractiveness of the NH_3 oxidation process.

3. COMMENT: Figure 2a and 2b nicely show the catalytic testing data. One of the noteworthy observations is that for the flow experiments, the N_2 selectivity decreases, before rising for the final temperature. In the discussion on the reaction pathways the only reaction route considered is NH_3 oxidation. However, there is a recent study that demonstrates that the NO_x produced through NH_3 oxidation can also create the conditions for a parallel SCR mechanism (<https://doi.org/10.1021/acscatal.0c05356>).

Considering the increase of N₂ at high temperature is this something the authors have considered?

RESPONSE: Thank you for this point. Yes, we have considered. The selective catalytic reduction (SCR) of NO_x (NO and NO₂) with NH₃ to N₂ and H₂O proceeds via the overall reactions

For the co-feed experiment as shown in Fig. R2, the N₂ selectivity decreases from 300-600 °C and increases at higher temperatures of 650 and 670 °C. If the increase of N₂ selectivity originates from selective catalytic reduction of NO_x, both an increase in NH₃ conversion and a decrease in NO_x selectivity would occur according to equation (1) and (2). While the activity test shown in Fig. R2 indicates that the NH₃ conversion plateaus at 500-650 °C. Hence, we tentatively attribute the increase of N₂ selectivity at higher temperatures to the decomposition of NO_x, which has been reported elsewhere^{8, 11, 25, 26}.

In our study, we mainly focused on validating the CLAO process via experimental and DFT investigation. The reaction pathways of the proposed CLAO process is illustrated in Fig. 6 in the main manuscript. In addition, the interaction of NH₃ and NO to produce N₂O has been investigated (Fig. 6c).

Fig. R2 NH₃ conversion and NO, N₂O, N₂ selectivity versus temperature over V₂O₅ in steady state NH₃ oxidation (co-feed mode).

4. COMMENT: Following on from point 3, clearly there are structural changes to the catalyst and compositional changes of the reactant environment. Are the authors able to comment on the impact of these with respect to the catalytic properties?

RESPONSE: Thank you for this point. For co-feed NH₃ oxidation, no apparent structural change was observed for V₂O₅ redox catalyst after the reactivity test (Fig. S5). For CLAO, powder X-ray diffraction (XRD) indicates that upon NH₃ pulse reduction, new reflections associated with V₆O₁₃ and VO₂, respectively, are identified, which progressively increased in intensity with increasing NH₃ pulse number as more lattice oxygen was extracted. XPS indicates vanadium existed in the form of V⁵⁺ and V⁴⁺ after reduction with NH₃. XANES and Raman spectroscopy results indicate that the reaction proceeds with the breaking and regeneration of V=O bonds.

In order to derive mechanistic insight into the effect of lattice oxygen extraction

on the NH₃ conversion and product distribution over V₂O₅ in the CLAO process, the NH₃ conversion, NO, N₂, N₂O selectivity as well as the accumulated lattice oxygen withdrawn from the original sample (δ) is plotted as a function of the NH₃ pulse numbers (Fig. 2d and Fig. S4). It is revealed that the selectivity of NH₃ oxidation toward NO formation is favored when the concentration of surface oxygen species (V=O) is sufficiently high for stripping all H atoms from NH₃ and for further oxidation of the resulting NH_x ($x < 3$) intermediates to NO [Eq.(5)]. The gas-phase ammonia adsorb over an active site is denoted as “s”.

The concentration of surface oxygen species decreases with increasing NH₃ pulse number due to its removal for ammonia oxidation to NO and H₂O. When the concentration of surface oxygen species (V=O) decreases, the reduced catalyst surface contains mainly strongly bound lattice oxygen species and anion vacancies in the oxide lattice, which can effectively activate the N–H bond in ammonia, dehydrogenate ammonia to form NH_x intermediates. In addition, the relative concentration of surface NH_x species should be higher. Consequently, recombination of two highly reactive NH_x fragments will prevail, leading to N₂ formation [Eq. (6)], with a decreased NO selectivity as well as insignificant amounts of N₂O (Fig. S4).

Afterward a re-oxidation of the catalyst resulted in an oxygen uptake, regenerating surface oxygen species (V=O) and restoring the initial NO selectivity when pulsing NH₃.

5. COMMENT: At 300°C there is 100% selectivity to N₂ for the CL experiment. What is the fate of the H₂. It is not seen in the MS. If it generates surface hydroxyls was this identified through XPS/Raman. Clearly, the focus of this paper is NH₃ oxidation, however, perhaps this methodology has translational impact to NH₃ decomposition and this could be an important point.

RESPONSE: Thank you for this point. Fig. 2a illustrates the ammonia conversion and product distribution over V₂O₅ as a function of temperature in typical chemical looping mode. The corresponding products evolution rates upon NH₃ injection are reproduced in Fig. S2a-h. While no H₂ was observed in the reaction temperature range, with H₂O being the only H-containing product. The surface oxygen species (V=O) catalyze NH₃ dehydrogenation, stripping all H atoms and for further oxidation of the resulting NH_x (x < 3) intermediates to N₂ and H₂O. To allow for an accurate determination of the outlet NH₃ (amu 17), H₂O (amu 17, 18) was condensed out to avoid interference before entering the quadrupole spectrometer.

Meanwhile, by decreasing the V₂O₅ loaded (0.05 g) for CLAO, H₂ signal emerged as there are less surface oxygen species available for NH_x (x < 3) oxidation (Fig. R3a). Additionally, we also measured the O 1s XPS spectra V₂O₅ sample after the CLAO experiment at 300 °C. As shown in Fig. R3b, the hydroxyl group centered at around 532 eV was identified.

Fig. R3 (a) Temporal gaseous product concentrations for V_2O_5 upon NH_3 injection at $630\text{ }^\circ\text{C}$ in chemical looping NH_3 oxidation process. (b) O 1s XPS spectra of the V_2O_5 sample after the CLAO experiment at $300\text{ }^\circ\text{C}$.

6. COMMENT: In figure 2e, why are only 9 injections shown for each repeat and not the full 30?

RESPONSE: Thank you very much for this point. The results in Fig. 2d and Fig. S4 reveals that the selectivity of NH_3 oxidation toward NO formation is determined by the degree of oxidation of the V_2O_5 catalyst surface or the concentration of surface oxygen species ($V=O$). By controlling the NH_3 pulses numbers, the reduction degree of V_2O_5 catalyst surface and the NH_3 oxidation toward NO formation can be finely tuned. In our work, NH_3 conversion and product distribution with 10 pulses over 5 cycles for more than 1,300 min of continuous operation is investigated. As can be seen, NH_3 conversion stays essentially constant with no obvious deactivation over the redox cycles, while NO selectivity keeps close to 100% with largely suppressed N_2O formation, demonstrating the excellent recyclability through redox cycling. Per your suggestion, the NH_3 conversion and product distribution with 30 pulses over 5 cycles is further investigated (Fig. R4). As can be seen, NH_3 conversion remains close to

constant over the redox cycles. Despite a higher reduction extend, and thus a decreased NO selectivity with increasing pulse number, a reoxidation of V_2O_5 catalyst fully restores the initial NO selectivity when pulsing NH_3 , indicating the excellent recyclability of the redox cycling.

Fig. R4 NH_3 conversion, NO, N_2O , N_2 selectivity versus the number of pulses over V_2O_5 at 650 °C during 5 consecutive chemical looping NH_3 oxidation redox cycles.

7. COMMENT: The physical characterisation is adequate and appear self-consistent, however in this work the XRD seems the most useful as it is able to identify the phases present. With the characterisation that is present can the authors present the degree of reduction through each technique (XRD, XPS, XANES) and comment on this comparison. For the XANES a LCF of reference standard would help.

RESPONSE: Thank you very much for this point. The phases (V_2O_5 , VO_2 , V_6O_{13}) percentage of V_2O_5 at different NH_3 pulse numbers obtained from XRD Rietveld refinement is presented in Table R1. As can be seen, the V_2O_5 percentage increases with increasing NH_3 pulse numbers. The reduction degree of V_2O_5 determined by

each technique (XRD, XPS, XANES) was compared in Table S3 in supporting information. The mean oxidation state of vanadium and the percentage of V^{4+} and V^{5+} determined by XRD and XANES (obtained from comparison of the main edge energies and XANES linear combination fitting) are in qualitative agreement with each other. The mean vanadium oxidation state is slightly higher than that calculated from the overall chemical composition of V_2O_5 at different NH_3 pulse numbers (Fig. S4), which can be attributed to the post-reaction air exposure. On the other hand, XPS results give a higher V^{4+} portion (a higher reduction extent) at different NH_3 pulse numbers, which is due to the surface sensitivity of the XPS measurement. Overall, the XRD, XPS and XANES techniques indicate a monotonic increase in vanadium reduction extent with increasing NH_3 pulses number, and vanadium undergoes reversible reduction and oxidation featuring a V^{5+}/V^{4+} redox cycle in the CLAO process.

Table R1. The phases (V_2O_5 , VO_2 , V_6O_{13}) percentage of V_2O_5 redox catalyst at different NH_3 pulse numbers obtained from XRD Rietveld refinement

Pulse number	V_2O_5 (%)	VO_2 (%)	V_6O_{13} (%)
30	93.0	7.0	-
60	86.9	13.1	1.0
90	79.0	11.0	10.0
120	61.0	14.0	25.0
150	56.0	22.0	22.0
180	52.0	30.0	18.0
Cycled	100	0	0

8. COMMENT: If XAS was measured can the EXAFS also be presented and discussed.

RESPONSE: Thank you for this point. Fourier transforms of V *k*-edge EXAFS for V₂O₅ catalysts as a function of NH₃ pulse numbers are shown in Fig. R5. The peak at 1.50 Å can be attributed to the V-O coordination in the first shell. There is a consistent increase of the amplitude of the V-O coordination sphere with increasing NH₃ pulse numbers, indicating the gradual increase of average coordination number of V-O units with enhanced reduction extent. Meanwhile, as compare to the initial sample, the averaged bond length for V₂O₅ catalysts after 90 and 180 pulses of NH₃ reduction increased from 1.80 Å to 1.92 Å (Table R2). The results are in quality agreement with the XRD observation where V₂O₅ (coordination number, 5; bond distance, 1.585-2.021 Å) gradually transformed into V₂O₄ (coordination number, 5; bond distance, 1.76-2.05 Å) and V₆O₁₃ (coordination number, 5; ~2.0 Å) with a higher V-O coordination number and averaged bond length upon NH₃ pulse reduction.

Fig. R5. Fourier transform (FT) of k-weighted V EXAFS spectra of V₂O₅ catalysts as a function of NH₃ pulse numbers.

Table R2. EXAFS fitting parameters at the V K-edge ($S_0^2=0.72$)

Sample	Path	R (\AA) ^a	$\sigma^2 \times 10^3$ (\AA^2) ^b	ΔE (eV) ^c	R factor
V ₂ O ₅	V-O	1.80±0.05	13.2±10.6	6.8±5.4	0.018
V-90	V-O	1.92±0.04	13.4±6.6	15.0±3.5	0.019
V-180	V-O	1.92±0.02	10.7±3.8	13.8±2.3	0.013

^aR: bond distance; ^b σ^2 : Debye-Waller factors; ^c ΔE : the inner potential correction. R factor: goodness of fit.

9. COMMENT: For the XANES discussion the authors comment that the pre-edge feature can change in intensity based on the symmetry of the V centres and the number of V=O units. It seems the group use the same change in the data to potentially support different ideas. There authors need to make clear what they think the major influence is here.

RESPONSE: Thank you for this point. As shown in Fig. S10, the V is fivefold coordinated in a distorted tetragonal pyramid of oxygen in V₂O₅. The apex-oxygen (V=O) distance is only 1.585 \AA , whereas the basal V-O distances vary from 1.779 to 2.017 \AA ²⁷. The distinct pre-peak in the XANES spectrum of V₂O₅ originates from the transition of a 1s electron to an empty antibonding state consisting of O 2p of the nearest oxygen and the V 3d orbitals^{28,29}. The characteristic pre-peak of the V K-edge XANES spectrum of V₂O₅ is concurrently caused by the breaking of the central symmetry and by the presence of the short V=O bond (1.585 \AA). Density functional theory calculations also assign the pre-edge feature to the V 3d–O 2p hybridization of the vanadyl bond (V=O) and the local geometry of the distorted [VO₅] square pyramid²⁷. In the case of V₆O₁₃ and VO₂, the V atom are six fold coordinated in a less

distorted octahedral [VO₆] groups, with an increased site symmetry and decreased pre-peak intensity compared to V₂O₅. The intense pre-edge peak of V in perfect octahedral symmetry was absent for VO. The decreased pre-peak intensity can be related to the decreased number of short V=O units as well as an increased site symmetry of the V atom, which is self-consistent.

10. COMMENT: For oxidation state changes in XANES analysis the edge position is not always reliable. Considering the size of the particles I think a XANES LCF with reference standards found in XRD would be more useful.

RESPONSE: Thank you very much for this point. There are several methods to determine the vanadium oxidation state and symmetry from XANES features including (1) comparison of the main edge energies; and (2) linear combination fitting.³⁰ In our study, comparison of the main edge energies (the energy measured half way up the normalized-edge step) was employed to get a quantitative determination of the average valence state of vanadium. Per your suggestion, a quantitative linear combination fit (LCF) was also performed to show the trend in the evolution of the average valence state of vanadium during the NH₃ pulse reduction process. Because V₆O₁₃ can be written as V₂O₅·2V₂O₄ and the vanadium transitions directly between two states (V⁵⁺ and V⁴⁺), the ratio of V⁵⁺ to V⁴⁺ present in each sample could be determined using V₂O₅ and V₂O₄ as the end members in a linear combination fit. As shown in Fig. R6a-e, for each spectrum the combination of standards with the lowest residual parameter was chosen (the LCF result of V₂O₅ after

30 pulses of NH_3 reduction is not shown due to the low reduction extent and relatively low signal-to-noise ratio). Fig. R7 presents the LCF results as well as the average valence state of vanadium with increasing NH_3 pulse numbers. LCF results indicates that the amount of V^{5+} decreases with simultaneous increase of the amount of V^{4+} with increasing NH_3 pulse numbers. The percentage of V^{5+} decreases from 86.6% to 50.4% as the NH_3 pulse numbers increases from 60 to 180, pointing to the gradual reduction of V^{5+} with NH_3 . The average vanadium oxidation state decreases from 4.87 to 4.50, which is close to the average vanadium oxidation determined from the comparison of the main edge energies in Fig. 3d. The mean vanadium oxidation state is slightly higher than that calculated from the overall chemical composition of V_2O_5 at different NH_3 pulse numbers (Fig. S4), which can be attributed to the post-reaction air exposure. We have added the XANES LCF results in Fig.S9 and Table S3.

Fig. R6. Normalized V-K edge XANES spectra as a function of NH_3 pulses numbers, (a) 60 pulses of NH_3 reduction, (b) 90 pulses of NH_3 reduction, (c) 120 pulses of NH_3 reduction, (d) 150 pulses of NH_3 reduction, (e) 180 pulses of NH_3 reduction, and the resultant linear combination fit with the LCF corrected reference spectra and the corresponding residuals.

Fig. R7. Oxidation state of vanadium with increasing NH₃ pulse numbers: (a) Average valence state vanadium. (b)

LCF results of XANES spectra using V₂O₅ and V₂O₄ as reference materials.

11. COMMENT: For figure 4 it would be better not to include the lines of fits (especially where the points are joined). Also the data quality is very noisy and should be commented on.

RESPONSE: Thank a lot for this suggestion. We eliminated the lines of fit for Figure 4 a, b according to your suggestion. Unfortunately, the resolution does not improve due to the overlap of the points (Fig. R8 a, b). Based on the third review's advice, we now add the *In situ* 2d Raman spectra of V₂O₅ redox catalyst during the reduction and subsequent re-oxidation as a function of time in Figure 4 a and b. Meanwhile, a clearer version of the *In situ* 3d Raman spectra of V₂O₅ redox catalyst highlighting the evolution of V=O during the reduction and subsequent re-oxidation as a function of time is provided in Fig S11 b and c.

In our work, the *In situ* Raman spectra of V₂O₅ redox catalyst was acquired at 650 °C. The characterization of the Raman signal at elevated temperatures has always been challenging. The ratio I_R(T)/I_R(298) can be expressed as

$$I_R(T)/I_R(298) = [1 + K(T)/S + (K(T)/S(K(T)/S + 2))^{1/2}]^{-1} / [1 + K(298)/S + (K(298)/S(K(298)/S + 2))^{1/2}]^{-1} \quad (5)$$

where I_R(T) and I_R(298) are the intensity of the Raman scattered light at the detection temperature of T and 298 K, respectively.³¹ K is the absorption coefficient of the powder, and S is the scattering coefficient. Precious study indicates that the intensity of the Raman signal of V₂O₅ crystallites decreases with increasing temperature^{31, 32}. This trend is well documented for solids and is usually ascribed to the effect of increased optical absorbance by the oxide³¹. Despite the high temperature, our *In situ* Raman spectra indicates that the V=O lost intensity with increasing reduction time, implying the removal of terminal oxygen. After re-oxidation, the V=O bands were reinstated.

Fig. R8 *In situ* Raman spectra of V_2O_5 redox catalyst highlighting the evolution of V=O during the (a) reduction and (b) subsequent re-oxidation as a function of time. (c) The dependence of band intensity for V=O stretching vibrations on temperature determined experimentally and theoretically (eq 7). Data are reproduced from reference [31].

Thank you once again for your valuable comments and suggestions. We worked hard to be responsive to them. We are grateful to you for taking the time and energy to help us improve the paper.

RESPONSES TO REFEREE #3's COMMENTS:

Remarks to the Author:

The manuscript titled "Selective catalytic oxidation of ammonia to nitric oxide via chemical looping" by Wang and coworkers is focused on the use of earth abundant V_2O_5 catalyst for the selective conversion of NH_3 to NO using a chemical looping approach, as an alternative to noble metal-based catalysts. Overall, the work has been well performed. The results are interesting and broadly applicable. I think this will be of interest to Nature Communications' readers. This manuscript may be published at the discretion of the editor after the following technical and general comments have been addressed.

Response: Thank you very much for your kind words about our paper. We appreciate you taking the time to offer us your insightful comments and constructive suggestions. In the following sections, you will find our responses to each of your point and suggestion.

1. COMMENT: For comparison of continuous and CLAO operations have been made at different chemical potential of O_2 . The reviewer suggests that in continuous operation, the effective O_2 percentage of 30% in the combined feed is perhaps a better comparison of selectivity. In this study, the O_2 is further diluted by 5 times giving an effective percentage of 6% I believe. It may not change the overall conclusion, but this will be a good experiment to show the readers. A more systematic investigation of O_2 partial pressure and its influence on selectivity for continuous operation may be

helpful.

RESPONSE: Thank you very much for this point. In our current study, the ammonia oxidation was performed using a feed containing an excess of O₂ over NH₃ (NH₃ 1.5 ml/min, O₂ 2.25 ml/min, Ar 33.75 ml/min) considering the stoichiometry of the overall reaction: $4\text{NH}_3 + 5\text{O}_2 = 4\text{NO} + 6\text{H}_2\text{O}$. The NH₃ conversion and NO selectivity of the co-feed mode are significantly lower compared to the CLAO process with a high N₂O selectivity. Per your advice, we now compared the continuous operation with the effective O₂ percentage of 30% (NH₃ 1.5 ml/min, O₂ 12.86 ml/min, Ar 28.5 ml/min) with the CLAO process. As shown in Fig. R9 a, ammonia conversion increases from 26.1% to 87.8% at increasing temperature and reaches a plateau from 500 to 650 °C. The maximum ammonia conversion (87.8%) is considerably lower than that in the chemical looping mode (97.0%). In terms of product selectivity, NO peaks at 300 °C with an inferior selectivity of 34.2% and decreases significantly when rising the temperature to 400 °C. Since in the ternary O₂-NH₃-NO system, a competition between the NO reduction by NH₃ and the NH₃ oxidation with O₂ to NO can be expected. The decline in NO formation at 400 °C may due to the secondary reactions involving NO and NH₃. A further increase in temperature resulted in increasing formation of NO. After passing through a maximum at 600 °C, NO formation slightly decreases with a further increase in temperature. This is in good agreement with literature data on co-feed ammonia oxidation using metal oxide redox catalysts²⁶. In addition, a high N₂O selectivity (13.6%-26.9%) is observed over the

reaction temperature range. Now we add this result in Fig. 2b in the main manuscript for comparison the NH_3 conversion and product selectivity.

To investigate the O_2 partial pressure and its influence on production formation for NH_3 oxidation via continuous operation, the reaction was further studied with the effective O_2 percentage of 18% and 42% with excess O_2 over NH_3 . As illustrated in Fig. R9 b, c, the NH_3 conversion and distribution of the reaction products (NO , NO_2 , N_2) versus temperature are similar. Ammonia conversion increases with increasing temperature and reaches a plateau from 500 to 650 °C. Concerning product selectivity, NO peaks at 300 °C decreases significantly when rising the temperature to 400 °C. A further increase in temperature led to an increase in NO_2 formation before passing through a maximum. The ammonia conversion and NO selectivity compared less favorable with CLAO process despite using feeds containing an excess of O_2 . We now add this result in Fig. S3a, b in supporting information for comparison the NH_3 conversion and product selectivity.

Fig. R9. NH₃ conversion and the distribution of reaction products vs. temperature for steady state NH₃ oxidation process (co-feed mode) with the effective O₂ percentage of (a) 30% (NH₃ 1.5 ml/min, O₂ 12.86 ml/min, Ar 28.5 ml/min), (b) 18% (NH₃ 1.5 ml/min, O₂ 6.59ml/min, Ar 28.5 ml/min), (c) 42% (NH₃ 1.5 ml/min, O₂ 21.72 ml/min, Ar 28.5 ml/min) at 650 °C.

2. COMMENT: The calculation of rates of oxygen removal are valid at low rates of oxygen removal. The reviewer is curious at what level of conversion of perhaps V₂O₅ were these oxygen removal rates estimated from. In other words, were these initial rates? It appears that these levels of V₂O₅ conversions are low as seen in Figure 2d. If it is the case, I would recommend the authors mention it in the context that these rates reflect the initial oxygen removal rates from V₂O₅.

RESPONSE: Thank you for this point. The calculation of rates of oxygen removal are based on the maximum lattice oxygen extraction rate for NH₃ oxidation during the

first pulse, i.e. at a low level of oxygen removal. Per your suggestion, I have added the initial maximum oxygen extraction rates for clarity.

3. COMMENT: The XPS of the catalyst post reaction in continuous mode would be helpful to include. From the data presented, it appears that the authors find that over reduction is the cause of the change in product distribution. Can this be arrested with using a higher concentration of O₂ in continuous mode (Please see comment 1)?

RESPONSE: Thank you very much for this point. In our current study, the ammonia oxidation was performed using a feed containing an excess of O₂ over NH₃ in continuous mode. The XRD patterns of the as prepared V₂O₅ sample before and after NH₃ oxidation reactivity test (in Fig. S5) indicate that no apparent structural change was observed. Per your advice, we have also studied the XPS of the V₂O₅ catalyst post reaction in continuous mode. In agreement with the XRD results, there was no apparent reduction for V₂O₅ catalyst after NH₃ oxidation reaction. As shown in Fig. R10, a strong peak centered at 517.4 eV along with a broad satellite peak at a higher binding energy was identified, which agrees well with the signature of V⁵⁺ in V₂O₅. The XPS results are incorporated in supporting information in Fig. S7.

The NH₃ oxidation performance was investigated using an excess of O₂ with different O₂ partial pressures in continuous mode. As shown in Fig. R9 a-c, the NH₃ conversion and distribution of the reaction products (NO, NO₂) versus temperature are similar. Ammonia conversion increases with increasing temperature and reaches a plateau from 500 to 650 °C. In regard to product selectivity, NO peaks at 300 °C

decreases significantly when rising the temperature to 400 °C. A further increase in temperature led to an increase in NO formation before passing through a maximum. Unfortunately, the ammonia conversion and NO selectivity are significantly lower as compared to CLAO process despite using feeds containing an excess of O₂. This result is in qualitative agreement with literature data on co-feed ammonia oxidation using metal oxide redox catalysts²⁶.

Due to the dynamic nature of redox catalysts undergoing chemical looping reactions, the NH₃ conversion and the molar distribution of nitrogen species depend strongly upon the concentration of surface oxygen species (V=O), the extent of redox catalyst reduction, the stoichiometric ratios for reducible oxygen in V₂O₅ to NH₃. The spatially and temporally separated feature of the chemical looping strategy allows for the precise control of the residence time of reactants, the reduction extent of the redox metal oxide and ratios of the reaction intermediates (O/NH_x) on the catalyst surface. In contrast, the conventional co-feed mode faces the difficulty of maintaining these precise ratios of the reaction intermediates at every point of the catalyst. Additionally, the ammonia oxidation is one of the most investigated reactions in heterogeneous catalysis from mechanistic and kinetic viewpoints. Selectivity in ammonia oxidation via the conventional co-feed process is suggested to be determined by competition between NH₃ and O₂ molecules for active sites, and the intermediates' surface concentration. From a reaction chemistry standpoint, the chemical looping approach we proposed here proceeds via a unique, temporally separated MvK mechanism associated with the breaking and regeneration of V=O bonds. Under the CLAO

reaction scheme, the active V=O sites selectively activate and react with ammonia, while being removed from the surface. In a subsequent, temporally separated oxidation step, the active V=O sites were restored when reacting with O₂. The CLAO reaction scheme eliminates the competition between NH₃ and O₂ molecules for same active sites, which can potentially contribute to the superior NH₃ oxidation performance.

Fig. R10. V 2p XPS spectra of V₂O₅ redox catalyst after activity measurement in conventional co-feed mode at 650 °C.

4. COMMENT: The reviewer suggests that Figure 4 a and b be shown in alternative 2 d format. In the current 3 d format, it is difficult to read and understand the figure.

RESPONSE: Thank you very much for this suggestion. Per your advice, we now add the *In situ* 2d Raman spectra of V₂O₅ redox catalyst during the reduction and subsequent re-oxidation as a function of time in Figure 4 a and b. Meanwhile, a

clearer version of the *In situ* 3d Raman spectra of V₂O₅ redox catalyst highlighting the evolution of V=O during the reduction and subsequent re-oxidation as a function of time is provided in Fig. S11 b and c. We hope you find these revisions rise to your expectations.

5. COMMENT: The authors are recommended to clearly mention whether they are using electronic energies and not enthalpies or free energies for the DFT calculations.

RESPONSE: We thank the reviewer for bringing this into our attention. The energies in the energy potential profiles are the total electronic energies computed by DFT. We have therefore added the corresponding descriptions in the Computational Details in Page 10: “*All the computed energies are the total electronic energies unless otherwise stated.*”

6. COMMENT: The interpretation of the DFT data on the temperature effects on product selectivity is weak in the opinion of the reviewer. Overall, this section needs some realignment. The authors mention, “At low temperature, both reactions can occur, however, the forward reaction of NO formation is slower than its reverse reaction due to its endothermic nature, making the overall NO formation rate lower than that of N₂ formation. Whereas, at high temperature, NO formation is more favourable due to its overall lower reaction barriers from the thermodynamics standpoint. The calculation results are coherent with experimental observations.”

In the opinion of the reviewer, this argument is not correct. The reaction barriers do

not determine thermodynamics. The only consistent feature here is the lower barrier towards NO formation than N₂ that explains the high temperature observation. If the authors want to use the calculations to justify the selectivity, I recommend using a microkinetic model that will help account for surface coverages and the relative product formation.

RESPONSE: Thank you for this point. We agree that these DFT results are not sufficient to justify the product selectivity and thus will definitely perform the microkinetic modeling in the near future, which would be very helpful to address the problems in the product selectivity. Nevertheless, considering the complexity of the reaction networks and the huge amount of workload in the microkinetic modeling, we believe that further microkinetic modeling would lead to an excellent theoretical project, but it is beyond the scope of this manuscript. At this stage, we mainly want to focus on the performances of V₂O₅ in catalyzing NH₃ to NO at high temperatures. Therefore, we calculated the Gibbs free energy corrections for the energy barriers of the key steps of NO (NH₂* → NH*) and N₂ (2NH₂* → NH₂NH₂*) formations, respectively. As shown in Fig. S16 in the revised version, the free energy barriers of NH₂* → NH* are always much lower than those of 2NH₂* → NH₂NH₂* with the increase of temperature. These results well explained the preferential selectivity of NO at high temperatures as observed by the experiments. Please refer to the added discussions highlight in page 8 in the revised draft.

“... their relative rates are largely determined by the second highest barriers as highlighted in orange, i.e. the second hydrogen transfer (0.79 eV) (NH₂ → NH*)*

for NO formation and the two NH₂ combination (2NH₂* → NH₂NH₂*) (0.90 eV) for N₂ formation. To take the temperature effects into account, the Gibbs free energy corrections over temperature for these two key steps were calculated as shown in Fig. S16. Results show that the free energy barriers of NH₂* → NH* are always much lower than those of 2NH₂* → NH₂NH₂*, which is coherent with experimental observations whereby NO prevails at high temperature. It should be note that at this stage, the existing DFT calculations are not sufficient to justify the product selectivity considering the complexity of the reaction networks. This problem may be addressed in future studies by using more advanced kinetic analysis methods like microkinetic modeling.”*

7. COMMENT: If the redox cycles are indeed useful for selective conversion of NH₃ to NO on V₂O₅, can the authors comment or have they tried to use other single metal oxides such as WO₃ or MoO₃ for the same reaction. If yes, can they include the data to generalize their conclusions. If not, can the authors perform these studies for comparison?

RESPONSE: Thank you for this point. In this study, typical single (V₂O₅, CeO₂, Fe₃O₄, SnO₂) and binary redox metal oxide catalysts (ZnFe₂O₄, CoFe₂O₄, NiFe₂O₄) were screened for CLAO (Table S1). The desirable thermodynamic property of V₂O₅ with proper surface oxygen bonding strength is a key factor for the generation of high quality NO. The nature of the oxygen species is also suggested to influence the product distribution of NO, N₂O, and N₂. Per you suggestion, WO₃ and MoO₃ were

also investigated for CLAO. Fig. R11 shows the normalized transient products (NO, N₂O, N₂, H₂) evolution rates upon NH₃ injection over WO₃ at (a) 600 °C, (b) 700 °C, and MoO₃ at (c) 750 °C, (d) 850 °C, (e) 950 °C for CLAO. For both redox metal oxide catalysts, N₂ is the dominate product. While the NH₃ conversion is 92.2% and 91.1% for MO₃ at 600 °C and 700 °C, respectively. The selectivity towards NO and N₂O are 1.4% and 6.6% at 600 °C, and 4.3% and 6.0% at 700 °C. Note that for temperature higher than 700 °C, severe sublimation occurs for MO₃, which hinders its catalytic performance evaluation. For WO₃, the NH₃ conversion is 95.0%, 92.4% and 93.3% at 750 °C, 850 °C and 950 °C, respectively. The selectivity towards NO and N₂O are 0 and 5.8% at 750 °C, 0 and 5.1% at 850 °C and 0 and 5.3% at 950 °C. We have included the CLAO performances of WO₃ and MoO₃ in Table S1 for comparison.

Fig. R11 Normalized transient products (NO , N_2O , N_2 , H_2) evolution rates upon NH_3 injection over WO_3 at (a) 600°C , (b) 700°C , and MoO_3 at (c) 750°C , (d) 850°C , (e) 950°C for CLAO.

Thank you once again for your valuable comments and suggestions. We worked hard to be responsive to them. We are grateful to you for taking the time and energy to help us improve the paper.

References

1. L. A. I. V.A. Sadykov, I.A. Zolotarskii, L.N. Bobrova, A.S. Noskov, V.N. Parmon, E.A. Brushtein, T.V. Telyatnikova, V.I. Chernyshev, V.V. Lunin., Falter, *Appl. Catal. A*, 2000, **204**, 59-87.
2. J. Schäffer, V. A. Kondratenko, N. Steinfeldt, M. Sebek and E. V. Kondratenko, *Journal of Catalysis*, 2013, **301**, 210-216.
3. J. Perez-Ramirez and B. Vigeland, *Angew. Chem., Int. Ed.*, 2005, **44**, 1112-1115.
4. L. Hannevold, O. Nilsen, A. Kjekshus and H. Fjellvåg, *Appl. Catal. A*, 2005, **284**, 163-176.
5. E. V. Kondratenko and J. Pérez-Ramírez, *Appl. Catal. A*, 2005, **289**, 97-103.
6. J. Perez-Ramirez and E. V. Kondratenko, *Chem Commun (Camb)*, 2004, 376-377.
7. M. Baerns, R. Imbihl, V. Kondratenko, R. Kraehnert, W. Offermans, R. Vansanten and A. Scheibe, *Journal of Catalysis*, 2005, **232**, 226-238.
8. J. Perezramirez, E. Kondratenko, V. Kondratenko and M. Baerns, *Journal of Catalysis*, 2005, **229**, 303-313.
9. J. Perezramirez, E. Kondratenko, G. Novellerruth and J. Ricart, *Journal of Catalysis*, 2009, **261**, 217-223.
10. J. Pérez-Ramírez, F. Kapteijn, K. Schöffel and J. A. Moulijn, *Applied Catalysis B: Environmental*, 2003, **44**, 117-151.
11. J. Perezramirez, E. Kondratenko, V. Kondratenko and M. Baerns, *Journal of Catalysis*, 2004, **227**, 90-100.
12. M. M. Sarafraz, M. Jafarian, M. Arjomandi and G. J. Nathan, *Applied Energy*, 2017, **195**, 702-712.
13. M. Jafarian, M. Arjomandi and G. J. Nathan, *Chemical Engineering Research and Design*, 2017, **120**, 69-81.
14. D. C. Upham, Z. R. Snodgrass, M. T. Zavareh, T. B. McConnaughy, M. J. Gordon, H. Metiu and E. W. McFarland, *Chemical Engineering Science*, 2017, **160**, 245-253.
15. X. Ao, H. Wang and Y. Wei, *Energy Conversion and Management*, 2008, **49**, 2063-2068.
16. M. M. Sarafraz, M. Jafarian, M. Arjomandi and G. J. Nathan, *Int. J. Hydrogen Energy*, 2018, **43**, 4195-4210.
17. D. C. Upham, V. Agarwal, A. Khechfe, Z. R. Snodgrass, M. J. Gordon, H. Metiu and E. W. McFarland, *Science*, 2017, **358**, 917-921.
18. C. Palmer, D. C. Upham, S. Smart, M. J. Gordon, H. Metiu and E. W. McFarland, *Nature Catalysis*, 2020, **3**, 83-89.
19. *Patent Number: 5,324,418*, 1994,
20. L. Qin, Z. Cheng, M. Guo, M. Xu, J. A. Fan and L.-S. Fan, *ACS Energy Letters*, 2017, **2**, 70-74.
21. M. Keller, H. Leion and T. Mattisson, *AIChE J.*, 2016, **62**, 38-45.
22. M. Rydén, M. Johansson, A. Lyngfelt and T. Mattisson, *Energy Environ. Sci.*, 2009, **2**, 970.
23. A. Thursfield, A. Murugan, R. Franca and I. S. Metcalfe, *Energy Environ. Sci.*, 2012, **5**, 7421.
24. M. Kathe, A. Empfield, P. Sandvik, C. Fryer, Y. Zhang, E. Blair and L.-S. Fan, *Energy Environ. Sci.*, 2017.
25. G. Biousque and Y. Schuurman, *Journal of Catalysis*, 2010, **276**, 306-313.
26. L. A. Isupova, E. F. Sutormina, N. A. Kulikovskaya, L. M. Plyasova, N. A. Rudina, I. A. Ovsyannikova, I. A. Zolotarskii and V. A. Sadykov, *Catalysis Today*, 2005, **105**, 429-435.
27. O. S ˇ ipr, A. S ˇ imu ˇ nek, S. Bocharov, Th. Kirchner and G. Dra ˇ ger, *Physical Review B*, 1999, **60**,

14115-14127.

28. T. D. Tullius, W. O. Gillum, R. M. K. Carlson and K. O. Hodgson, *J. Am. Chem. Soc.*, 1980, **102**, 5670-5676.
29. T. Yamamoto, *X-Ray Spectrometry*, 2008, **37**, 572-584.
30. P. Chaurand, J. Rose, V. Briois, M. Salome, O. Proux, V. Nassif, L. Olivi, J. Susini, J. L. Hazemann and J. Y. Bottero, *J Phys Chem B*, 2007, **111**, 5101-5110.
31. Shuibo Xie, Enrique Iglesia and a. A. T. Bell, *J. Phys. Chem. B*, 2001, **105**, 5144-5152.
32. Bryan Olthof, Andrei Khodakov, Alexis T. Bell and E. Iglesia, *J. Phys. Chem. B* 2000, **104**, 1516-1528.

REVIEWER COMMENTS

Reviewer #1 (Remarks to the Author):

The reviewer finds that the authors revised the manuscript and now it is suitable for publication.

Reviewer #2 (Remarks to the Author):

I have uploaded a point by point response to the rebuttal as an attachment.

In summary, I would like the authors to more fully engage with some of the constructive comments and for this to be better reflected in the main text / ESI.

Reviewer #3 (Remarks to the Author):

The authors have significantly improved the manuscript after incorporating all the reviewers' comments. The editor may accept the manuscript after the following comments are addressed.

1. The methods used for entropy calculations in Gibbs free energy barriers are recommended to be included in the methods section.
2. Can the authors clarify whether the catalysts after continuous mode were exposed to air prior to XPS. If yes, what procedures were used to minimize the exposure? If the spent catalysts were exposed to air, this reviewer recommends a control where the authors can show how the XPS with limited exposure times compare with the data shown.
3. R10 is an important figure in my view to be included in SI and the relevant discussion is recommended to be added. The comparison with the XPS in main text is suggested to be included.

RESPONSES TO REFEREE #2's COMMENTS:

Remarks to the Author

The paper by Ruan et al. describes selective production of NO via a chemical looping experiment (NH₃/Ar and O₂/Ar) on V₂O₅. Overall, the most impressive part of the manuscript are the results from the chemical looping experiment. The side-by-side comparison of the continual flow experiment and CL is really nice and shows the value of this approach. It is difficult to assess the viability of this as a replacement technology, however, it is still extremely valuable that these results have been established. They will be of value of the field and are able to shape future thinking on how catalytic processes can be performed and optimised. Elsewhere, the manuscript contains physical characterisation (with in situ Raman) and DFT calculations of the reaction pathways. These seem a useful addition, however, it was unclear if they really enhanced the paper significantly beyond the core result in the catalytic testing. I was hoping to find more on the origin of the improved selectivity through CL - perhaps I am too greedy here - but I think it would help to expand on this point to be accepted in Nature Communications. To help improve the manuscript below, a point-by-point list comments are detailed below:

Response: Thank you very much for your kind words about our paper. We appreciate you taking the time to offer us your insightful comments and constructive suggestions. In the following sections, you will find our responses to each of your point and suggestion.

Despite the core role and large utilization of metal oxide catalysts in chemical looping, the lack of a better understanding of the chemical nature of metal oxide catalysts and reaction pathway has limited the establishment of extensive quantitative and universal structure–reactivity–selectivity–stability relationships with the redox metal oxides. Meanwhile, the highly dynamic nature of redox catalysts undergoing chemical looping reactions also makes mechanistic investigation even more challenging than in heterogeneous catalysis. This, in turn, has delayed the discovery of new catalysts and the rational design of the chemical looping process. In order to gain insights into the dynamic nature of the active sites in metal oxide catalysts, we are coupling operando Raman spectroscopy, XRD, XPS, XANES as well as DFT calculations with catalytic reactivity studies to contribute to the endeavor for a more rational design of metal oxide catalysts. These are also fundamental to extending our understanding of redox metal oxide catalysts for the chemical looping reactions. The above techniques show that the proposed CLAO process operates via a modified, temporally separated MvK redox mechanism featuring a reversible V^{5+}/V^{4+} redox cycle. The vanadyl ($V=O$) centers act as active sites for NH_3 oxidation, leading to the formation of the oxidation products. Meanwhile both $V=O$ and doubly coordinated oxygen (O^{II}) participate in the hydrogen transfer process. The superior activity and selectivity can be attributed to the low activation energies for the successive hydrogen abstraction and facile NO formation facilitated by $V=O$.

Due to the dynamic nature of redox catalysts undergoing chemical looping reactions, the NH_3 conversion and the molar distribution of nitrogen species depend

strongly upon the concentration of surface oxygen species ($V=O$), the extent of redox catalyst reduction, the stoichiometric ratios for reducible oxygen in V_2O_5 to NH_3 . The spatially and temporally separated feature of the chemical looping strategy allows for the precise control of the residence time of reactants, the reduction extent of the redox metal oxide and ratios of the reaction intermediates (O/NH_x) on the catalyst surface. In contrast, the conventional co-feed mode faces the difficulty of maintaining these precise ratios of the reaction intermediates at every point of the catalyst. Additionally, the ammonia oxidation is one of the most investigated reactions in heterogeneous catalysis from mechanistic and kinetic viewpoints. Selectivity in ammonia oxidation via the conventional co-feed process is suggested to be determined by competition between NH_3 and O_2 molecules for the same active sites, and the intermediates' surface concentration. From a reaction chemistry standpoint, the chemical looping approach we proposed here proceeds via a unique, temporally separated MvK mechanism associated with the breaking and regeneration of $V=O$ bonds. Under the CLAO reaction scheme, the active $V=O$ sites selectively activate and react with ammonia, while being removed from the surface. In a subsequent, temporally separated oxidation step, the active $V=O$ sites were restored when reacting with O_2 . The CLAO reaction scheme eliminates the competition between NH_3 and O_2 molecules for same active sites, which can potentially contribute to the superior NH_3 oxidation performance. Therefore, we tentatively attribute high NO selectivity of CLAO to a desirable thermodynamic property of the redox metal oxide, an optimal ratio of reducible oxygen in V_2O_5 to NH_3 (O/NH_x) on the catalyst surface, as well as

the elimination of the competition between NH_3 and O_2 molecules for same active sites enabled by the unique periodic feature of the chemical looping strategy. We hope that you agree.

Response: Since the authors have just restated the points in the original paper, I can only restate what I suggested initially – I was hoping for a more precise explanation on the rationale behind the improvement presented in the work. I did not find the characterization that informative, as I already indicated. One needs to consider the limitations of each technique, including in relation to the size domains/heterogeneity of the sample, and the conditions in which the study has been carried out. I do not feel the authors have fully engaged in these constructive comments. I am a supporter of the work, but do feel that the characterization presented has some limitations. Essentially, whether it is XANES, XRD, or Raman, the data show that under reducing conditions the sample is reduced and that under oxidizing conditions the sample is oxidized – this is hardly a surprise. It does confirm that O is abstracted from the metal oxide, so I accept the MvK inference. However, the emphasis of my question was for the authors to put forward a more precise description of why the chemical looping experiment reduces the amount of over-oxidation products – I did not find this in the original manuscript and this point is still missing as far as I can see.

1. COMMENT: In Fig. 1, the schematic says O_2 depleted air. However, the O_2 source is 30% O_2 in Ar.

RESPONSE: Thank you for this point. Figure 1 is a representation of the chemical looping ammonia oxidation (CLAO) concept. During the reduction step, NH_3 is oxidized by the active lattice oxygen derived from the metal oxide redox catalyst, producing concentrated NO. In the subsequent oxidation step, the reduced metal oxide redox catalyst is re-oxidized with air, completing the redox cycle. From the potential application standpoint, Air instead of O_2 is used, which can make the system more cost-effective. To prove our proposed concept, 30% O_2/Ar rather than Air is applied for the reoxidation step in our experiment, which can eliminate the interference of N_2 and CO_2 (amu 44, the mass overlaps with N_2O), allowing for an accurate determination of the NH_3 oxidation products. Meanwhile, for chemical looping research community, air and O_2 depleted air are commonly used as regeneration agent and depleted regeneration outlet²⁰⁻²⁴. For the sake of consistency, Air instead of 30% O_2/Ar is employed for schematic description of the CLAO process. We hope that you agree.

No – I do not agree. If you have used a certain amount of oxygen the captions used should reflect that. Anything else is disingenuous to the readership and poor scientific practice. This is a rather simple point and I do not understand the reluctance of the authors here. Ultimately, it just improves the clarity of the manuscript.

2. COMMENT: Can the authors comment on the productivity per unit time as well as selectivity - this is helpful to understand the comparison between flow and CL.

RESPONSE: Thank you for this point. Kinetically, chemical looping reactions are generally slower than conventional catalytic oxidation/reduction reactions, which are commonly encountered in chemical looping process. This is due to the fact that the rates of redox reactions are limited by the removal and replenishment of lattice oxygen from/to the bulk of the redox catalyst. The CLAO process proposed here resulted in a normalized NO production rate of $\sim 0.94 \text{ ml g}_{\text{catalyst}}^{-1} \text{ min}^{-1}$ with superior NH_3 (97.0%) conversion and NO selectivity (99.8%) at 650 °C. The NO production rate of conventional co-feeding mode reported here is $2.13 \text{ ml g}_{\text{catalyst}}^{-1} \text{ min}^{-1}$ with inferior NH_3 (87.8%) conversion and NO selectivity (22.6%) at 650 °C. While the NO production rates are on the same order magnitude; however, the CLAO approach eliminates the costly separation units, ensures safer operation and inhibits non-selective oxidation reactions, which can contribute to the economic attractiveness of the NH_3 oxidation process.

The authors need to include a discussion of these production rates in the main text, I could not find this in the revision. Apologies if I have missed this. Rather than restating conversion/selectivity values, the authors should comment on the relative usefulness of the values and how the chemical looping value could be further improved.

3. COMMENT: Figure 2a and 2b nicely show the catalytic testing data. One of the noteworthy observations is that for the flow experiments, the N_2 selectivity decreases,

before rising for the final temperature. In the discussion on the reaction pathways the only reaction route considered is NH₃ oxidation. However, there is a recent study that demonstrates that the NO_x produced through NH₃ oxidation can also create the conditions for a parallel SCR mechanism (<https://doi.org/10.1021/acscatal.0c05356>). Considering the increase of N₂ at high temperature is this something the authors have considered?

RESPONSE: Thank you for this point. Yes, we have considered. The selective catalytic reduction (SCR) of NO_x (NO and NO₂) with NH₃ to N₂ and H₂O proceeds via the overall reactions

For the co-feed experiment as shown in Fig. R2, the N₂ selectivity decreases from 300-600 °C and increases at higher temperatures of 650 and 670 °C. If the increase of N₂ selectivity originates from selective catalytic reduction of NO_x, both an increase in NH₃ conversion and a decrease in NO_x selectivity would occur according to equation (1) and (2). While the activity test shown in Fig. R2 indicates that the NH₃ conversion plateaus at 500-650 °C. Hence, we tentatively attribute the increase of N₂ selectivity at higher temperatures to the decomposition of NO_x, which has been reported elsewhere^{8, 11, 25, 26}.

In our study, we mainly focused on validating the CLAO process via experimental

and DFT investigation. The reaction pathways of the proposed CLAO process is illustrated in Fig. 6 in the main manuscript. In addition, the interaction of NH_3 and NO to produce N_2O has been investigated (Fig. 6c).

Fig. R2 NH_3 conversion and NO , N_2O , N_2 selectivity versus temperature over V_2O_5 in steady state NH_3 oxidation (co-feed mode).

I can see the authors have engaged with this comment, which is welcome, but I am not sure I necessarily agree. The NH_3 conversion plateaus from ~ 500C. However, the selectivity profile continues to evolve. For example, at 550C there is more NO and less N_2 produced than at 500C. It does not necessarily follow, that a change in selectivity profile requires the conversion of products produced at the previous temperature, e.g. in this case it is likely that NH_3 is converted into different products. The same could also be true at higher temperature – you don't need to see an increase in NH_3 conversion for there to be the possibility of an SCR type process. Of course, this could be easily resolved with spatially resolved (rather than end pipe) gas analysis, however, this is not within the scope of the work and I accept that. However, it would

improve the manuscript to add a discussion about the changing selectivity profile, which I do not find in the main text.

4. COMMENT: Following on from point 3, clearly there are structural changes to the catalyst and compositional changes of the reactant environment. Are the authors able to comment on the impact of these with respect to the catalytic properties?

RESPONSE: Thank you for this point. For co-feed NH_3 oxidation, no apparent structural change was observed for V_2O_5 redox catalyst after the reactivity test (Fig. S5). For CLAO, powder X-ray diffraction (XRD) indicates that upon NH_3 pulse reduction, new reflections associated with V_6O_{13} and VO_2 , respectively, are identified, which progressively increased in intensity with increasing NH_3 pulse number as more lattice oxygen was extracted. XPS indicates vanadium existed in the form of V^{5+} and V^{4+} after reduction with NH_3 . XANES and Raman spectroscopy results indicate that the reaction proceeds with the breaking and regeneration of $\text{V}=\text{O}$ bonds.

In order to derive mechanistic insight into the effect of lattice oxygen extraction on the NH_3 conversion and product distribution over V_2O_5 in the CLAO process, the NH_3 conversion, NO , N_2 , N_2O selectivity as well as the accumulated lattice oxygen withdrawn from the original sample (δ) is plotted as a function of the NH_3 pulse numbers (Fig. 2d and Fig. S4). It is revealed that the selectivity of NH_3 oxidation toward NO formation is favored when the concentration of surface oxygen species ($\text{V}=\text{O}$) is sufficiently high for stripping all H atoms from NH_3 and for further oxidation of the resulting NH_x ($x < 3$) intermediates to NO [Eq.(5)]. The gas-phase ammonia

adsorb over an active site is denoted as “s”.

The concentration of surface oxygen species decreases with increasing NH_3 pulse number due to its removal for ammonia oxidation to NO and H_2O . When the concentration of surface oxygen species ($\text{V}=\text{O}$) decreases, the reduced catalyst surface contains mainly strongly bound lattice oxygen species and anion vacancies in the oxide lattice, which can effectively activate the N–H bond in ammonia, dehydrogenate ammonia to form NH_x intermediates. In addition, the relative concentration of surface NH_x species should be higher. Consequently, recombination of two highly reactive NH_x fragments will prevail, leading to N_2 formation [Eq. (6)], with a decreased NO selectivity as well as insignificant amounts of N_2O (Fig. S4).

Afterward a re-oxidation of the catalyst resulted in an oxygen uptake, regenerating surface oxygen species ($\text{V}=\text{O}$) and restoring the initial NO selectivity when pulsing NH_3 .

My comment was more focused on localized variations to the catalyst structure at different points of the reaction profile. I do not feel this is an essential point for the manuscript, but it is something the authors may wish to consider as they progress their work.

5. COMMENT: At 300°C there is 100% selectivity to N_2 for the CL experiment.

What is the fate of the H_2 . It is not seen in the MS. If it generates surface hydroxyls

was this identified through XPS/Raman. Clearly, the focus of this paper is NH_3 oxidation, however, perhaps this methodology has translational impact to NH_3 decomposition and this could be an important point.

RESPONSE: Thank you for this point. Fig. 2a illustrates the ammonia conversion and product distribution over V_2O_5 as a function of temperature in typical chemical looping mode. The corresponding products evolution rates upon NH_3 injection are reproduced in Fig. S2a-h. While no H_2 was observed in the reaction temperature range, with H_2O being the only H-containing product. The surface oxygen species ($\text{V}=\text{O}$) catalyze NH_3 dehydrogenation, stripping all H atoms and for further oxidation of the resulting NH_x ($x < 3$) intermediates to N_2 and H_2O . To allow for an accurate determination of the outlet NH_3 (amu 17), H_2O (amu 17, 18) was condensed out to avoid interference before entering the quadrupole spectrometer.

Meanwhile, by decreasing the V_2O_5 loaded (0.05 g) for CLAO, H_2 signal emerged as there are less surface oxygen species available for NH_x ($x < 3$) oxidation (Fig. R3a). Additionally, we also measured the O 1s XPS spectra V_2O_5 sample after the CLAO experiment at 300 °C. As shown in Fig. R3b, the hydroxyl group centered at around 532 eV was identified.

Fig. R3 (a) Temporal gaseous product concentrations for V_2O_5 upon NH_3 injection at $630\text{ }^\circ\text{C}$ in chemical looping NH_3 oxidation process. (b) O 1s XPS spectra of the V_2O_5 sample after the CLAO experiment at $300\text{ }^\circ\text{C}$.

Thi

This does not answer the question. What proportion of H ends up in H_2O , surface hydroxyls, or H_2 ? Can the authors include a discussion of this also in the main text.

6. COMMENT: In figure 2e, why are only 9 injections shown for each repeat and not the full 30?

RESPONSE: Thank you very much for this point. The results in Fig. 2d and Fig. S4 reveals that the selectivity of NH_3 oxidation toward NO formation is determined by the degree of oxidation of the V_2O_5 catalyst surface or the concentration of surface oxygen species ($V=O$). By controlling the NH_3 pulses numbers, the reduction degree of V_2O_5 catalyst surface and the NH_3 oxidation toward NO formation can be finely tuned. In our work, NH_3 conversion and product distribution with 10 pulses over 5 cycles for more than 1,300 min of continuous operation is investigated. As can be seen, NH_3 conversion stays essentially constant with no obvious deactivation over the redox cycles, while NO selectivity keeps close to 100% with largely suppressed N_2O formation, demonstrating the excellent recyclability through redox cycling. Per your

suggestion, the NH_3 conversion and product distribution with 30 pulses over 5 cycles is further investigated (Fig. R4). As can be seen, NH_3 conversion remains close to constant over the redox cycles. Despite a higher reduction extend, and thus a decreased NO selectivity with increasing pulse number, a reoxidation of V_2O_5 catalyst fully restores the initial NO selectivity when pulsing NH_3 , indicating the excellent recyclability of the redox cycling.

Fig. R4 NH_3 conversion, NO, N_2O , N_2 selectivity versus the number of pulses over V_2O_5 at 650 °C during 5 consecutive chemical looping NH_3 oxidation redox cycles.

The 30 pulses was not a suggestion but was written in the experimental part in the original manuscript. The main text still only shows the initial 9 cycles. To restate my point – why does the main text only show the first 9 pulses?

7. COMMENT: The physical characterisation is adequate and appear self-consistent, however in this work the XRD seems the most useful as it is able to identify the phases present. With the characterisation that is present can the authors present the

degree of reduction through each technique (XRD, XPS, XANES) and comment on this comparison. For the XANES a LCF of reference standard would help.

RESPONSE: Thank you very much for this point. The phases (V_2O_5 , VO_2 , V_6O_{13}) percentage of V_2O_5 at different NH_3 pulse numbers obtained from XRD Rietveld refinement is presented in Table R1. As can be seen, the V_2O_5 percentage increases with increasing NH_3 pulse numbers. The reduction degree of V_2O_5 determined by each technique (XRD, XPS, XANES) was compared in Table S3 in supporting information. The mean oxidation state of vanadium and the percentage of V^{4+} and V^{5+} determined by XRD and XANES (obtained from comparison of the main edge energies and XANES linear combination fitting) are in qualitative agreement with each other. The mean vanadium oxidation state is slightly higher than that calculated from the overall chemical composition of V_2O_5 at different NH_3 pulse numbers (Fig. S4), which can be attributed to the post-reaction air exposure. On the other hand, XPS results give a higher V^{4+} portion (a higher reduction extent) at different NH_3 pulse numbers, which is due to the surface sensitivity of the XPS measurement. Overall, the XRD, XPS and XANES techniques indicate a monotonic increase in vanadium reduction extent with increasing NH_3 pulses number, and vanadium undergoes reversible reduction and oxidation featuring a V^{5+}/V^{4+} redox cycle in the CLAO process.

Table R1. The phases (V_2O_5 , VO_2 , V_6O_{13}) percentage of V_2O_5 redox catalyst at different NH_3 pulse numbers obtained from XRD Rietveld refinement

Pulse number	V_2O_5 (%)	VO_2 (%)	V_6O_{13} (%)
30	93.0	7.0	-

60	86.9	13.1	1.0
90	79.0	11.0	10.0
120	61.0	14.0	25.0
150	56.0	22.0	22.0
180	52.0	30.0	18.0
Cycled	100	0	0

Can the authors include a discussion on this in the main text and not just in the rebuttal. I think table S3 is really nice as it provides a more complete picture of the redox process.

8. COMMENT: If XAS was measured can the EXAFS also be presented and discussed.

RESPONSE: Thank you for this point. Fourier transforms of V *k*-edge EXAFS for V₂O₅ catalysts as a function of NH₃ pulse numbers are shown in Fig. R5. The peak at 1.50 Å can be attributed to the V-O coordination in the first shell. There is a consistent increase of the amplitude of the V-O coordination sphere with increasing NH₃ pulse numbers, indicating the gradual increase of average coordination number of V-O units with enhanced reduction extent. Meanwhile, as compare to the initial sample, the averaged bond length for V₂O₅ catalysts after 90 and 180 pulses of NH₃ reduction increased from 1.80 Å to 1.92 Å (Table R2). The results are in quality agreement with the XRD observation where V₂O₅ (coordination number, 5; bond distance, 1.585-2.021 Å) gradually transformed into V₂O₄ (coordination number, 5; bond distance, 1.76-2.05 Å) and V₆O₁₃ (coordination number, 5; ~2.0 Å) with a higher V-O coordination number and averaged bond length upon NH₃ pulse reduction.

Fig. R5. Fourier transform (FT) of k-weighted V EXAFS spectra of V_2O_5 catalysts as a function of NH_3 pulse numbers.

Table R2. EXAFS fitting parameters at the V K-edge ($S_0^2=0.72$)

Sample	Path	R (Å) ^a	$\sigma^2 \times 10^3$ (Å ²) ^b	ΔE (eV) ^c	R factor
V_2O_5	V-O	1.80±0.05	13.2±10.6	6.8±5.4	0.018
V-90	V-O	1.92±0.04	13.4±6.6	15.0±3.5	0.019
V-180	V-O	1.92±0.02	10.7±3.8	13.8±2.3	0.013

^aR: bond distance; ^b σ^2 : Debye-Waller factors; ^c ΔE : the inner potential correction. R factor: goodness of fit.

A 1 path fit for each system is not appropriate. Again this is not just for the rebuttal but also in the published work (main text or ESI). Once a more complete fit is performed the discussion should be updated.

9. COMMENT: For the XANES discussion the authors comment that the pre-edge feature can change in intensity based on the symmetry of the V centres and the number of V=O units. It seems the group use the same change in the data to

potentially support different ideas. There authors need to make clear what they think the major influence is here.

RESPONSE: Thank you for this point. As shown in Fig. S10, the V is fivefold coordinated in a distorted tetragonal pyramid of oxygen in V_2O_5 . The apex-oxygen (V=O) distance is only 1.585 Å, whereas the basal V-O distances vary from 1.779 to 2.017 Å²⁷. The distinct pre-peak in the XANES spectrum of V_2O_5 originates from the transition of a 1s electron to an empty antibonding state consisting of O 2p of the nearest oxygen and the V 3d orbitals^{28,29}. The characteristic pre-peak of the V K-edge XANES spectrum of V_2O_5 is concurrently caused by the breaking of the central symmetry and by the presence of the short V=O bond (1.585 Å). Density functional theory calculations also assign the pre-edge feature to the V 3d–O 2p hybridization of the vanadyl bond (V=O) and the local geometry of the distorted $[VO_5]$ square pyramid²⁷. In the case of V_6O_{13} and VO_2 , the V atom are six fold coordinated in a less distorted octahedral $[VO_6]$ groups, with an increased site symmetry and decreased pre-peak intensity compared to V_2O_5 . The intense pre-edge peak of V in perfect octahedral symmetry was absent for VO. The decreased pre-peak intensity can be related to the decreased number of short V=O units as well as an increased site symmetry of the V atom, which is self-consistent.

This is fine. Can this be reflected in the main text.

10. COMMENT: For oxidation state changes in XANES analysis the edge position is not always reliable. Considering the size of the particles I think a XANES LCF with

reference standards found in XRD would be more useful.

RESPONSE: Thank you very much for this point. There are several methods to determine the vanadium oxidation state and symmetry from XANES features including (1) comparison of the main edge energies; and (2) linear combination fitting.³⁰ In our study, comparison of the main edge energies (the energy measured half way up the normalized-edge step) was employed to get a quantitative determination of the average valence state of vanadium. Per your suggestion, a quantitative linear combination fit (LCF) was also performed to show the trend in the evolution of the average valence state of vanadium during the NH₃ pulse reduction process. Because V₆O₁₃ can be written as V₂O₅·2V₂O₄ and the vanadium transitions directly between two states (V⁵⁺ and V⁴⁺), the ratio of V⁵⁺ to V⁴⁺ present in each sample could be determined using V₂O₅ and V₂O₄ as the end members in a linear combination fit. As shown in Fig. R6a-e, for each spectrum the combination of standards with the lowest residual parameter was chosen (the LCF result of V₂O₅ after 30 pulses of NH₃ reduction is not shown due to the low reduction extent and relatively low signal-to-noise ratio). Fig. R7 presents the LCF results as well as the average valence state of vanadium with increasing NH₃ pulse numbers. LCF results indicate that the amount of V⁵⁺ decreases with simultaneous increase of the amount of V⁴⁺ with increasing NH₃ pulse numbers. The percentage of V⁵⁺ decreases from 86.6% to 50.4% as the NH₃ pulse numbers increase from 60 to 180, pointing to the gradual reduction of V⁵⁺ with NH₃. The average vanadium oxidation state decreases from 4.87 to 4.50, which is close to the average vanadium oxidation determined from the

comparison of the main edge energies in Fig. 3d. The mean vanadium oxidation state is slightly higher than that calculated from the overall chemical composition of V_2O_5 at different NH_3 pulse numbers (Fig. S4), which can be attributed to the post-reaction air exposure. We have added the XANES LCF results in Fig.S9 and Table S3.

Fig. R6. Normalized V-K edge XANES spectra as a function of NH_3 pulse numbers, (a) 60 pulses of NH_3 reduction, (b) 90 pulses of NH_3 reduction, (c) 120 pulses of NH_3 reduction, (d) 150 pulses of NH_3 reduction, (e) 180 pulses of NH_3 reduction, and the resultant linear combination fit with the LCF corrected reference spectra and the corresponding residuals.

This is fine.

Fig. R7. Oxidation state of vanadium with increasing NH₃ pulse numbers: (a) Average valence state vanadium. (b)

LCF results of XANES spectra using V₂O₅ and V₂O₄ as reference materials.

11. COMMENT: For figure 4 it would be better not to include the lines of fits (especially where the points are joined). Also the data quality is very noisy and should be commented on.

RESPONSE: Thank a lot for this suggestion. We eliminated the lines of fit for Figure 4 a, b according to your suggestion. Unfortunately, the resolution does not improve due to the overlap of the points (Fig. R8 a, b). Based on the third review's advice, we now add the *In situ* 2d Raman spectra of V₂O₅ redox catalyst during the reduction and subsequent re-oxidation as a function of time in Figure 4 a and b. Meanwhile, a clearer version of the *In situ* 3d Raman spectra of V₂O₅ redox catalyst highlighting the evolution of V=O during the reduction and subsequent re-oxidation as a function of time is provided in Fig S11 b and c.

In our work, the *In situ* Raman spectra of V₂O₅ redox catalyst was acquired at 650 °C. The characterization of the Raman signal at elevated temperatures has always been challenging. The ratio I_R(T)/I_R(298) can be expressed as

$$I_R(T)/I_R(298) = [1 + K(T)/S + (K(T)/S(K(T)/S + 2))^{1/2}]^{-1} / [1 + K(298)/S + (K(298)/S(K(298)/S + 2))^{1/2}]^{-1} \quad (5)$$

where I_R(T) and I_R(298) are the intensity of the Raman scattered light at the detection temperature of T and 298 K, respectively.³¹ K is the absorption coefficient of the powder, and S is the scattering coefficient. Precious study indicates that the intensity of the Raman signal of V₂O₅ crystallites decreases with increasing temperature^{31, 32}. This trend is well documented for solids and is usually ascribed to the effect of increased optical absorbance by the oxide³¹. Despite the high temperature, our *In situ* Raman spectra indicates that the V=O lost intensity with increasing reduction time, implying the removal of terminal oxygen. After re-oxidation, the V=O bands were reinstated.

This is fine

Fig. R8 *In situ* Raman spectra of V₂O₅ redox catalyst highlighting the evolution of V=O during the (a) reduction and (b) subsequent re-oxidation as a function of time. (c) The dependence of band intensity for V=O stretching vibrations on temperature determined experimentally and theoretically (eq 7). Data are reproduced from reference [31].

Thank you once again for your valuable comments and suggestions. We worked hard to be responsive to them. We are grateful to you for taking the time and energy to help us improve the paper.

GENERAL COMMENTS TO THE REFEREES AND EDITOR:

We want to extend our appreciation for all the reviewers for their insightful guidance.

We carefully considered the constructive comments offered by the three reviewers and tried our best to revise the paper based on these comments and recommendations. We hope that these revisions improve the paper such that you and the reviewers now deem it worthy of publication. A point-by-point response to each reviews' concerns and comments are provided below.

RESPONSES TO REFEREE #1's COMMENTS:

Reviewer #1 (Remarks to the Author):

The reviewer finds that the authors revised the manuscript and now it is suitable for publication.

Response: Thank you very much. We appreciate you taking the time to offer us your insightful comments and constructive suggestions.

RESPONSES TO REFEREE #2's COMMENTS:

Remarks to the Author

The paper by Ruan et al. describes selective production of NO via a chemical looping experiment (NH₃/Ar and O₂/Ar) on V₂O₅. Overall, the most impressive part of the

manuscript are the results from the chemical looping experiment. The side-by-side comparison of the continual flow experiment and CL is really nice and shows the value of this approach. It is difficult to assess the viability of this as a replacement technology, however, it is still extremely valuable that these results have been established. They will be of value of the field and are able to shape future thinking on how catalytic processes can be performed and optimised. Elsewhere, the manuscript contains physical characterisation (with in situ Raman) and DFT calculations of the reaction pathways. These seem a useful addition, however, it was unclear if they really enhanced the paper significantly beyond the core result in the catalytic testing. I was hoping to find more on the origin of the improved selectivity through CL - perhaps I am too greedy here - but I think it would help to expand on this point to be accepted in Nature Communications. To help improve the manuscript below, a point-by-point list comments are detailed below:

Response: Thank you very much for your kind words about our paper. We appreciate you taking the time to offer us your insightful comments and constructive suggestions. In the following sections, you will find our responses to each of your point and suggestion.

Despite the core role and large utilization of metal oxide catalysts in chemical looping, the lack of a better understanding of the chemical nature of metal oxide catalysts and reaction pathway has limited the establishment of extensive quantitative and universal structure–reactivity–selectivity–stability relationships with the redox metal oxides. Meanwhile, the highly dynamic nature of redox catalysts undergoing

chemical looping reactions also makes mechanistic investigation even more challenging than in heterogeneous catalysis. This, in turn, has delayed the discovery of new catalysts and the rational design of the chemical looping process. In order to gain insights into the dynamic nature of the active sites in metal oxide catalysts, we are coupling operando Raman spectroscopy, XRD, XPS, XANES as well as DFT calculations with catalytic reactivity studies to contribute to the endeavor for a more rational design of metal oxide catalysts. These are also fundamental to extending our understanding of redox metal oxide catalysts for the chemical looping reactions. The above techniques show that the proposed CLAO process operates via a modified, temporally separated MvK redox mechanism featuring a reversible V^{5+}/V^{4+} redox cycle. The vanadyl ($V=O$) centers act as active sites for NH_3 oxidation, leading to the formation of the oxidation products. Meanwhile both $V=O$ and doubly coordinated oxygen (O^{II}) participate in the hydrogen transfer process. The superior activity and selectivity can be attributed to the low activation energies for the successive hydrogen abstraction and facile NO formation facilitated by $V=O$.

Due to the dynamic nature of redox catalysts undergoing chemical looping reactions, the NH_3 conversion and the molar distribution of nitrogen species depend strongly upon the concentration of surface oxygen species ($V=O$), the extent of redox catalyst reduction, the stoichiometric ratios for reducible oxygen in V_2O_5 to NH_3 . The spatially and temporally separated feature of the chemical looping strategy allows for the precise control of the residence time of reactants, the reduction extent of the redox metal oxide and ratios of the reaction intermediates (O/NH_x) on the catalyst surface.

In contrast, the conventional co-feed mode faces the difficulty of maintaining these precise ratios of the reaction intermediates at every point of the catalyst. Additionally, the ammonia oxidation is one of the most investigated reactions in heterogeneous catalysis from mechanistic and kinetic viewpoints. Selectivity in ammonia oxidation via the conventional co-feed process is suggested to be determined by competition between NH_3 and O_2 molecules for the same active sites, and the intermediates' surface concentration. From a reaction chemistry standpoint, the chemical looping approach we proposed here proceeds via a unique, temporally separated MvK mechanism associated with the breaking and regeneration of $\text{V}=\text{O}$ bonds. Under the CLAO reaction scheme, the active $\text{V}=\text{O}$ sites selectively activate and react with ammonia, while being removed from the surface. In a subsequent, temporally separated oxidation step, the active $\text{V}=\text{O}$ sites were restored when reacting with O_2 . The CLAO reaction scheme eliminates the competition between NH_3 and O_2 molecules for same active sites, which can potentially contribute to the superior NH_3 oxidation performance. Therefore, we tentatively attribute high NO selectivity of CLAO to a desirable thermodynamic property of the redox metal oxide, an optimal ratio of reducible oxygen in V_2O_5 to NH_3 (O/NH_x) on the catalyst surface, as well as the elimination of the competition between NH_3 and O_2 molecules for same active sites enabled by the unique periodic feature of the chemical looping strategy. We hope that you agree.

COMMENT: Since the authors have just restated the points in the original paper, I can only restate what I suggested initially – I was hoping for a more precise

explanation on the rationale behind the improvement presented in the work. I did not find the characterization that informative, as I already indicated. One needs to consider the limitations of each technique, including in relation to the size domains/heterogeneity of the sample, and the conditions in which the study has been carried out. I do not feel the authors have fully engaged in these constructive comments. I am a supporter of the work, but do feel that the characterization presented has some limitations. Essentially, whether it is XANES, XRD, or Raman, the data show that under reducing conditions the sample is reduced and that under oxidizing conditions the sample is oxidized – this is hardly a surprise. It does confirm that O is abstracted from the metal oxide, so I accept the MvK inference. However, the emphasis of my question was for the authors to put forward a more precise description of why the chemical looping experiment reduces the amount of over-oxidation products – I did not find this in the original manuscript and this point is still missing as far as I can see.

Response: Thank you very much for your support of our work and your valuable suggestions. We do appreciate you taking the time to offer us your insightful comments and constructive suggestions.

Since the chemical potential of O₂ supplied in the co-feed mode and the chemical looping mode was the same. Meanwhile, the reaction temperature for both modes was also the same; the strength of the bonding of oxygen to the surface should also be comparable. The difference between the co-feed mode and the chemical looping mode was the way that oxygen was delivered. In the co-feed mode, adsorption and

reduction of gaseous oxygen and the oxidation of ammonia take place in the same compartment. In contrast, in the chemical looping mode the activation of oxygen is temporally separated from the catalytic oxidation of ammonia where oxygen is consumed. Hence, the dominating oxygen species of each mode play a significant role in the final product distribution. It has been proposed that the NH_3 can be directly oxidized by lattice oxygen to NO, and a low NO selectivity is attributed to the high concentrations of adsorbed molecular oxygen species that react with adsorbed NH_3 to form N_2 . For example, Cao. et al.¹ demonstrated that using a NH_3/O_2 mixture as co-feed in NH_3 oxidation, adsorbed surface molecular oxygen species led to a lower NO selectivity. The NO selectivity could be improved from 77 to 95% by using an oxygen-permeable membrane to suppress the formation of nonselective surface molecular oxygen species at 850 °C instead of a co-feed fixed bed reactor. Biauxque et al.² reported that one key route for N_2 formation in the oxidation of NH_3 over perovskite LaCoO_3 lay in the reaction of adsorbed ammonia with adsorbed oxygen species such as peroxide (O_2^{2-}) and superoxide (O_2^-) species. In our proposed CLAO process, the catalytically active $\text{V}=\text{O}$ species was found to play a very important role in NO formation. The formation of lattice oxygen (regeneration of $\text{V}=\text{O}$) by reoxidation of the reduced metal oxide with gaseous oxygen is suggested as follows:³

First, oxygen molecules adsorb on the oxide surface, and the adsorbed O_2 is reduced and incorporated into the metal oxide as lattice oxygen $\text{O}_{2(\text{lattice})}^{2-}$. The activation of gaseous oxygen is spatially and temporally separated from NH_3 oxidation to

effectively suppress the formation of nonselective surface molecular oxygen species. Additionally, the chemical looping strategy allows for the precise control of the residence time of reactants, the reduction extent of the redox metal oxide and ratios of the reaction intermediates (O/NH_x) on the catalyst surface. Therefore, a high NH₃ conversion can be achieved, with can potentially suppress the N₂ formation via the catalytic reduction of NO with NH₃. Consequently, a high selectivity for NO is achieved for our proposed CLAO process. We agree with you the limitation of the each characterization technique in the present study. A better understanding of the highly dynamic nature of metal oxide catalysts in CL, the active species and the difference reaction pathway of the CL and co-feeding modes is still necessary for the establishment of the structure–activity relationships. There are the focus of our ongoing research and the challenge facing the chemical looping research community. We hope our current research can provide some fundamental basis to design and optimize a completely different approach for NO synthesis and some catalytic reactions based on chemical looping strategy.

We have now added a precise explanation on the rationale behind the improvement of NO selectivity through CL in the reaction mechanism part in the main text. “This temporally separated MvK scheme avoids direct competition of reduction and reoxidation on the same V=O active sites that would otherwise be inevitable in a co-feed scheme, leading to remarkable enhancement in catalyst performance. Furthermore, in the proposed CLAO process, the activation of oxygen is spatially and temporally separated from the catalytic NH₃ oxidation to effectively

suppress the formation of nonselective surface molecular oxygen species and thus the undesired nitrogen byproduct^{1, 2}. Additionally, the chemical looping strategy allows for the precise control of the residence time of reactants, the reduction extent of the redox metal oxide and ratios of the reaction intermediates (O/NH_x) on the catalyst surface. Therefore, a high NH₃ conversion can be achieved, with can potentially suppress the N₂ formation via the catalytic reduction of NO with NH₃. Consequently, a high selectivity for NO can be anticipated.”

1. COMMENT: In Fig. 1, the schematic says O₂ depleted air. However, the O₂ source is 30% O₂ in Ar.

RESPONSE: Thank you for this point. Figure 1 is a representation of the chemical looping ammonia oxidation (CLAO) concept. During the reduction step, NH₃ is oxidized by the active lattice oxygen derived from the metal oxide redox catalyst, producing concentrated NO. In the subsequent oxidation step, the reduced metal oxide redox catalyst is re-oxidized with air, completing the redox cycle. From the potential application standpoint, Air instead of O₂ is used, which can make the system more cost-effective. To prove our proposed concept, 30% O₂/Ar rather than Air is applied for the reoxidation step in our experiment, which can eliminate the interference of N₂ and CO₂ (amu 44, the mass overlaps with N₂O), allowing for an accurate determination of the NH₃ oxidation products. Meanwhile, for chemical looping research community, air and O₂ depleted air are commonly used as regeneration agent and depleted regeneration outlet⁴⁻⁸. For the sake of consistency, Air instead of

30% O₂/Ar is employed for schematic description of the CLAO process. We hope that you agree.

COMMENT: No – I do not agree. If you have used a certain amount of oxygen the captions used should reflect that. Anything else is disingenuous to the readership and poor scientific practice. This is a rather simple point and I do not understand the reluctance of the authors here. Ultimately, it just improves the clarity of the manuscript.

Response: Thank you once again for your valuable suggestion. We have now included a corrected version of Fig. 1 (Fig. R1) with the O₂ source of 30% O₂/Ar in the main text.

Fig. R2 Schematic of the chemical looping ammonia oxidation (CLAO) process. The red and gray spheres represent metal and oxygen atoms, respectively.

2. COMMENT: Can the authors comment on the productivity per unit time as well as selectivity - this is helpful to understand the comparison between flow and CL.

RESPONSE: Thank you for this point. Kinetically, chemical looping reactions are generally slower than conventional catalytic oxidation/reduction reactions, which are commonly encountered in chemical looping process. This is due to the fact that the rates of redox reactions are limited by the removal and replenishment of lattice oxygen from/to the bulk of the redox catalyst. The CLAO process proposed here resulted in a normalized NO production rate of $\sim 0.94 \text{ ml g}_{\text{catalyst}}^{-1} \text{ min}^{-1}$ with superior NH_3 (97.0%) conversion and NO selectivity (99.8%) at 650 °C. The NO production rate of conventional co-feeding mode reported here is $2.13 \text{ ml g}_{\text{catalyst}}^{-1} \text{ min}^{-1}$ with inferior NH_3 (87.8%) conversion and NO selectivity (22.6%) at 650 °C. While the NO production rates are on the same order magnitude; however, the CLAO approach eliminates the costly separation units, ensures safer operation and inhibits non-selective oxidation reactions, which can contribute to the economic attractiveness of the NH_3 oxidation process.

COMMENT: The authors need to include a discussion of these production rates in the main text, I could not find this in the revision. Apologies if I have missed this. Rather than restating conversion/selectivity values, the authors should comment on the relative usefulness of the values and how the chemical looping value could be further improved.

Response: Thank you very much for this valuable suggestion. We have now included a discussion regarding the comparison of the production rates in the main text: “Kinetically, chemical looping reactions are generally slower than conventional catalytic oxidation/reduction reactions, which are commonly encountered in chemical

looping process. This is because the rates of redox reactions are limited by the removal and replenishment of lattice oxygen from/to the bulk of the redox catalyst. The CLAO process proposed here resulted in a normalized NO production rate of $\sim 56.4 \text{ ml g}_{\text{catalyst}}^{-1} \text{ min}^{-1}$ (Table S1). The NO production rate of conventional co-feeding mode reported here ranged between $105.2\text{-}127.6 \text{ ml g}_{\text{catalyst}}^{-1} \text{ h}^{-1}$ (Table S1). While the NO production rates are on the same order of magnitude, the superior NH_3 conversion and NO selectivity of CLAO approach can potentially eliminate the costly separation units, ensure safer operation and inhibit non-selective oxidation reactions, which contribute to the economic attractiveness of the NH_3 oxidation process. Furthermore, to ensure a relatively faster surface oxygen removal rate, two potential strategies have been proposed for the chemical looping process.⁹ One is through doping or lattice site substitution, in an attempt to introduce crystallographic defects that facilitate oxygen ion diffusion. The second strategy is to promote the redox metal oxide with catalytically active species on the surface to decrease the activation energy for reactant activation and to enhance the lattice oxygen consumption from the bulk. Both strategies may open up extensive possibilities for the CLAO process.”

3. COMMENT: Figure 2a and 2b nicely show the catalytic testing data. One of the noteworthy observations is that for the flow experiments, the N_2 selectivity decreases, before rising for the final temperature. In the discussion on the reaction pathways the only reaction route considered is NH_3 oxidation. However, there is a recent study that demonstrates that the NO_x produced through NH_3 oxidation can also create the

conditions for a parallel SCR mechanism (<https://doi.org/10.1021/acscatal.0c05356>).

Considering the increase of N₂ at high temperature is this something the authors have considered?

RESPONSE: Thank you for this point. Yes, we have considered. The selective catalytic reduction (SCR) of NO_x (NO and NO₂) with NH₃ to N₂ and H₂O proceeds via the overall reactions

For the co-feed experiment as shown in Fig. R2, the N₂ selectivity decreases from 300-600 °C and increases at higher temperatures of 650 and 670 °C. If the increase of N₂ selectivity originates from selective catalytic reduction of NO_x, both an increase in NH₃ conversion and a decrease in NO_x selectivity would occur according to equation (1) and (2). While the activity test shown in Fig. R2 indicates that the NH₃ conversion plateaus at 500-650 °C. Hence, we tentatively attribute the increase of N₂ selectivity at higher temperatures to the decomposition of NO_x, which has been reported elsewhere^{2, 10-12}.

In our study, we mainly focused on validating the CLAO process via experimental and DFT investigation. The reaction pathways of the proposed CLAO process is illustrated in Fig. 6 in the main manuscript. In addition, the interaction of NH₃ and NO to produce N₂O has been investigated (Fig. 6c).

Fig. R2 NH₃ conversion and NO, N₂O, N₂ selectivity versus temperature over V₂O₅ in steady state NH₃ oxidation (co-feed mode).

COMMENT: I can see the authors have engaged with this comment, which is welcome, but I am not sure I necessarily agree. The NH₃ conversion plateaus from ~ 500 °C. However, the selectivity profile continues to evolve. For example, at 550 °C there is more NO and less N₂ produced than at 500 °C. It does not necessarily follow, that a change in selectivity profile requires the conversion of products produced at the previous temperature, e.g. in this case it is likely that NH₃ is converted into different products. The same could also be true at higher temperature – you don't need to see an increase in NH₃ conversion for there to be the possibility of an SCR type process. Of course, this could be easily resolved with spatially resolved (rather than end pipe) gas analysis, however, this is not within the scope of the work and I accept that. However, it would improve the manuscript to add a discussion about the changing selectivity profile, which I do not find in the main text.

Response: Thank you very much for this valuable suggestion. We have now added the following discussion regarding the changing selectivity profile in the main text: “The decreased NO formation with a concurrent increase in N₂ formation at higher temperatures can be related to the selective catalytic reduction (SCR) of NO with NH₃. A similar parallel SCR mechanism was also proposed in a recent study of Decarolis et al.,^[ref] which indicates the oxidation of NH₃ to NO_x by PdO and the subsequent catalytic reduction of NO_x by NH₃ to produce N₂.”

Ref: Decarolis, D. et. al. Spatial Profiling of a Pd/Al₂O₃ Catalyst during Selective Ammonia Oxidation. ACS Catal. 2021, 11, 2141–2149.

4. COMMENT: Following on from point 3, clearly there are structural changes to the catalyst and compositional changes of the reactant environment. Are the authors able to comment on the impact of these with respect to the catalytic properties?

RESPONSE: Thank you for this point. For co-feed NH₃ oxidation, no apparent structural change was observed for V₂O₅ redox catalyst after the reactivity test (Fig. S5). For CLAO, powder X-ray diffraction (XRD) indicates that upon NH₃ pulse reduction, new reflections associated with V₆O₁₃ and VO₂, respectively, are identified, which progressively increased in intensity with increasing NH₃ pulse number as more lattice oxygen was extracted. XPS indicates vanadium existed in the form of V⁵⁺ and V⁴⁺ after reduction with NH₃. XANES and Raman spectroscopy results indicate that the reaction proceeds with the breaking and regeneration of V=O bonds.

In order to derive mechanistic insight into the effect of lattice oxygen extraction

on the NH₃ conversion and product distribution over V₂O₅ in the CLAO process, the NH₃ conversion, NO, N₂, N₂O selectivity as well as the accumulated lattice oxygen withdrawn from the original sample (δ) is plotted as a function of the NH₃ pulse numbers (Fig. 2d and Fig. S4). It is revealed that the selectivity of NH₃ oxidation toward NO formation is favored when the concentration of surface oxygen species (V=O) is sufficiently high for stripping all H atoms from NH₃ and for further oxidation of the resulting NH_x ($x < 3$) intermediates to NO [Eq.(5)]. The gas-phase ammonia adsorb over an active site is denoted as “s”.

The concentration of surface oxygen species decreases with increasing NH₃ pulse number due to its removal for ammonia oxidation to NO and H₂O. When the concentration of surface oxygen species (V=O) decreases, the reduced catalyst surface contains mainly strongly bound lattice oxygen species and anion vacancies in the oxide lattice, which can effectively activate the N–H bond in ammonia, dehydrogenate ammonia to form NH_x intermediates. In addition, the relative concentration of surface NH_x species should be higher. Consequently, recombination of two highly reactive NH_x fragments will prevail, leading to N₂ formation [Eq. (6)], with a decreased NO selectivity as well as insignificant amounts of N₂O (Fig. S4).

Afterward a re-oxidation of the catalyst resulted in an oxygen uptake, regenerating surface oxygen species (V=O) and restoring the initial NO selectivity when pulsing NH₃.

COMMENT: My comment was more focused on localized variations to the catalyst structure at different points of the reaction profile. I do not feel this is an essential point for the manuscript, but it is something the authors may wish to consider as they progress their work.

Response: Thank you very much for your valuable suggestions. According to your constructive suggestion, in our future study, we will further investigate the spatial profiling of the V₂O₅ catalyst during CLAO oxidation combined with operando spectrometry along the catalyst bed. The changes of reactant environmental may affect the form of the catalyst, which necessitates the assessment the structure of the catalyst across the fixed bed reactor. The spatial analysis inspired by this excellent work^[ref] may provide further insights into the structure–activity relationships governing the selective NH₃ oxidation.

Ref: Decarolis, D. et. al. Spatial Profiling of a Pd/Al₂O₃ Catalyst during Selective Ammonia Oxidation. ACS Catal. 2021, 11, 2141–2149.

5. COMMENT: At 300 °C there is 100% selectivity to N₂ for the CL experiment. What is the fate of the H₂. It is not seen in the MS. If it generates surface hydroxyls was this identified through XPS/Raman. Clearly, the focus of this paper is NH₃ oxidation, however, perhaps this methodology has translational impact to NH₃ decomposition and this could be an important point.

RESPONSE: Thank you for this point. Fig. 2a illustrates the ammonia conversion and product distribution over V₂O₅ as a function of temperature in typical chemical

looping mode. The corresponding products evolution rates upon NH_3 injection are reproduced in Fig. S2a-h. While no H_2 was observed in the reaction temperature range, with H_2O being the only H-containing product. The surface oxygen species ($\text{V}=\text{O}$) catalyze NH_3 dehydrogenation, stripping all H atoms and for further oxidation of the resulting NH_x ($x < 3$) intermediates to N_2 and H_2O . To allow for an accurate determination of the outlet NH_3 (amu 17), H_2O (amu 17, 18) was condensed out to avoid interference before entering the quadrupole spectrometer.

Meanwhile, by decreasing the V_2O_5 loaded (0.05 g) for CLAO, H_2 signal emerged as there are less surface oxygen species available for NH_x ($x < 3$) oxidation (Fig. R3a). Additionally, we also measured the O 1s XPS spectra V_2O_5 sample after the CLAO experiment at 300 °C. As shown in Fig. R3b, the hydroxyl group centered at around 532 eV was identified.

Fig. R3 (a) Temporal gaseous product concentrations for V_2O_5 upon NH_3 injection at 630 °C in chemical looping NH_3 oxidation process. (b) O 1s XPS spectra of the V_2O_5 sample after the CLAO experiment at 300 °C.

COMMENT: This does not answer the question. What proportion of H ends up in H₂O, surface hydroxyls, or H₂? Can the authors include a discussion of this also in the main text.

Response: Thank you very much for this point. No H₂ was observed in the reaction temperature range (300-650 °C). All the H ends up into H₂O, which was condensed out before entering the quadrupole spectrometer. This is explained in the experimental part with the following description: “Outlet gas concentrations were monitored online after condensation of moisture by a quadrupole spectrometer (Pfeiffer Omnistar QMS 200), using Ar as internal standard.” And also in the Results and Discussion part: “The corresponding products evolution upon NH₃ injection are reproduced in Fig. S2a-h, which indicates that NO, N₂O and N₂ were the only N-containing products. While no H₂ was observed in the reaction temperature range, with water being the only H-containing product that has been condensed out to avoid interference.”

Per your suggestion, we also add a following discussion of it in the main text for further clarify. “To elucidate the dynamics of surface lattice oxygen removal during the reduction step, the initial maximum lattice oxygen extraction rate for NH₃ oxidation during the first pulse was determined from Fig. S2a-h, considering the following stoichiometry reaction pathways as no H₂ is produced (all the H ends up into H₂O, which was condensed out before entering the quadrupole spectrometer)”

6. COMMENT: In figure 2e, why are only 9 injections shown for each repeat and not the full 30?

RESPONSE: Thank you very much for this point. The results in Fig. 2d and Fig. S4 reveals that the selectivity of NH_3 oxidation toward NO formation is determined by the degree of oxidation of the V_2O_5 catalyst surface or the concentration of surface oxygen species ($\text{V}=\text{O}$). By controlling the NH_3 pulses numbers, the reduction degree of V_2O_5 catalyst surface and the NH_3 oxidation toward NO formation can be finely tuned. In our work, NH_3 conversion and product distribution with 10 pulses over 5 cycles for more than 1,300 min of continuous operation is investigated. As can be seen, NH_3 conversion stays essentially constant with no obvious deactivation over the redox cycles, while NO selectivity keeps close to 100% with largely suppressed N_2O formation, demonstrating the excellent recyclability through redox cycling. Per your suggestion, the NH_3 conversion and product distribution with 30 pulses over 5 cycles is further investigated (Fig. R4). As can be seen, NH_3 conversion remains close to constant over the redox cycles. Despite a higher reduction extend, and thus a decreased NO selectivity with increasing pulse number, a reoxidation of V_2O_5 catalyst fully restores the initial NO selectivity when pulsing NH_3 , indicating the excellent recyclability of the redox cycling.

Fig. R4 NH₃ conversion, NO, N₂O, N₂ selectivity versus the number of pulses over V₂O₅ at 650 °C during 5 consecutive chemical looping NH₃ oxidation redox cycles.

COMMENT: The 30 pulses was not a suggestion but was written in the experimental part in the original manuscript. The main text still only shows the initial 9 cycles. To restate my point – why does the main text only show the first 9 pulses?

Response: Thank you very much for this point. We now add 30 pulses of recyclability results in Fig. 2e to exemplify the NH₃ conversion and product distribution for the redox cycling with the following discussion on this in the main text:

“Fig. 2e exemplifies NH₃ conversion and product distribution over 5 cycles for more than 1,300 min of continuous operation. As can be seen, NH₃ conversion remains close to constant over the redox cycles. Despite the decreased in NO selectivity with increasing pulse number, a reoxidation of V₂O₅ catalyst fully restores the initial NO selectivity when pulsing NH₃, indicating the excellent recyclability through redox cycling.”

Figure 2d highlights the NH₃ conversion, NO, N₂, N₂O selectivity in the first 10 pulses in the CLAO process. For the sake of consistency, the authors suggested it's more logical to exemplify the NH₃ conversion and product distribution with the initial 10 cycles. We apologized that we misinterpreted your suggestion. We have now added 30 pulses of recyclability results in Fig. 2e.

7. COMMENT: The physical characterisation is adequate and appear self-consistent, however in this work the XRD seems the most useful as it is able to identify the

phases present. With the characterisation that is present can the authors present the degree of reduction through each technique (XRD, XPS, XANES) and comment on this comparison. For the XANES a LCF of reference standard would help.

RESPONSE: Thank you very much for this point. The phases (V_2O_5 , VO_2 , V_6O_{13}) percentage of V_2O_5 at different NH_3 pulse numbers obtained from XRD Rietveld refinement is presented in Table R1. As can be seen, the V_2O_5 percentage increases with increasing NH_3 pulse numbers. The reduction degree of V_2O_5 determined by each technique (XRD, XPS, XANES) was compared in Table S3 in supporting information. The mean oxidation state of vanadium and the percentage of V^{4+} and V^{5+} determined by XRD and XANES (obtained from comparison of the main edge energies and XANES linear combination fitting) are in qualitative agreement with each other. The mean vanadium oxidation state is slightly higher than that calculated from the overall chemical composition of V_2O_5 at different NH_3 pulse numbers (Fig. S4), which can be attributed to the post-reaction air exposure. On the other hand, XPS results give a higher V^{4+} portion (a higher reduction extent) at different NH_3 pulse numbers, which is due to the surface sensitivity of the XPS measurement. Overall, the XRD, XPS and XANES techniques indicate a monotonic increase in vanadium reduction extent with increasing NH_3 pulses number, and vanadium undergoes reversible reduction and oxidation featuring a V^{5+}/V^{4+} redox cycle in the CLAO process.

Table R1. The phases (V_2O_5 , VO_2 , V_6O_{13}) percentage of V_2O_5 redox catalyst at different NH_3 pulse numbers obtained from XRD Rietveld refinement

Pulse number	V ₂ O ₅ (%)	VO ₂ (%)	V ₆ O ₁₃ (%)
30	93.0	7.0	-
60	86.9	13.1	1.0
90	79.0	11.0	10.0
120	61.0	14.0	25.0
150	56.0	22.0	22.0
180	52.0	30.0	18.0
Cycled	100	0	0

COMMENT: Can the authors include a discussion on this in the main text and not just in the rebuttal. I think table S3 is really nice as it provides a more complete picture of the redox process.

Response: Thank you very much for this point. According to your suggestion, we have now added the following discussion on this in the main text:

“The reduction degree of V₂O₅ determined by XRD, XPS and XANES was compared in Table S3. The mean oxidation state of vanadium and the percentage of V⁴⁺ and V⁵⁺ determined by XRD and XANES (obtained by comparison of the main edge energies and XANES linear combination fitting) are in qualitative agreement with each other. The mean vanadium oxidation state is slightly higher than the average value of +4.4 of a global composition of V₂O_{5-0.6} presented in Fig. S4. The small difference is most likely induced by post-reaction air exposure due to the ex situ XRD and XNAES measurement. Meanwhile, XPS results give a higher V⁴⁺ portion (a higher reduction extent) at different NH₃ pulse numbers, which is due to the surface sensitivity of the XPS measurement. Overall, the XRD, XPS and XANES techniques indicate a monotonic increase in vanadium reduction extent with increasing NH₃ pulses number.”

8. COMMENT: If XAS was measured can the EXAFS also be presented and discussed.

RESPONSE: Thank you for this point. Fourier transforms of V *k*-edge EXAFS for V₂O₅ catalysts as a function of NH₃ pulse numbers are shown in Fig. R5. The peak at 1.50 Å can be attributed to the V-O coordination in the first shell. There is a consistent increase of the amplitude of the V-O coordination sphere with increasing NH₃ pulse numbers, indicating the gradual increase of average coordination number of V-O units with enhanced reduction extent. Meanwhile, as compare to the initial sample, the averaged bond length for V₂O₅ catalyst after 90 and 180 pulses of NH₃ reduction increased from 1.80 Å to 1.92 Å (Table R2). These results are in quality agreement with the XRD observation where V₂O₅ (coordination number, 5; bond distance, 1.585-2.021 Å) gradually transformed into V₂O₄ (coordination number, 6; bond distance, 1.76-2.05 Å) and V₆O₁₃ (coordination number, 6; ~2.0 Å) with a higher V-O coordination number and averaged V-O and V-V bond lengths upon NH₃ pulse reduction.

Fig. R5. Fourier transform (FT) of k-weighted V EXAFS spectra of V_2O_5 catalysts as a function of NH_3 pulse numbers.

Table R2. EXAFS fitting parameters at the V K-edge ($S_0^2=0.72$)

Sample	Path	R (Å) ^a	$\sigma^2 \times 10^3$ (Å ²) ^b	ΔE (eV) ^c	R factor
V_2O_5	V-O	1.80±0.05	13.2±10.6	6.8±5.4	0.018
V-90	V-O	1.92±0.04	13.4±6.6	15.0±3.5	0.019
V-180	V-O	1.92±0.02	10.7±3.8	13.8±2.3	0.013

^aR: bond distance; ^b σ^2 : Debye-Waller factors; ^c ΔE : the inner potential correction. R factor: goodness of fit.

COMMENT: A 1 path fit for each system is not appropriate. Again this is not just for the rebuttal but also in the published work (main text or ESI). Once a more complete fit is performed the discussion should be updated.

Response: Thank you very much for this point. Based on your suggestion, we have performed a more complete fitting and with the following discussion added in the ESI.

“Fourier transforms of V K-edge EXAFS for V_2O_5 catalysts as a function of NH_3 pulse

numbers are shown in Fig. R2. The first two peaks at 1.50 Å and 2.85 Å are dominated by single scattered contributions from the first and second coordination spheres of V-O and V-V correlations, respectively. The peaks above 3 Å are complicated by the contributions from a large number of single as well as multiple-scattered contributions from other paths. Hence, only the first and two peaks are discussed here. There is a consistent increase of the amplitude of the V-O coordination sphere with increasing NH₃ pulse numbers, indicating the gradual increase of average coordination number of V-O units with enhanced reduction extent. In order to gain quantitative information, fitting of the EXAFS data were performed for initial V₂O₅ and V₂O₅ after 90 and 180 pulses of NH₃ reduction. The results of fitting are presented in Fig. R3, with the best fit parameters summarized in Table R1. As compare to the initial sample, the averaged V-O distance for V₂O₅ catalysts after 90 and 180 pulses of NH₃ reduction increased from 1.80 Å to 1.92 Å. On the contrary, a decrease in the distance of the second coordination sphere of V-V is observed for V₂O₅ catalysts after 90 and 180 pulses of NH₃ reduction. The results are in quality agreement with the XRD observation where V₂O₅ (coordination number, 5; average V-O bond distance, 1.825 Å; average V-V bond distance, 3.564 Å) gradually transformed into V₂O₄ (coordination number, 6; average V-O bond distance, 1.939 Å; average V-V bond distance, 2.892 Å) and V₆O₁₃ (coordination number, 6; average V-O bond distance, 1.949 Å; average V-V bond distance, 3.063 Å) with a higher V-O coordination number and averaged V-O distance as well as a lower averaged V-V distance upon NH₃ pulse reduction.”

Fig. R2. Fourier transform (FT) of k-weighted V EXAFS spectra of V_2O_5 catalysts as a function of NH_3 pulse numbers.

Fig. R3. $k^3\chi(k)$ spectra and Fourier transforms of $k^3\chi(k)$ spectra at the V K-edge of V_2O_5 (a and b), V_2O_5 subjected to 90 pulses of NH_3 reduction (c and d) and V_2O_5 subjected to 180 pulses of NH_3 reduction (e and f) from experiment (—) and fit results (···).

Table R1. EXAFS fitting parameters at the V K-edge for V_2O_5 and V_2O_5 subjected to 90 pulses (V-90) and 180 pulses (V-180) of NH_3 reduction ($S_0^2=0.72$)

Sample	Path	R (Å)	$\sigma^2 \times 10^3$ (Å ²)	ΔE (eV)	R factor
V_2O_5	V-O	1.80±0.05	13.2±10.6	6.8±5.4	0.016
	V-V	3.35±0.06	6.6±9.4	9.8±3.1	
V-90	V-O	1.92±0.04	13.4±6.6	15.0±3.5	0.014
	V-V	3.13±0.04	2.7±6.5	8.8±2.5	
V-180	V-O	1.92±0.02	10.7±3.8	13.8±2.3	0.015
	V-V	3.12±0.05	3.4±8.5	10.0±3.1	

^a*C.N.*: coordination numbers; ^b*R*: bond distance; ^c σ^2 : Debye-Waller factors; ^d ΔE : the inner potential correction. *R* factor: goodness of fit.

9. COMMENT: For the XANES discussion the authors comment that the pre-edge feature can change in intensity based on the symmetry of the V centres and the number of V=O units. It seems the group use the same change in the data to potentially support different ideas. There authors need to make clear what they think the major influence is here.

RESPONSE: Thank you for this point. As shown in Fig. S10, the V is fivefold coordinated in a distorted tetragonal pyramid of oxygen in V_2O_5 . The apex-oxygen (V=O) distance is only 1.585 Å, whereas the basal V-O distances vary from 1.779 to 2.017 Å¹³. The distinct pre-peak in the XANES spectrum of V_2O_5 originates from the

transition of a 1s electron to an empty antibonding state consisting of O 2p of the nearest oxygen and the V 3d orbitals^{14, 15}. The characteristic pre-peak of the V K-edge XANES spectrum of V₂O₅ is concurrently caused by the breaking of the central symmetry and by the presence of the short V=O bond (1.585 Å). Density functional theory calculations also assign the pre-edge feature to the V 3d–O 2p hybridization of the vanadyl bond (V=O) and the local geometry of the distorted [VO₅] square pyramid¹³. In the case of V₆O₁₃ and VO₂, the V atom are six fold coordinated in a less distorted octahedral [VO₆] groups, with an increased site symmetry and decreased pre-peak intensity compared to V₂O₅. The intense pre-edge peak of V in perfect octahedral symmetry was absent for VO. The decreased pre-peak intensity can be related to the decreased number of short V=O units as well as an increased site symmetry of the V atom, which is self-consistent.

COMMENT: This is fine. Can this be reflected in the main text.

Response: Thank you very much for this point. Per your suggestion, we have now added the following description to the main text for clarity:

“The characteristic pre-peak of the V K-edge XANES spectrum of V₂O₅ is concurrently related to the breaking of the central symmetry and the presence of the short V=O bond (1.585 Å). Density functional theory calculations also assign the pre-edge feature to the V 3d–O 2p hybridization of the vanadyl bond (V=O) and the local geometry of the distorted [VO₅] square pyramid.”

Thank you once again for your valuable comments and suggestions. We worked hard

to be responsive to them. We are grateful to you for taking the time and energy to help us improve the paper.

RESPONSES TO REFEREE #3's COMMENTS:

Reviewer #3 (Remarks to the Author):

The authors have significantly improved the manuscript after incorporating all the reviewers' comments. The editor may accept the manuscript after the following comments are addressed.

Response: Thank you very much for your kind words about our paper. We appreciate you taking the time to offer us your insightful comments and constructive suggestions.

1. The methods used for entropy calculations in Gibbs free energy barriers are recommended to be included in the methods section.

Response: Thank you very much for this point. We have added the following descriptions in the Computational Details:

“The Gibbs free energy corrections were calculated using VASPKIT,¹⁶ which includes the zero-point energy (ZPE), enthalpy, and entropy contributions. For surface adsorbates, free energies were approximated using the harmonic approximation that treats all degrees of freedom as vibrational modes.”

Ref: Wang, Vei, et al. "VASPKIT: a user-friendly interface facilitating high-throughput computing and analysis using VASP code." *Computer Physics Communications* (2021): 108033.

2. Can the authors clarify whether the catalysts after continuous mode were exposed to air prior to XPS. If yes, what procedures were used to minimize the exposure? If the spent catalysts were exposed to air, this reviewer recommends a control where the authors can show how the XPS with limited exposure times compare with the data shown.

Response: Thank you very much for this point. The V_2O_5 catalysts after continuous mode was not exposed to air prior to XPS measurement. After the continuous reactivity test, the catalyst was purged in an inert environment (Ar) followed by cooling to room temperature. Then, the reactor was transferred into an argon-filled glove box without exposure to air with both ends sealed. The sample loadings were handled in a glove box filled with argon, and the sample holder was then sealed to minimize the Air exposure.

3. R10 is an important figure in my view to be included in SI and the relevant discussion is recommended to be added. The comparison with the XPS in main text is suggested to be included.

Response: Thank you very much for this point. We have added Fig. R10 in the supporting information in Fig. S7 with the relevant discussion. Per your suggestion,

the comparison with the XPS in main text is also included.

Thank you once again for your valuable comments and suggestions. We are grateful to you for taking the time and energy to help us improve the paper.

References

1. Z. Cao, H. Jiang, H. Luo, S. Baumann, W. A. Meulenbergh, H. Voss and J. Caro, *ChemCatChem*, 2014, **6**, 1190-1194.
2. G. Biaisque and Y. Schuurman, *Journal of Catalysis*, 2010, **276**, 306-313.
3. A. Bielański and J. Haber, *Catalysis Reviews*, 1979, **19**, 1-41.
4. L. Qin, Z. Cheng, M. Guo, M. Xu, J. A. Fan and L.-S. Fan, *ACS Energy Letters*, 2017, **2**, 70-74.
5. M. Keller, H. Leion and T. Mattisson, *AIChE J.*, 2016, **62**, 38-45.
6. M. Rydén, M. Johansson, A. Lyngfelt and T. Mattisson, *Energy Environ. Sci.*, 2009, **2**, 970.
7. A. Thursfield, A. Murugan, R. Franca and I. S. Metcalfe, *Energy Environ. Sci.*, 2012, **5**, 7421.
8. M. Kathe, A. Empfield, P. Sandvik, C. Fryer, Y. Zhang, E. Blair and L.-S. Fan, *Energy Environ. Sci.*, 2017, DOI: 10.1039/c6ee03701a.
9. X. Zhu, Q. Imtiaz, F. Donat, C. R. Müller and F. Li, *Energy Environ. Sci.*, 2020, DOI: 10.1039/c9ee03793d.
10. J. Perezramirez, E. Kondratenko, V. Kondratenko and M. Baerns, *Journal of Catalysis*, 2004, **227**, 90-100.
11. J. Perezramirez, E. Kondratenko, V. Kondratenko and M. Baerns, *Journal of Catalysis*, 2005, **229**, 303-313.
12. L. A. Isupova, E. F. Sutormina, N. A. Kulikovskaya, L. M. Plyasova, N. A. Rudina, I. A. Ovsyannikova, I. A. Zolotarskii and V. A. Sadykov, *Catalysis Today*, 2005, **105**, 429-435.
13. O. S ˇ ipr, A. S ˇ imu ˇ nek, S. Bocharov, Th. Kirchner and G. Dra ˇ ger, *Physical Review B*, 1999, **60**, 14115-14127.
14. T. D. Tullius, W. O. Gillum, R. M. K. Carlson and K. O. Hodgson, *J. Am. Chem. Soc.*, 1980, **102**, 5670-5676.
15. T. Yamamoto, *X-Ray Spectrometry*, 2008, **37**, 572-584.
16. V. Wang, N. Xu, J.-C. Liu, G. Tang and W.-T. Geng, *Computer Physics Communications*, 2021, **267**.

REVIEWERS' COMMENTS

Reviewer #2 (Remarks to the Author):

The proposed changes improve the manuscript and the paper can be accepted.

Reviewer #3 (Remarks to the Author):

The authors have addressed all of my comments and in my view the article is suitable for publication.

RESPONSES TO REVIEWER'S COMMENTS:

Reviewer #2 (Remarks to the Author):

The proposed changes improve the manuscript and the paper can be accepted.

Response: Thank you very much. We appreciate you taking the time to offer us your insightful comments and constructive suggestions.

Reviewer #3 (Remarks to the Author):

The authors have addressed all of my comments and in my view the article is suitable for publication.

Response: Thank you very much. We appreciate you taking the time to offer us your insightful comments and constructive suggestions.